# Multiplexed quantitative proteomics provides mechanistic cues for malaria severity and complexity

Vipin Kumar[1], Sandipan Ray [1,7,8], Shalini Aggarwal[1,8], Deeptarup Biswas[1], Manali Jadhav[1], Radha Yadav[2], Sanjeev V. Sabnis[2], Soumaditya Banerjee[3], Arunansu Talukdar[3], Sanjay K. Kochar[4], Suvin Shetty[5], Kunal Sehgal[6], Swati Patankar[1] & Sanjeeva Srivastava[1✉]

Management of severe malaria remains a critical global challenge. In this study, using a multiplexed quantitative proteomics pipeline we systematically investigated the plasma proteome alterations in non-severe and severe malaria patients. We identified a few parasite proteins in severe malaria patients, which could be promising from a diagnostic perspective. Further, from host proteome analysis we observed substantial modulations in many crucial physiological pathways, including lipid metabolism, cytokine signaling, complement, and coagulation cascades in severe malaria. We propose that severe manifestations of malaria are possibly underpinned by modulations of the host physiology and defense machinery, which is evidently reflected in the plasma proteome alterations. Importantly, we identified multiple blood markers that can effectively define different complications of severe falci-parum malaria, including cerebral syndromes and severe anemia. The ability of our identified blood markers to distinguish different severe complications of malaria may aid in developing new clinical tests for monitoring malaria severity.

[1] Department of Biosciences and Bioengineering, Indian Institute of Technology Bombay, Mumbai 400076, India. [2] Department of Mathematics, Indian Institute of Technology Bombay, Mumbai 400076, India. [3] Medicine Department, Medical College Hospital Kolkata, 88, College Street, Kolkata 700073, India. [4] Department of Medicine, Malaria Research Centre, S.P. Medical College, Bikaner 334003, India. [5] Dr. L H Hiranandani Hospital, Mumbai 400076, India. [6] Sehgal Path Lab, Mumbai 400053, India. [7] Present address: Department of Systems Pharmacology and Translational Therapeutics, University of Pennsylvania, Philadelphia, PA 19104, USA. [8] These authors contributed equally: Sandipan Ray, Shalini Aggarwal. ✉email: sanjeeva@iitb.ac.in

Malaria is a vector-borne infectious disease caused by the protozoan parasites of the *Plasmodium* genus[1], and the vector involved is female *Anopheles* mosquito. It is the most widespread tropical parasitic disease with a worldwide occurrence of 228 million clinical cases and 0.4 million deaths in 2018 (ref. [2]). India majorly contributes to the global malaria burden and has the largest population in the world at risk of malaria, with 85% of the total Indian population living in malarious zones[3]. Worrisomely, the subsidiary burdens of malaria, such as malnutrition and anemia, increase the risk of complications and severity of the disease[4,5]. Among the five parasites causing malaria in humans, *Plasmodium falciparum* and *Plasmodium vivax* have the most extensive global distributions and are capable of leading severe fatal clinical manifestations[6]. Particularly, *P. falciparum* infections often turn severe and life-threatening, specifically when managed inappropriately[7,8]. One of the prime causes behind the progression of this parasitic infection from mild through complicated to severe disease is missed or delayed diagnosis[9,10]. Of note, neither parasite density nor parasitemia can consistently define malaria severity[11]. Hyper-parasitemia does not necessarily provide primary prognostic significance in semi-immune individuals, as they often tolerate high parasitemia burden without any physiological signs of disease or severe effects[12].

There are several limitations for the existing diagnostic methods for malaria, which include microscopic examination of thick and thin blood smears, polymerase chain reaction (PCR)-based molecular diagnostics, and rapid diagnostic tests (RDTs)[13–16]. Additionally, the RDT-based detection approaches target PfHRPII for *P. falciparum* only, pLDH for all *Plasmodium* species, and aldolase for *P. falciparum* and *P. vivax*. Importantly, many population-based studies have shown partial or complete deletion of *pfhrpII* and *pfhrpIII* genes, which may lead to false-negative results in RDTs[17–19]. Overall, these issues amply highlight the need for identification of additional parasite proteins and host factors for better diagnosis of malaria. Moreover, the establishment of predictive blood biomarkers for malaria severity and complications would be highly promising for prognosis, monitoring disease progression and responses to therapy, and predicting outcomes.

In recent years, plasma/serum proteomics studies have contributed substantially to elucidate the complex pathogenesis of malaria and other infectious diseases[17,18]. Intriguingly, several studies from our and other research groups have defined promising panels of plasma and serum markers in falciparum[19–24] and vivax[19,25,26] malaria. A considerable amount of further research is required to understand the complex pathophysiology of severe malaria[27–29]. In particular, the blood markers that can effectively define different complications of severe malaria are not clearly demarcated hitherto. Indeed, the precise mechanisms that cause a transition from non-severe to severe fatal clinical manifestations in malaria, which often happens very rapidly, remains largely obscure. These are the critical questions that remained to be addressed to understand the molecular basis of severe malaria.

In this study, we performed comprehensive proteomics analysis of plasma samples from falciparum malaria patients with different severity levels and clinical manifestations to understand the mechanisms of malaria severity and its complications. We further carried out a comparative analysis with differentially abundant plasma proteins identified in vivax malaria (VM) and dengue patients to specify the alterations observed in falciparum malaria. Importantly, we identified several blood-based host proteins marker for malaria severity and complexity, which can aid in monitoring disease progression. Further, we have also identified parasite proteins such as Serine Repeat Antigen 4 and Fructose-Bisphosphate Aldolase in severe malaria patients'

plasma samples, which can help in the diagnosis of severe malaria. To the best of our knowledge, this is one of the most comprehensive blood-based proteomic studies on falciparum and VM. Our findings provided insights regarding malaria pathogenesis and molecular cues of severity and complicated disease manifestations associated with this parasitic infection.

## Results

**Analysis of clinicopathological parameters of malaria and dengue patients**. A total of 98 subjects were analyzed in the discovery-phase quantitative proteomics, while 111 subjects were included in the targeted validation study. The following cohorts were included in the present study—Healthy control (HC), Non-severe falciparum malaria (NSFM), Severe falciparum malaria (SFM), Cerebral malaria (CB), Severe anemia (SA), Control for severe anemia (CSA), Control for cerebral malaria (CCB), Non-severe dengue fever (NSD), Severe dengue (SD), Non-severe vivax malaria (NSVM), and Severe vivax malaria (SVM). The subjects were recruited from different malaria epidemic regions in India. The number of samples analyzed for each group is provided in Supplementary Data 1. Dengue patients were incorporated in this study as non-malaria febrile infectious disease control. In falciparum malaria patients, platelet counts and Hb levels were found to be significantly lower ($p < 0.05$) (in both NSFM and SFM) as compared to HC. The magnitude of alterations (decrease) in platelet counts and hemoglobin (Hb) levels were found to be more prominent in severe malaria. Liver function parameters including total bilirubin, serum glutamic-oxaloacetic transaminase (SGOT), serum glutamic-pyruvic transaminase (SGPT), and alkaline phosphatase (ALP) were found to be higher in SFM patients as compared to NSFM and HC (Supplementary Fig. 1). Of note, Hb was significantly lower ($p < 0.05$) in SA as compared to CB and other different types of complications of SFM, while the other parameters were almost comparable. These liver function parameters were slightly higher in non-severe malaria as compared to HC, but the level of alteration was minimal (Supplementary Fig. 2). Lower blood levels of Hb were observed in malaria patients [both falciparum malaria (FM) and VM], but no significant alteration was observed in dengue fever (DF) as compared to HC. Platelet count was found to be extremely low in DF as compared to malaria and HC, while SGOT and SGPT levels were found to be higher in DF as compared to HC and malaria (FM and VM). ALP and total bilirubin levels were very high in FM patients compared to all the other study cohorts (HC, VM, and DF) (Supplementary Fig. 3).

**Workflow for comprehensive plasma proteomic analysis of malaria and dengue patients**. Malaria and dengue samples were confirmed using different diagnostic techniques, and the positive cases were incorporated in the quantitative proteomic analysis. Such multiplexing using stable isotope labeling provides increased throughput, higher precision, better reproducibility, reduced technical variations, and lower number of missing values[30–32]. TMT-based multiplexed quantitative proteomics was used to map the plasma proteome (host) alterations, while we used a label-free quantitation (LFQ) approach for detection and quantification of the parasite (*P. falciparum*) proteins in host plasma. Quality control check (QC) of the datasets was performed by plotting the density plots for FM, VM, and DF using raw and normalized abundances at a proteome scale (Supplementary Fig. 4a–c). The significantly ($p < 0.05$) altered proteins were considered for machine learning, and the elastic net regularized logistic regression method was applied to predict the best panel of proteins. Some selected targets were validated using mass spectrometry (MS)-based multiple reaction monitoring (MRM) assays.

Eventually, we investigated the physiological pathways over-expressed in falciparum and VM (Fig. 1).

**Differential plasma proteomic analysis of non-severe and severe FM.** A comprehensive proteomic analysis was performed for FM patients. Overall, 239,034 peptide spectral matches (PSMs) and 11,530 peptides were identified corresponding to 1495 proteins identified combining all the samples, among which 296 proteins were present in more than 60% of the clinical samples used for the label-based FM study (Fig. 2a). The proteins that were quantified with ≥1 unique peptide and detected in at least 60% of the samples were selected for the subsequent differential analysis. Importantly, 91% of these proteins (271 out of 296) were quantified with ≥2 unique peptides. Twenty-five plasma proteins were found to be significantly ($p < 0.05$) altered in SFM as compared to NSFM, and their abundance profiles enabled us to differentiate SFM and NSFM (Fig. 2b, Supplementary Fig. 5a, and Supplementary Data 2). Partial least-squares discriminant analysis (PLS-DA) was performed on the differentially abundant proteins ($p < 0.05$), and it distinguished between the three study populations: HC, NSFM, and SFM (Fig. 2c and Supplementary Table 1). Supervised hierarchical clustering of the proteome profiles stratifies the different groups of malaria patients (Fig. 2d). Significantly altered proteins from this study and other published literature[20,21] were considered for protein network analysis (Fig. 2e and Supplementary Data 2) using NetworkAnalyst.

The platelet degranulation process was found to be highly active in FM. Plasminogen (PLG), Platelet factor 4 (PF4), Profilin-1 (PFN1), Kininogen-1 (KNG1), and Pro-platelet basic protein (PPBP) were found to be down-regulated in NSFM as compared to SFM, while Alpha1-antitrypsin (SERPINA1) and Alpha2-antiplasmin (SERPINF2) were dysregulated in SFM as compared to NSFM (Fig. 2e). Proteins involved in binding and uptake of ligand scavenging functions such as haptoglobin (HP), hemopexin (HPX), haptoglobin-related proteins (HPR), and protein AMBP (AMBP) were down-regulated in SFM as compared to NSFM. However, plasma levels of several proteins such as hemoglobin subunit delta (HBD), hemoglobin subunit beta (HBB), hemoglobin subunit alpha (HBA1), and carbonic anhydrase 1 (CA1) were almost similar in SFM and NSFM (Fig. 2e). Gene Ontology (GO) analysis revealed that several vital biological processes such as immune system activation and response to stimulus involved more entities (altered proteins) in NSFM as compared to SFM. Biological adhesion and multicellular organismal processes were over-expressed in SFM as compared to NSFM (Supplementary Fig. 6a). Molecular functions associated with the altered proteins in NSFM are mainly regulatory and catalytic activity, while binding, molecular transducer, and transcription activities were enriched in SFM (Supplementary Fig. 6b). The cellular component analysis revealed overexpression of the extracellular region and cell in NSFM as compared to SFM. At the same time, protein-containing complex and membrane were over-expressed in SFM (Supplementary Fig. 6c).

**Differential plasma proteomic analysis of multiple complications of severe FM.** Plasma proteomics of different complications of SFM such as severe anemia ($n = 6$) and cerebral malaria (CB; $n = 6$) exhibited 28 and 35 differentially abundant proteins ($p < 0.05$), respectively, in comparison to HC and 17 proteins in SA as compared to CB (Fig. 3a, Supplementary Table 2 and Data 2), and heat map profiles were able to distinguish the individual samples of CB and SA (Supplementary Fig. 5b). Five proteins in CB, 8 proteins in SA, and 15 proteins in other severe malaria cases were found to be specifically altered in various complications of FM

(Fig. 3b). 3D-PLS-DA plot of the differentially abundant proteins was able to segregate CB, SA, and SFM with other complications (Fig. 3c). Expectedly, the significantly altered proteins ($p < 0.05$) can effectively distinguish CB or SA as compared to HC (Fig. 3d, e). Supervised clustering indicated that a group of proteins could differentiate among the different complications of SFM. Cluster 1 and 2 showed differentially altered proteins in severe falciparum malaria (SFM_Others + CB + SA) as compared to the disease controls (CSA and CCB). Cluster 3 showed down-regulated proteins in CB, while cluster 4 showed down-regulated proteins in SA as compared to others (Fig. 3f). Hemostasis, ligand scavenging, and complement cascades were found to be associated with the differentially abundant plasma proteins identified in SFM_Others, CB, and SA (Fig. 3g).

Quantitative proteomics analysis of CB patients showed 12 proteins significantly altered ($p < 0.05$) as compared to NSFM. Most of these proteins were found to be down-regulated in CB. Out of these 12 altered proteins, 3 candidates—HGFA[33], SERPINA3, and ALX1 protein—were found to be up-regulated, while the other 9 proteins were down-regulated as compared to NSFM. Five proteins were significantly altered, specifically in CB as compared to NSFM. Of note, these proteins were unaltered in meningitis patients [a control used for cerebral malaria (CCB) (Supplementary Data 2)].

Quantitative proteomics analysis of SA patients showed 12 significantly altered proteins as compared to NSFM ($p < 0.05$). Most of these proteins were found to be down-regulated in SA. Eight proteins, including C1QB, TF, SAA1, IGFBP3, C3, APOE, CYST3, and B2M were significantly altered in SA only. Importantly, plasma levels of these proteins were not altered in anemic subjects [CSA] (Supplementary Data 2).

**Differential plasma proteome landscape of non-severe and severe VM.** Further, we performed comparative proteomic analysis on another form of malaria caused by *P. vivax*. The abundance profile of each sample of HC, NSVM, and SVM showed uniform distribution across all the TMT 10-plex reactions (Supplementary Fig. 4b). We found 23 and 37 differentially abundant proteins ($p < 0.05$) in NSVM and SVM as comparison to HC and 45 differentially abundant proteins ($p < 0.05$) in SVM as compared to NSVM, respectively (Fig. 4a). Heat map profiles of these altered proteins were able to distinguish individual samples of NSVM and SVM (Supplementary Fig. 7). Altered proteins identified in *P. vivax* patients segregated into three groups of clinical conditions, as shown in the PLS-DA plot (Fig. 4b). Correlation analysis of the altered proteins ($p < 0.05$) in NSVM, SVM, and DF also showed three distinct clusters (Fig. 4c).

Based on the abundance profiles of the altered proteins, we were also able to effectively define SVM, NSVM, and DF (Fig. 4d). A comparison of these three groups generated three distinct clusters. Cluster 1, 2, and 3 are representing the up-regulated proteins in SVM, NSVM, and DF, respectively (Fig. 4d). The differentially abundant proteins were enriched to four major pathways contributing towards VM, viz. complement cascade, and platelet degranulation along with platelet activation, lipid metabolism, and hemostasis. In complement pathways, Complement C1q subcomponent subunit A (C1QA) was found to be up-regulated significantly in NSVM, whereas Complement C3 (C3) was significantly down-regulated in both NSVM and SVM patients. In the platelet degradation pathway, fructose-bisphosphate aldolase (ALDOA), Clusterin (CLU), and PLG were significantly down-regulated in both NSVM and SVM, whereas Serotransferrin (TF) exhibited significant up-regulation in all the VM patients. Furthermore, Alpha2-macroglobulin (A2M) exhibited up-regulation in NSVM, while its down-

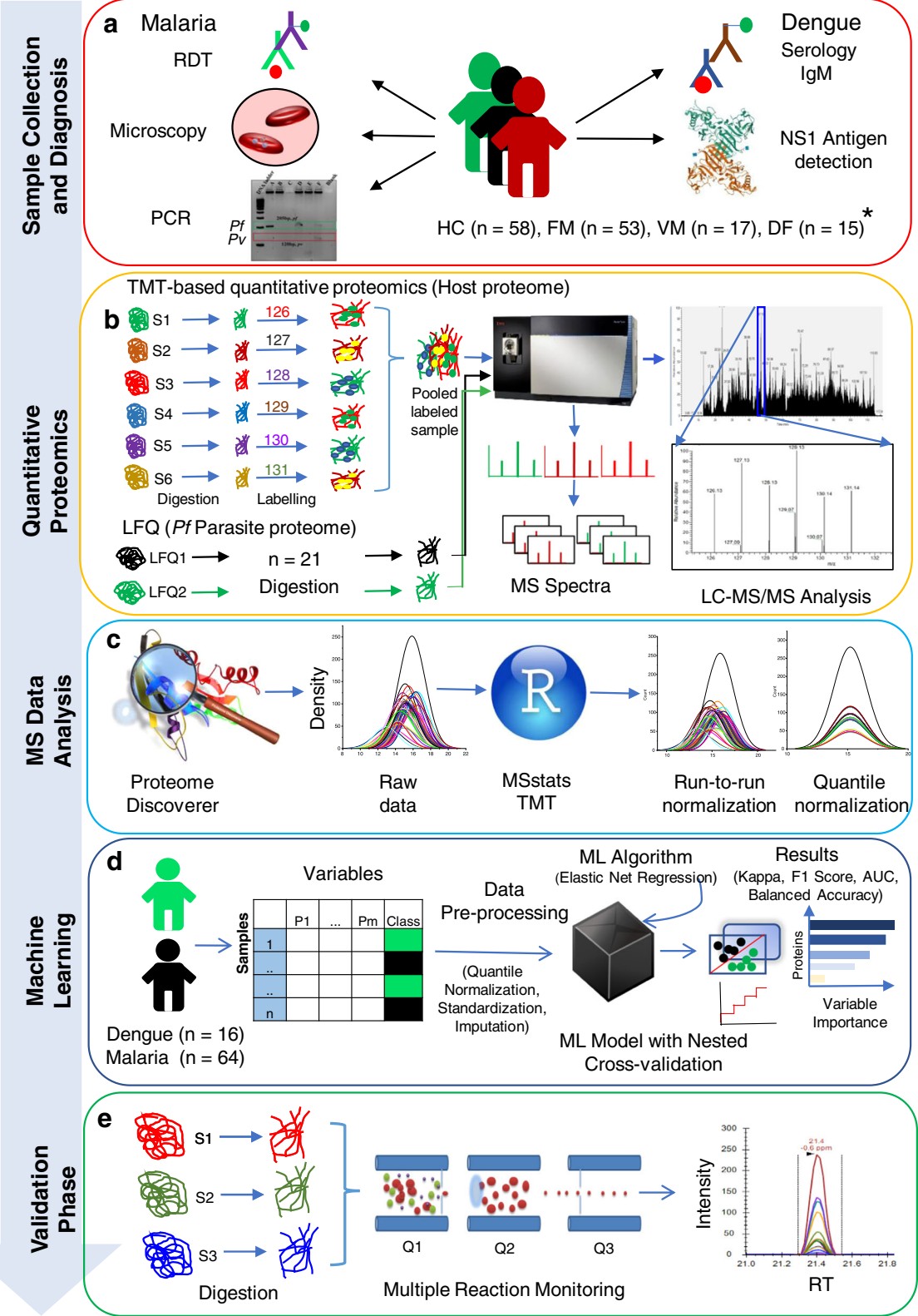

**Fig. 1 Schematic representation of the experimental strategy used in discovery-phase proteomics and for targeted validation of potential biomarkers in plasma samples. a** Malaria patients were diagnosed by microscopy and RDT, and some randomly selected samples were confirmed further using PCR. Dengue patients were included as a non-malaria febrile infectious disease control and were diagnosed using IgM and NS1 antigen. **b** Depleted plasma samples were trypsin digested, and TMT labeled for studying the plasma proteome (host) alterations in malaria patients. A label-free quantitation (LFQ) approach was used for the detection and quantification of the parasite (*P. falciparum*) proteins in host plasma. **c** Protein search was performed using proteome discoverer 2.2, and then PSMs files were analyzed using the MSstatsTMT package. **d** Machine learning was performed using the elastic net regression method. **e** Top hits of differentially abundant proteins were selected for validation using multiple reaction monitoring (MRM) assays. *Patients number is based on available clinical data.

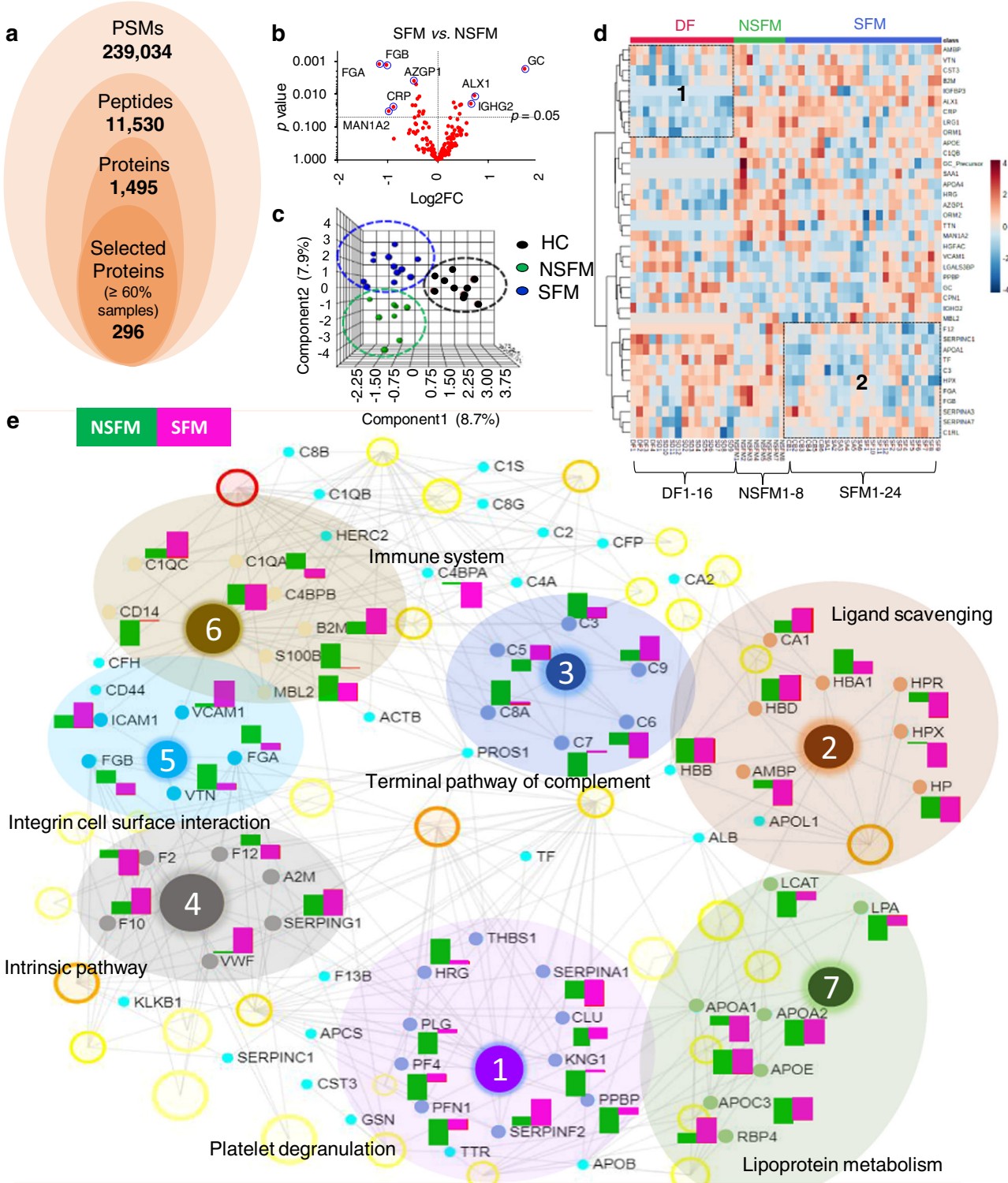

**Fig. 2 Comprehensive quantitative plasma proteomics analysis of non-severe and severe falciparum malaria. a** Overview of the proteome coverage obtained in TMT-based quantitative analysis. **b** Differentially abundant proteins ($p < 0.05$) in SFM as compared to NSFM. **c** Three-dimensional PLS-DA plot showing clear segregation among HC, NSFM, and SFM. **d** Heat map representation showing abundances of the altered proteins in NSFM, SFM, and DF, and different clusters were identified as 1, 2, and 3 on the basis of protein abundance in DF, malaria (NSFM+SFM) and SFM, respectively. **e** Physiological pathways associated with the differentially abundant plasma proteins identified in NSFM and SFM. Sequential numbering for the networks is provided based on the statistical significance (high to low significance, FDR < 0.05). NSFM non-severe falciparum malaria, SFM severe falciparum malaria, DF dengue fever, HC healthy control, CSA control for severe anemia malaria, CCB control for cerebral malaria.

regulation was observed in SVM. Additionally, components of the lipid metabolism pathways, such as Apolipoprotein A-I (APOA1) and Apolipoprotein B-100 (APOB) were significantly down-regulated in both NSVM and SVM (Fig. 4e and Supplementary Data 3). The altered plasma proteins identified in NSVM were involved in the immune system process, metabolic process, and localization, while the proteins altered in SVM were associated with response to stimulus and diverse cellular processes (Supplementary Fig. 8a). Prime molecular functions for the majority of the altered proteins were catalytic activity and binding in SVM and NSVM, respectively (Supplementary Fig. 8b). GO analysis indicated almost similar cellular localizations for the

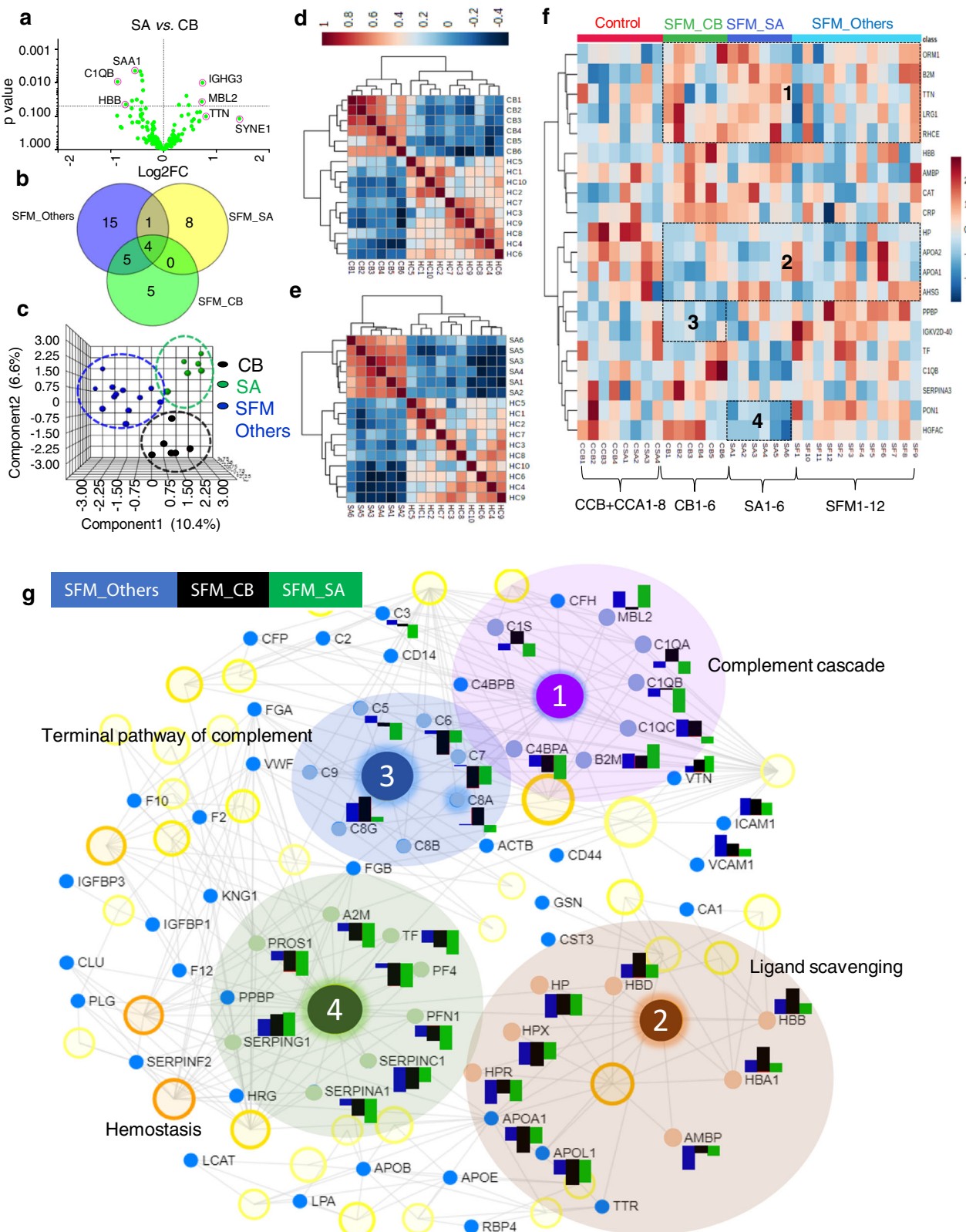

**Fig. 3 Comparative proteomic analysis of different complications of severe falciparum malaria. a** Differentially abundant proteins ($p < 0.05$) in severe malaria anemia (SA) as compared to cerebral malaria (CB). **b** Three-dimensional PLS-DA plot showing effective segregation among the different complications of SFM (CB, SA, and SFM with other complications). **c** Common and specific differentially altered proteins in CB, SA, and SFM with other complications. **d** Correlation analysis of differentially altered proteins in CB as compared to HC. **e** Correlation analysis of differentially altered proteins in SA as compared to HC. **f** Heat map representing discrimination of plasma protein abundances in CB, SA, and SFM with other complications, and their negative controls (CCB+CSA). Different clusters were identified as 1 and 2 in severe malaria (CB+SA+SFM), 3 in CB, and 4 in SA on the basis of proteins abundance. **g** Physiological pathways associated with the differentially abundant plasma proteins identified in severe falciparum, cerebral, and severe anemia malaria. Sequential numbering for the networks is provided on the basis of the statistical significance (high to low significance, FDR < 0.05). SFM_Others: severe falciparum malaria, except CB and SA.

altered proteins in NSVM and SVM, except the extracellular region, which was more prominent in NSVM (Supplementary Fig. 8c).

We identified eight proteins as commonly altered candidates in NSFM and NSVM. Of note, several vital physiological pathways/ biological processes such as platelet degranulation, response to elevated platelet cytosolic Ca$^{2+}$, and platelet activation, signaling and aggregation pathways are associated with these eight commonly altered plasma proteins identified in NSFM and NSVM (Supplementary Table 3).

**Identification of *P. falciparum* proteins in host plasma samples.** We identified 23 parasite proteins in the plasma samples of SFM patients ($n = 21$) using an LFQ approach. (Supplementary Data 4). We further selected 10 proteins that were identified with ≥2 unique peptides for validation using the MRM approach (Table 1). Six out of these 10 proteins—heat-shock protein (HSP) 70 and HSP 90, enolase, actin I, fructose-bisphosphate aldolase (FBA), and serine repeat antigen 4 (SERA4) were detected consistently (>80%) in the malaria patients. These parasite proteins have catalytic activity and may play a vital role in the survival and virulence of the pathogen in the host system.

HSP 70 and HSP 90 were reported as up-regulated at temperature 38 °C and above, which helps the survival of the parasite in the erythrocytic stage of its life cycle in the hyperthermic condition of the host[34]. Enolase along with HSP 70 and iron superoxide dismutase forms the DegP complex which protects the parasite from heat and oxidative stress in the host system[35]. FBA and actin have been reported to interact with TRAP and TRAP like protein (TLP) for sporozoites gliding and invasion[36]. SERA4, along with the other SERA member proteins, helps in maintaining the blood stage of the pathogen's life cycle. However, their clear physiological functions still remain unknown[37], and need to be investigated further.

**Elastic net regularized logistic regression model for feature selection to predict malaria and its complications.** Machine learning was performed on the significantly altered ($p < 0.05$) plasma proteins identified in malaria and dengue. Of note, we are able to separate VM, FM, dengue, and HC samples based on the abundance profiles of the altered plasma proteins (Supplementary Fig. 9a). The elastic net regularized logistic regression method was used to classify: dengue vs. malaria, FM vs. VM, and cerebral vs. severe malaria anemia (Fig. 5a).

The elastic net regularized logistic regression model hyper-parameters (alpha and lambda) and performance metrics (balanced accuracy, *F*1-score and Kappa) for the three models were as follows: (i) dengue vs. malaria: 0.1, 0.26, 0.975, 0.967, and 0.96; (ii) FM vs. VM: 0, 80.2, 1, 1, and 1; and (iii) CB vs. SA: 0.04, 41.73, 1, 1, and 1, respectively. It is evident from the model performance metrics that the resulting elastic net regularized logistic regression model can predict and classify dengue vs. malaria, FM vs. VM, and cerebral vs. severe malaria anemia cases

almost perfectly. These results are quite fairly stable across all (outer) *k*-fold iterations (Supplementary Data 5).

A total of 44 proteins were selected out of 66 proteins, which could differentiate malaria (including FM and VM) from dengue patients. The three-dimensional PLS-DA plot for all altered proteins and heat map profiles of the top 20 significantly altered proteins ($p < 0.05$) across malaria and Dengue effectively differentiated between these two infections (Fig. 5b, e). We found 49 proteins, which can distinguish FM from VM; importantly, unsupervised clustering of the top 20 significantly altered proteins effectively separated the individual subjects in the respective groups (Fig. 5c, f). Intriguingly, 19 proteins were able to differentiate between cerebral and severe malaria anemia (Fig. 5d). However, these two different complications of SFM were not easily separable based on the clinicopathological parameters (Supplementary Fig. 2). The receiver operating characteristic (ROC) curves of six selected biomarkers indicate that the model performs very well with area under the ROC curve (AUC) > 0.8, even for a small number of proteins. There was no further improvement in AUC after including a larger number of proteins. Keeping parsimony in mind and using this as a criterion to decide the panel size, we defined the final panel of two biomarkers for each model using MRM-based mass spectrometric assays (Supplementary Data 5).

**Validation of potential protein biomarkers of malaria using MRM assays.** Finally, we validated the differential abundance of a few selected human (host) proteins and parasite proteins in plasma samples using MRM-based mass spectrometric assays. We have considered 7048 transitions corresponding to 86 human plasma proteins (403 unique peptides), which were identified as the best classifier in our machine learning analysis for the optimization of MRM assays. Initially, we started with 40 different method files having a maximum of ~200 transitions, then we refined our methods and finalized 847 transition corresponding to 128 peptides and 46 proteins (Fig. 6a). MRM assays involved HC ($n = 24 \times 3$), FM ($n = 30 \times 3$), VM ($n = 30 \times 3$), and DF ($n = 27 \times 3$) along with 15 synthetic heavy peptides. The measurements obtained in MRM correlated well with the TMT-based discovery-phase proteomics (Supplementary Data 6).

Similarly, we validated the detection and quantification of the parasite proteins in human plasma samples. Ninety-five transitions corresponding to eight parasite proteins were optimized, and we were able to quantify those in the plasma samples using MRM assays (Supplementary Data 6). We monitored the system suitability using a heavy synthetic peptide. The area of a few representative heavy synthetic peptides is shown in the Supplementary Information (Supplementary Fig. 10a–c). Top-ranked proteins were validated using the MRM approach. We analyzed ROC curves and determined AUC for all the proteins that were common in both machine learning model and MRM, and were able to differentiate between malaria vs. dengue (Fig. 6b), FM vs. VM (Fig. 6c) and cerebral vs. severe malaria anemia (Fig. 6d and Supplementary Fig. 9b–d). Differential

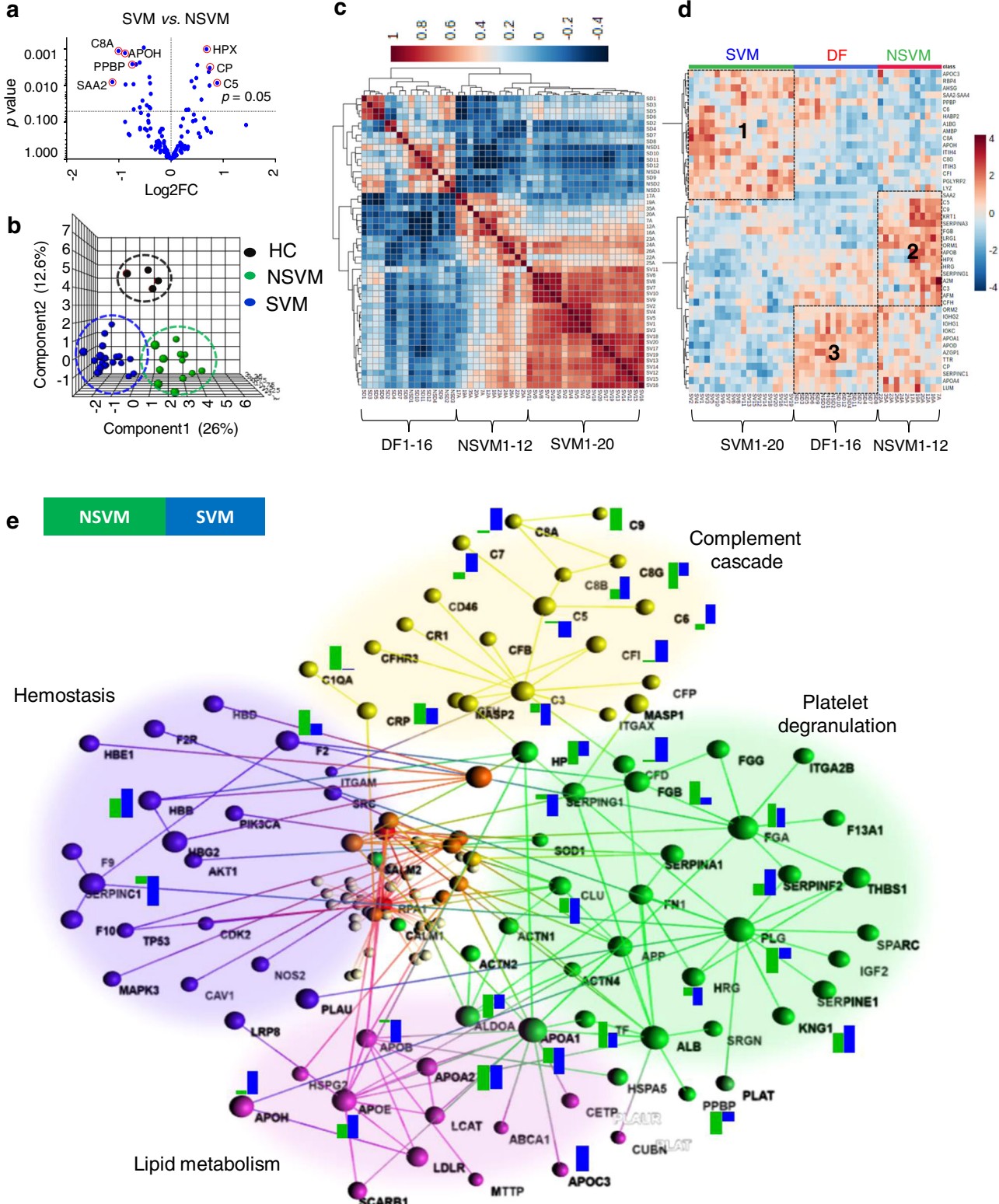

**Fig. 4 Differential plasma proteome maps of non-severe and severe vivax malaria and dengue. a** Differentially abundant proteins ($p < 0.05$) in non-severe (NSVM) and severe vivax malaria (SVM) as compared to healthy controls (HC). **b** PLS-DA plots segregated HC, SVM, and NSVM patients. **c** Correlation analysis of differentially altered proteins in SVM as compared to NSVM. **d** Heat map representing discrimination of plasma protein abundances in NSVM, SVM, and dengue fever (DF) and different clusters were identified as 1, 2, and 3 based on protein abundance in SVM, NSVM, and DF, respectively. **e** Physiological pathways associated with the differentially abundant plasma proteins identified in NSVM and SVM.

**Table 1 Identification of parasite proteins (*P. falciparum*) in plasma samples of severe falciparum malaria patients.**

| Accession | Description | # unique peptides | Biological process | Cellular component | Molecular function | Present in sample (*n* = 21) |
|---|---|---|---|---|---|---|
| PF3D7_1015900.1-p1 | Enolase | 7 | Cell organization and biogenesis; metabolic process; regulation of the biological process | Cell surface; cytoplasm; cytoskeleton; membrane; nucleus; vacuole | Catalytic activity; metal ion binding; protein binding | 20 |
| PF3D7_1444800.1-p1 | Fructose-bisphosphate aldolase | 6 | Cell organization and biogenesis; metabolic process | Cytoplasm | Catalytic activity; protein binding | 19 |
| P04933 | Merozoite surface protein 1 | 5 | | Membrane | | 10 |
| PF3D7_0708400.1-p1 | Heat-shock protein 90 | 4 | Metabolic process; response to stimulus | | Catalytic activity; nucleotide binding; protein binding | 21 |
| PF3D7_1246200.1-p1 | Actin I | 3 | Cell organization and biogenesis; cellular component movement | Cytoplasm; cytoskeleton | Nucleotide binding; structural molecule activity | 20 |
| PF3D7_0207700.1-p1 | Serine repeat antigen 4 | 3 | Metabolic process; regulation of biological process | Vacuole | Catalytic activity; protein binding | 17 |
| P04927 | S-antigen protein | 3 | | | | 8 |
| P11144 | Heat-shock 70 kDa protein | 2 | Cell organization and biogenesis | | Catalytic activity; nucleotide binding | 20 |
| PF3D7_0207600.1-p1 | Serine repeat antigen 5 | 2 | Metabolic process; regulation of biological process | Vacuole | Catalytic activity; protein binding | 12 |
| PF3D7_1035200.1-p1 | S-antigen | 2 | | | | 6 |

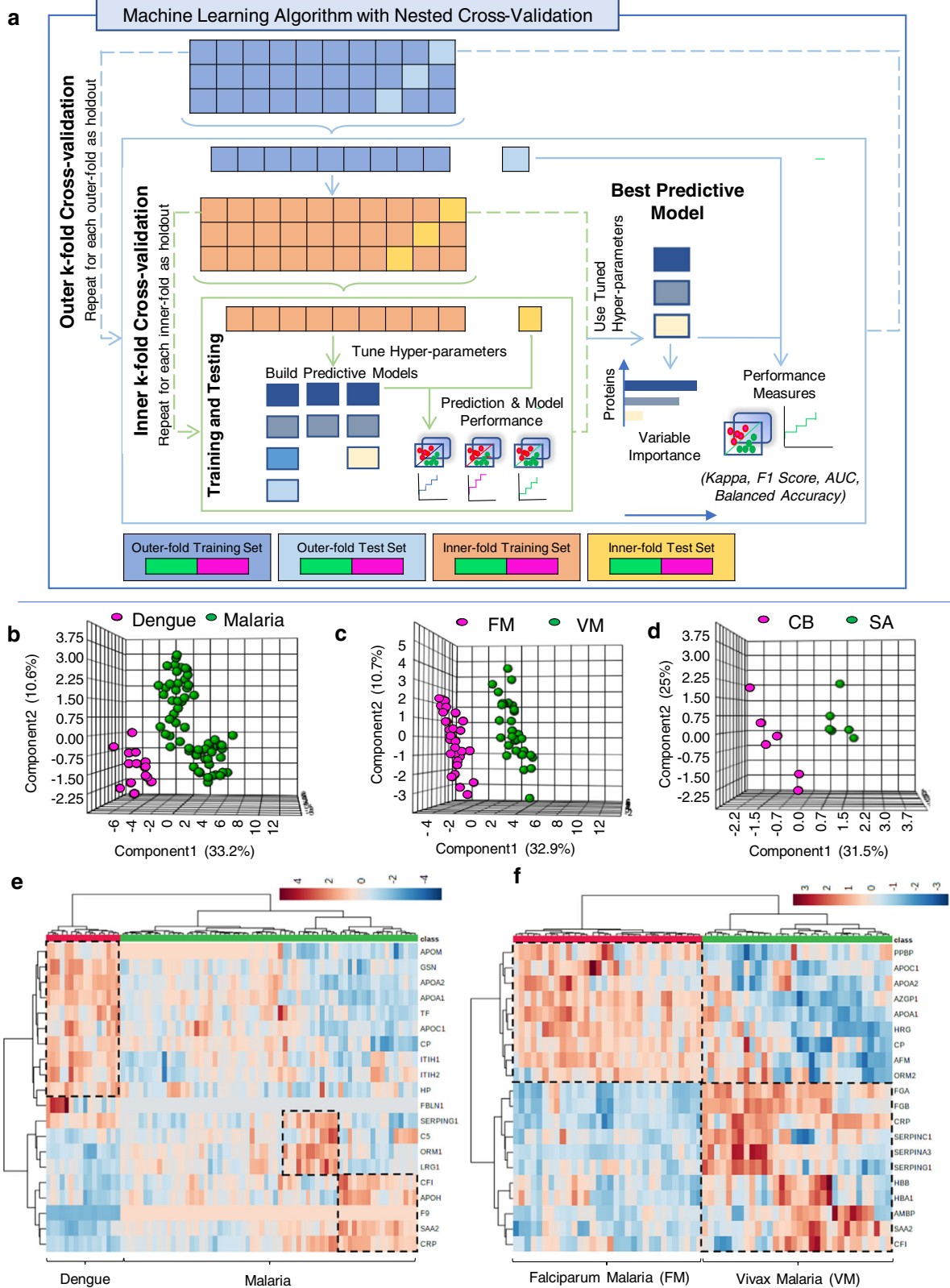

**Fig. 5 Machine learning model to identify biomarker signatures. a** Schematic overview of the machine learning model with nested *k*-fold cross-validation to identify the important proteins for effective diagnosis and prognosis of malaria and dengue. The three-dimensional PLS-DA plot of **b** dengue vs. malaria, **c** FM vs.VM, and **d** CB vs. SA. Heat map distribution of the top 20 (most significant) differentially abundant proteins—**e** malaria vs. dengue and **f** FM vs. VM.

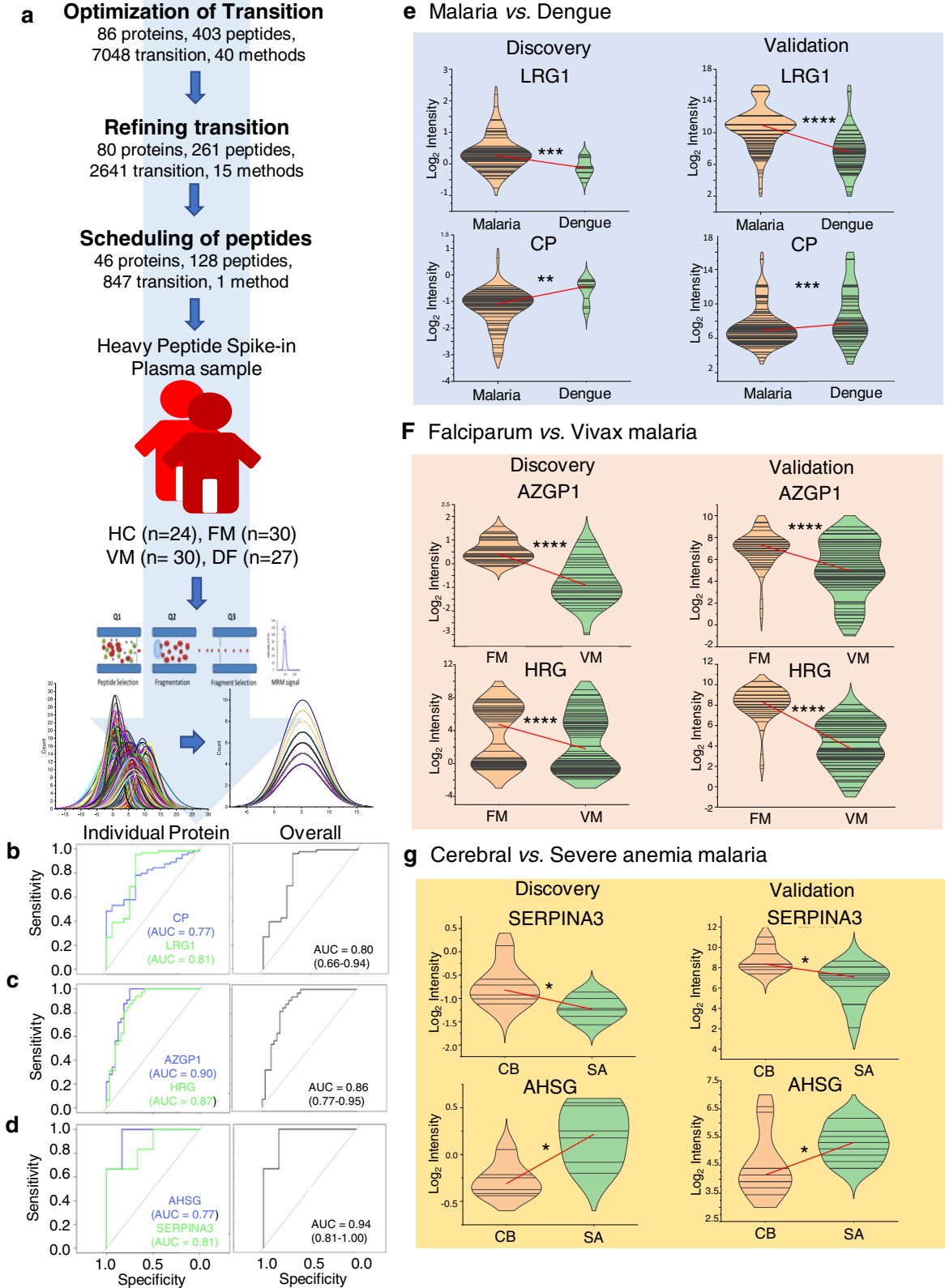

**Fig. 6 Validation of potential target proteins in malaria and dengue using MRM assays. a** Workflow for the validation of potential target proteins in malaria and dengue using MRM. Eighty-six proteins were considered for optimization for MRM method. ROC curves for the best panels of protein biomarkers are represented, **b** dengue vs. malaria, **c** falciparum vs. vivax, and **d** cerebral vs. severe anemia malaria. Violin plots showing the comparison of discovery and validation phase measurements for a few selected differentially abundant plasma proteins. **e** Dengue vs. malaria, **f** falciparum vs. vivax malaria, and **g** cerebral vs. severe anemia malaria. All data are represented as median (*$p \leq 0.05$, **$p \leq 0.01$, ***$p \leq 0.001$, and ****$p \leq 0.0001$) in a Student's *t*-test.

abundance of LRG1 and CP was observed between dengue and malaria using MRM assays as well as in TMT-based quantitation (Fig. 6e). Similarly, AZGP1 and HRG were able to distinguish between FM and VM (Fig. 6f), and SERPINA3 and AHSG were able to differentiate among the different complications of SFM (Fig. 6g). The elastic net regularized logistic regression model equations for the final biomarker models identified from MRM data are provided in Supplementary Methods.

## Discussion

In this study, using a multiplexed quantitative approach for plasma proteomics, we have provided insights into the progression of malaria from non-severe to severe infection and its different complications. Apart from the identification of potential disease monitoring and prognostic protein biomarkers for malaria, the differentially abundant plasma proteins mapped in the pathways provided mechanistic cues for various aspects of the pathogenesis of severe malaria and the host immune responses against the parasites. In this study, we identified the landscape of differentially abundant plasma proteins associated with diverse biological functions in malaria patients.

*P. falciparum* expresses *var* genes, which encodes erythrocytes membrane protein 1 (PfEMP1). The variants of PfEMP1 are involved in cytoadherence and mediate binding of the infected erythrocytes to endothelial vasculature[6]. Some of the parasite proteins that were identified in the plasma samples of SFM patients, such as FBA, HSP 70, HSP 90, enolase, and SRA4 can generate antibody responses, as we have shown previously in VM[38]. Similarly, Moussa et al.[24] reported the presence of FBA, SRA protein, and histone H3 in plasma samples of children with CB. Most of these proteins play a very important role in catalytic activity and protein binding (Table 1). However, *P. vivax* does not express var genes, and hence binding to the erythrocytes is very less as compared to *P. falciparum*[6]. We observed that the major complications associated with SFM are CB, SA, acidosis, and multiple organ failure. However, most of the complications observed in the SFM patients were similar to those observed in SVM, except renal failure, splenic rupture, and hepatic dysfunction along with gastrointestinal symptoms[39]. Of note, earlier we identified five unique parasite proteins in plasma and parasite isolates from VM patients[38].

Several altered plasma or serum proteins identified in SFM patients as described here and in our previous studies[20,26] were mapped to biological adhesion and extracellular matrix (Supplementary Fig. 6). It indicates the roles of cell-to-cell adhesion-related host proteins such as von Willebrand factor (vWF), ICAM-1, VCAM-1, VTN, and LGALS3BP in *P. falciparum* infection. We observed the up-regulation of these proteins in SFM, which may help in erythrocyte invasion and adherence to endothelial cells. Surprisingly, these proteins were not found to be altered in SVM, indicating some clear differences between these two plasmodial infections (Fig. 7). Similarly, we mapped the proteome of SVM and observed dysregulation of biological functions such as catalytic activity and cellular process along with the classical complement system being more active in SVM as compared to SFM (Supplementary Fig. 11).

Additionally, we investigated plasma proteome from various complications of SFM. Platelet degranulation acts as exocytosis cells and secretes a plethora of effector molecules at sites of vascular injury[40]. In our study, most of the proteins related to platelet degranulation were found to be down-regulated in FM. It indicates that this could be due to the highly active state of the immune system in the initial stage of FM (NSFM) possibly due to alterations in the levels of proteins such as PF4, PPBP, PFN1, KNG1, CLU, and PLG in NSFM (Fig. 2e). PF4 (CXCL4) acts as a

chemokine and initiates a killing of the infected erythrocytes in malaria[41–43]. Pro-platelet basic protein (PPBP), a platelet-secreted chemokine, and platelet activation marker that takes part in the process of clearing the parasites by inducing macrophage chemotaxis and mediating neutrophil accumulation[44] (Fig. 8a). This protein was found to be significantly dysregulated in malaria patients in our study. PFN1 is a cytoskeleton protein, and it is involved in actin-polymerization during the parasite invasion[45]. Clusterin, involved in innate immune response, has been reported to be down-regulated in malaria[23]. Many of the dysregulated plasma proteins identified in NSFM were strongly associated with immune responses (Supplementary Fig. 6a). Along with the findings obtained from plasma proteome profiling, hematological (Hb, platelets) and liver function (SGOT, SGPT, ALP, total bilirubin) parameters were found to be significantly altered in NSFM and SFM. It reflects the commencement of inflammatory responses in NSFM, which in turn proceeds towards the severity of the disease.

We observed up-regulation of the proteins related to intrinsic pathways such as vWF, F10, and F2 in SFM. Infected RBCs induce the expression of tissue factors on endothelial cells and monocytes, which results in the expression of cytokines via signaling pathways, ultimately leading to endothelial cell activation. vWF is a secondary marker for endothelial cell activation[46], and we observed a high plasma level of this protein in SFM. It binds to the activated platelets and the infected erythrocytes and promotes in sequestration of the infected erythrocytes. This causes activation of the coagulation system driving towards disease severity (Fig. 8b)[47]. Importantly, proteins related to cell-to-cell adhesion such as ICAM-1, VCAM-1, and VTN were found to be dysregulated in severe FM. Host proteins such as ICAM-1 and VCAM-1 are expressed on the endothelium of leukocytes[48–54] and are involved in sequestration of the infected erythrocytes in microvasculature with the assistance of a parasite-derived protein PfEMP1 (refs. [55,56]). This leads to the activation of endothelium, disruption of blood flow, and ultimately causes tissue hypoxia. These events lead to an increase of endothelial receptors, and therefore shedding of soluble endothelial receptors, eventually leading to endothelial damage[57]. We anticipate that this could be one of the possible mechanisms associated with the progression of NSFM to CB (Fig. 8c). Some of the up-regulated proteins in CB identified in our study were previously reported in the murine malaria model[48,58], which substantially enhances the strength of our findings. For instance, Bauer et al. reported the up-regulation of ICAM-1, P-selectin, and VCAM-1 on brain vascular endothelium in *P. berghei* ANKA infection. Up-regulated ICAM-1 levels may help the parasite sequestration on the epithelial cells causing brain injury and creating hypoxia conditions[49]. Blocking of the ICAM-1 receptor can cause a more than 150% increase of schizonts in the peripheral blood in mice. Importantly, VCAM-1 plays an important role in the resetting of the parasite to the blood vessels. However, it is limited to large blood vessels, unlike ICAM-1 (ref. [59]).

Severe anemia in malaria results mainly due to hemolysis and phagocytosis of the parasitized and the non-parasitized RBCs, and also to a large extent, by suppression of erythropoiesis that is driven by parasite hemozoin generated from the infected erythrocytes[60]. We observed increased levels of C3 which is an active component of the complement system. Complement C5, C6, and C7 form a complex c5c6c7 (ref. [61]), and bind to the cell membrane and membrane attack complex with the help of C8 and C9 (refs. [62,63]). In our study, we observed all of these three proteins forming the complex (C5, C6, and C7) were down-regulated while the free monomers (C8 and C9) were up-regulated in NSFM (Fig. 8d). This may indicate a higher involvement of complement cascade during the initial stage of malaria[61], leading

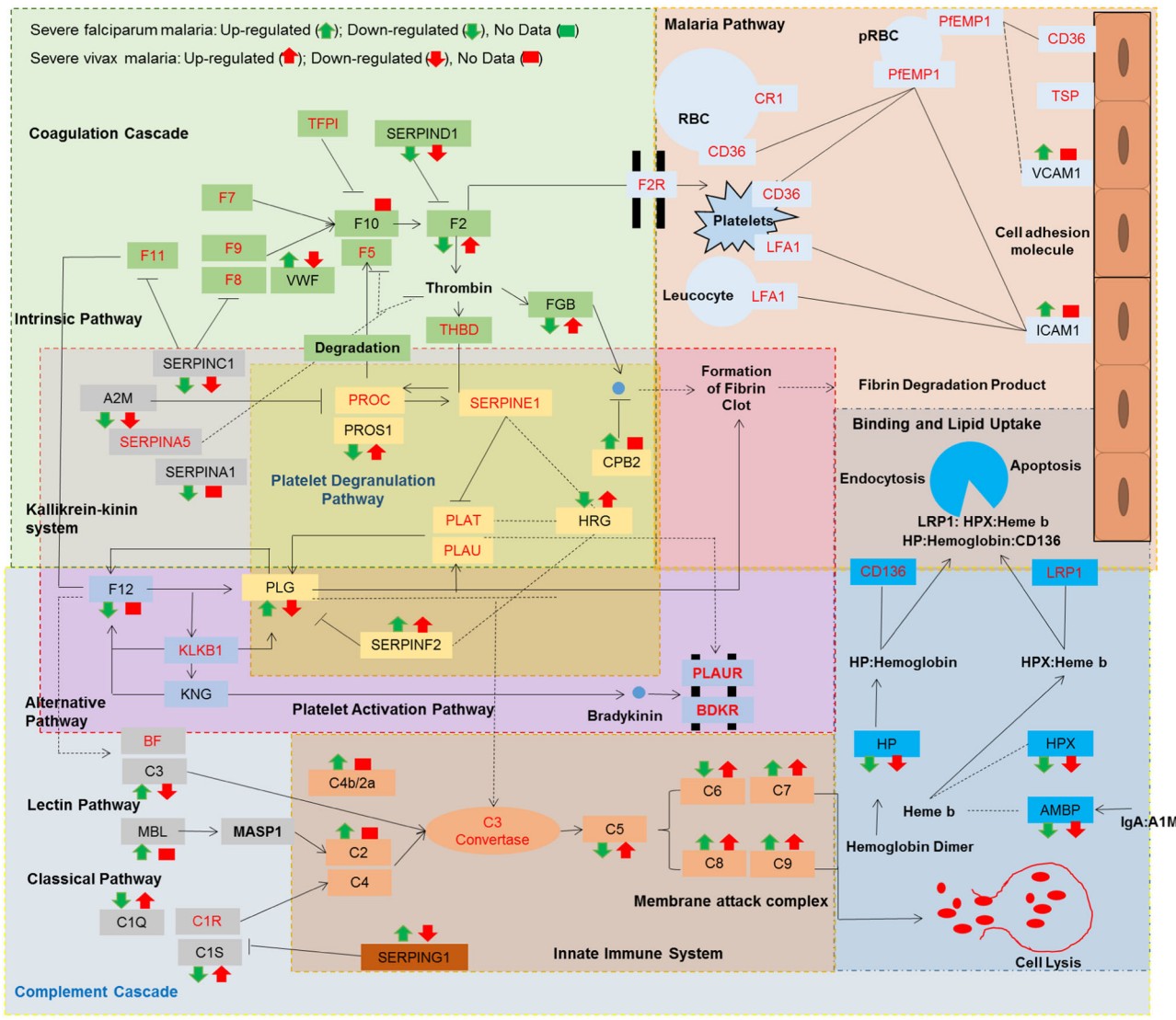

**Fig. 7 Landscape of physiological pathways altered in severe falciparum and vivax malaria.** Physiological pathways associated with the differentially abundant plasma proteins identified in severe falciparum malaria (SFM) and severe vivax malaria (SVM). The proteins written in black are identified in our study, while the red ones represent unaltered associated proteins within the pathways.

to hemolysis of infected erythrocytes. Such destruction of a high number of erythrocytes results in increased systemic concentrations of free Hb, heme, and reactive oxygen species (ROS) in circulation[39,64,65]. We observed a significant elevation in blood levels of Hb subunit alpha and beta in FM patients, as also described previously by Kassa et al.[23] The free Hb is oxidized by free radicals, thereby releasing free heme. Of note, free heme is very toxic to the cells and induces inflammation, macrophage activation, and oxidative stress. However, the down-regulation of these proteins (HP, HPX, HPR, and AMBP) indicate their involvement in scavenging of free heme and ROS in FM. RBC lysis and highly activated complement system may have direct effects on the destruction of erythrocytes and cytokines indirectly affect the inhibition of erythropoiesis. This clearly indicates an increased removal of the infected and uninfected erythrocytes that plays an essential role in severe anemia during malaria (Fig. 8d).

In summary, we provide here a comprehensive landscape of plasma proteome alterations in different severity levels of malaria. Our findings indicated the association of the dysregulated proteins with a few vital physiological pathways such as platelet

degranulation and integrin cell surface interaction, which in term explain the mechanisms of fatal complications in SFM. We observed that proteins related to the inflammatory system are highly dysregulated in NSFM, which may eventually cause the severity of infection leading to either cerebral syndrome or severe anemia. We also observed the alterations in the plasma levels of PF4 and PPBP, which help to destroy the infected erythrocytes and cytokines along with other mediators (hemozoin) and reduce the process of erythropoiesis[41,44]. This may eventually lead to severe anemia in malaria patients. Importantly, dysregulation of the hemostasis-related proteins was observed in severe anemia, while alterations in the blood levels of the proteins associated with endothelial cell activation strongly correlated with CB.

Parasite proteins that were consistently detected in almost all the severe malaria patients could be promising for developing different diagnostic approaches. Of note, those could be an indicator of severe infection as they are specifically detected in severe malaria patients. However, it remains to be seen whether these blood-based protein markers for malaria severity and complexity identified in our study are validated in bigger heterogeneous clinical cohorts. One potential caveat to this study is

the possibility for asymptomatic infection in the control cohorts, although these were entirely devoid of any clinical symptoms of malaria. Moreover, genetic analysis of human plasma samples for glucose 6 phosphate dehydrogenase (G6PD) deficiency and hemoglobinopathies was outside the scope of this study. This leaves a lack of information on certain key host responses during the regulation of host proteins in the response of malaria. These could be an effective future continuation of the present investigation, especially as the findings continue to move towards translational research. Collectively, our findings provided some novel mechanistic insights into malaria severity, and we anticipate that this will accelerate the opportunities for developing clinical tests integrating these host and parasite proteins for monitoring malaria severity and complexity.

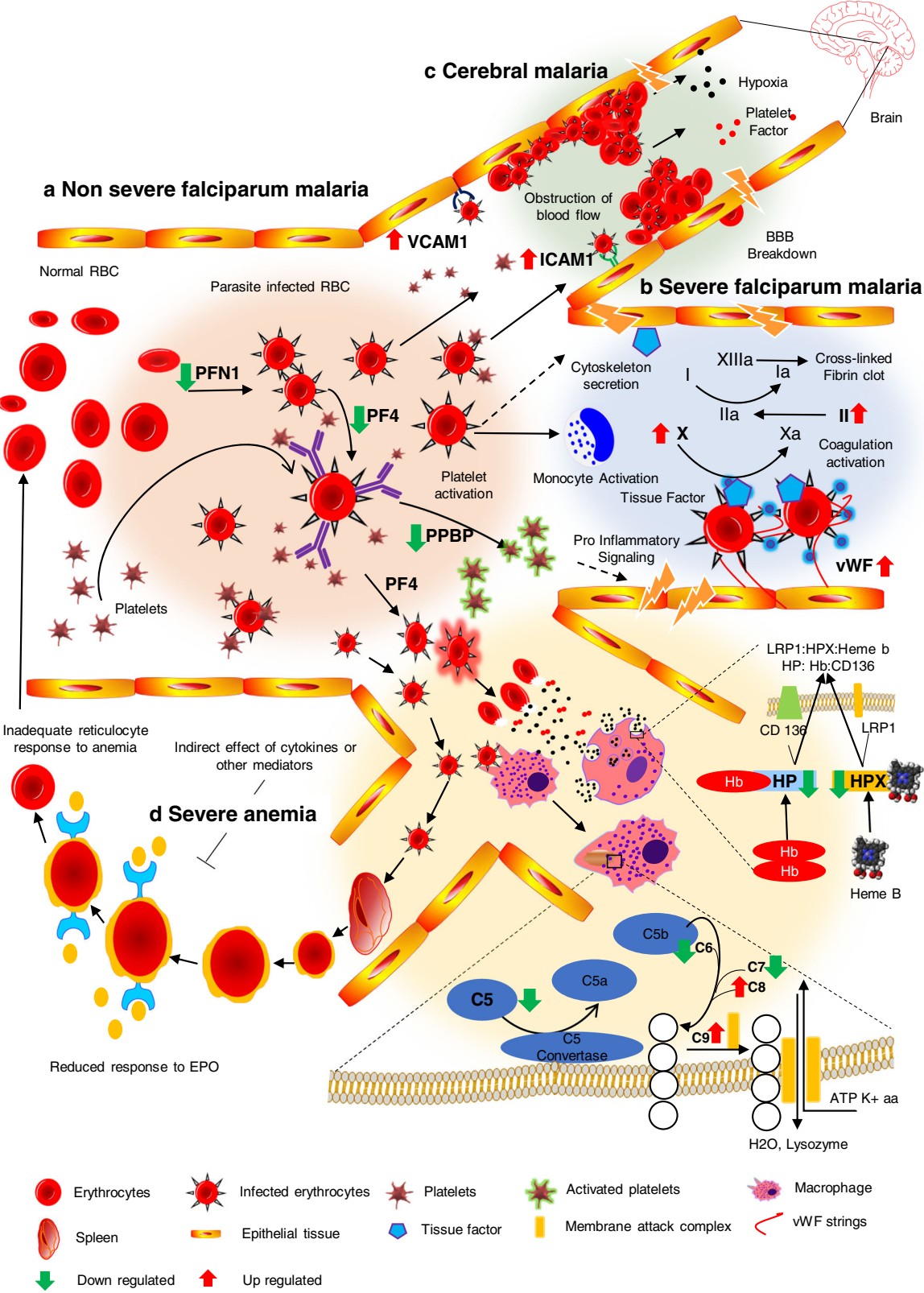

**Fig. 8 Possible molecular mechanisms driving the different complications of severe falciparum malaria.** a Disease progression from non-severe to different complications of severe falciparum malaria, the infected erythrocytes bind to platelets with aid of host proteins (PFN1) and leads to secretion of chemokines (PF4, PPBP) into blood circulation from platelets. The activated platelets start degranulation and release a plethora of cytokines and produce pro-inflammatory signals, which aggravate the inflammatory system, b once the coagulation system gets activated, prothrombin converted into thrombin and then it helps in secretion of cytokines, which enhance the severity of this parasitic infection. vWF strings attach to the infected erythrocytes, and together form a cluster. c Pro-inflammatory signals activate the macrophages, which engulf the infected erythrocytes and destroy those with the help of activated complement system. Infected erythrocytes get recognized by the spleen and are destroyed rapidly. Some infected erythrocytes burst in circulation and release diverse types of free radicals, which are subsequently scavenged by the host proteins such as HP and HPX. Cytokines and other mediators (hemozoin) slow down the process of erythropoiesis, which leads to an inadequate number of erythrocytes, and consequently, the patients suffer from an anemic condition. d The changes in and around the cerebral vessel, which causes the breakdown of the blood–brain barrier. The increased level of thrombin and production of cytokines via activated platelets leads to the expression of endothelial receptors such as ICAM-1 and VCAM-1. These receptors play a key role in cytoadherence of the parasitized erythrocytes on endothelial cell line and obstruct the blood flow. The release of toxins from the sequestered parasites leads to the recruitment of leukocytes and platelets. These mediators lead to endothelial cell activation, increase junctional permeability, and eventually the breakdown of the blood–brain barrier followed by secondary neuropathological events that can lead to cerebral edema or coma. The schematic diagram is structured using the information obtained from our quantitative plasma proteomics analyses.

## Methods

**Subject recruitment and blood collection.** This study was conducted involving malaria and DF patients, and control subjects from two different malaria-endemic regions of India. Subjects were enrolled in Calcutta medical college (CMC), Kolkata, India, and Sardar Patel medical college (SPMC), Bikaner, India. More precisely, this comprehensive proteomics study was accomplished involving HC subjects, and patients suffering from NSFM, SFM, and different complications (severe malaria anemia (SA) and CB and their corresponding control subjects, anemic [CSA], and meningitis patients [CCB], NSD; SD, NSVM, and SVM. This study was approved by the institutional review boards and ethics committee of the Indian Institute of Technology Bombay (IITB-IEC/2016/026). Prior to the sample collection process, written informed consent was received from each participant after giving detailed explanations about the experimental procedure in the language best understood by the potential participants.

The clinicopathological details of all the subjects enrolled in this study were thoroughly documented (Supplementary Table 4). The diagnosis of malaria-positive samples was carried out primarily by microscopic examination of thin peripheral blood smears followed by RDTs (FalciVax, Zephyr Biomedicals). The parasitemia count in individual samples was not reordered (in particular, due to the scarcity of time in handling the extreme burden of malaria patients in the tertiary care hospitals in a highly populated country like India). Additionally, samples were confirmed using PCR-based molecular diagnosis[66]. The brief workflow is depicted in Supplementary Materials and Methods, and schematic representation is provided in Supplementary Fig. 12 and Supplementary Tables 5 and 6. Patients with any other infectious diseases such as leptospirosis, chronic liver diseases, and mixed infections (infected with both *P. falciparum* and *P. vivax*) were excluded from this study. The case definitions of severe malaria[67] were adopted from standard WHO guidelines (Supplementary Table 7). The negative CSA (anemic patients without malaria) patients were defined by hemoglobin <7 g/dl and negative CCB (meningitis patients without malaria) were defined as clinically altered sensorium, neck rigidity, meningeal enhancement on MRI brain, and abnormal CSF study.

According to the recent WHO guideline[68], we classified the dengue patients into two categories—(1) NSD and (2) SD based on clinical features (blood pressure, bleeding manifestation, etc.) and hematological parameters (hematocrit, platelet counts, organ failure, etc.). In our study, IgM was used as the confirmatory marker for dengue infection. If NS1 was found to be positive, we retested for IgM during the 5–7 days of fever to confirm the diagnosis.

Blood samples were collected into ethylenediaminetetraacetic acid (EDTA), BD vacutainer, tube, and gently mixed by inverting 8–10 times. EDTA-anticoagulated blood (3 ml) was centrifuged at 1000g for 10 min, and plasma samples were stored at −80 °C until further processing[69].

**Species confirmation using nested PCR.** Blood samples of malaria-infected patients were collected from the different endemic regions of India. The collected samples were microscopy and RDT-positive samples. Further confirmation of the malaria parasite, nested PCR was performed using dried blood spots of the obtained samples. Only PCR-positive singly infected patient samples were taken forward for multi-omic study. For nested PCR from dried blood spots, Whatman filter paper 1 was taken, and 20 µl of RBC pellet was spotted on the Whatman paper. For each sample, 3–5 spots were made as per the availability of the RBC pellet. Two dried spots were punched in a 1.5 ml eppendorf tube. Added 200 µl of 0.5% freshly prepared saponin in PBS (or up to the volume required to submerge the punched spots). A short spin was given to settle down the droplets on the brim. Samples were incubated for 10 min at RT. Six hundred microliters of PBS was added to the blood spots for wash and supernatant was discarded. Washing was done for 2–3 times, and then the PCR reaction was set.

**Depletion of high-abundance proteins from plasma.** Removal of high abundant proteins is crucial for the detection of the low-abundant proteins, many of which frequently act as potential biomarkers. Therefore, before processing the plasma samples for proteomic analyses, the top 12 highly abundant proteins (albumin, IgG, Alpha1-acid glycoprotein, alpha1-antitrypsin, alpha2-macroglobulin, apoli-poprotein A-I, apolipoprotein A-II, fibrinogen, HP, IgA, IgM, and transferrin) were removed by using Pierce™ Top 12 abundant depletion column following the manufacturer's instructions (Thermo Scientific, Catalog no. 85164). Briefly, 15 µl of crude plasma was directly added to the resin slurry in the columns, and the mixture was incubated for 60 min at room temperature with gentle vortexing in between. The filtrate was collected in 2 ml Eppendorf tubes by centrifuging at 1000g for 2 min. The depleted plasma samples were concentrated using a vacuum centrifuge.

Protein extracted from the plasma samples of HCs and infected patients were quantified using Bradford Assay kit (Bio-Rad) following the manufacturer's instructions. The depleted plasma samples were loaded onto one-dimensional sodium dodecyl sulfate-polyacrylamide gel electrophoresis (SDS-PAGE) gels (12.5% acrylamide-bisacrylamide) for the quality check control (Supplementary Fig. 13a–c). The gels were stained with SDS-staining solution (methanol—40%, acetic acid—7%, D/W—53%, Coomassie blue—1 tablet) for 4 h and destained in SDS destaining solution (methanol—40%, acetic acid—7%, D/W—53%) for 5 h. The gels were scanned using a Image scanner III (GE Healthcare).

**Sample preparation for TMT-based quantitative proteomics analysis.** Prior to the enzymatic digestion, protein quantification was performed using Bradford assay (Bio-Rad) following the manufacturer's instructions. One hundred micrograms of depleted protein sample from each group (HC, CSA, CCB, SA, CB, SFM, and NSFM) were denatured using 15 µl of 6 M urea. The denatured proteins were reduced by adding tris (2-carboxyethyl) phosphine (TCEP) (Sigma Aldrich) to a final concentration of 20 mM and were incubated at 37 °C for 60 min. The reduced samples were alkylated by adding iodoacetamide [(IAA), Sigma Aldrich] to a final concentration of 40 mM and were incubated at room temperature in dark for 30 min. The samples were then diluted eight times with 50 mM ammonium bicarbonate buffer, and trypsin (Pierce Trypsin) was added to a 1:30 ratio (trypsin: protein) for performing in-solution digestion. Samples were incubated at 37 °C for 18 h for efficient digestion. The digestion was quenched by adding formic acid to a final concentration of 1%[70]. Subsequently, sample cleaning was performed using C18 Ziptip. Activation of the C18 ziptip was done by using 50% ACN in 0.1% FA, 40 µl, 1 min, 1500g thrice, followed by 99% ACN in 0.1% FA 40 µl, 1 min, 1500g thrice. Equilibration of ziptip by using 0.1% FA 40 µl, 1 min, 1500g was done thrice. Sample addition in ziptip up to 80 µl volume max, 1 min, 1000g five times. Clean-up using 0.1% FA 40 µl, 1 min, 1500g thrice. Elution by using 50% ACN in 0.1% FA 60 µl, 1 min, 1000g twice and 50% ACN in 0.1% FA 60 µl, 3 min, 4000g.

TMT six-plex isobaric tags (TMTsixplex™ Isobaric Label Reagent Set, 1 × 0.8 mg, Thermo Fisher Scientific, Catalog number: 9006) were used for FM and Dengue (FC) samples and TMT 10-plex isobaric tags (TMT 10plex™ Isobaric Label Reagent Set, 1 × 0.8 mg, Thermo Fisher Scientific, Catalog number: 90110) were used for VM samples for labeling of the digested peptides as per the manufacturer's instructions. In brief, digested peptides were reconstituted in dissolution buffer and were vortexed to mix well. A pooled sample was prepared from each individual patient sample and was considered as a reference for performing normalization of the different mass spectrometric datasets. TMT reagents were reconstituted in 40 µl of anhydrous acetonitrile (ACN) and the reagents were added to the corresponding aliquot of digested plasma protein sample following the labeling strategy (Supplementary Fig. 13d–f); 15 µg of each digested peptide sample was labeled at ~1:13 ratio (digested peptides: TMT reagents) for performing efficient labeling. The solution was mixed well and incubated at room temperature (RT) for 1 h. The reactions were quenched by adding 2 µl of 5% hydroxylamine and were incubated for 15 min at RT. All the respective samples were pooled and were dried completely

                                              COMMUNICATIONS BIOLOGY | https://doi.org/10.1038/s42003-020-01384-4

in a vacuum centrifuge. Samples were then fractionated into nine fractions using high-pH reverse-phase chromatography following the manufacturer's instructions (Pierce™ High pH Reversed-Phase Peptide Fractionation Kit, Thermo Fisher Scientific, Catalog number: 84868).

**Liquid chromatography–mass spectrometry/mass spectrometry (LC-MS/MS) analysis**. Multiplexed TMT-labeled samples (6/10-plex) were analyzed as biological replicates using an Orbitrap Fusion Tribrid mass spectrometer interfaced with an Easy-nLC 1200 system (Thermo Fisher Scientific). The mobile phase consisted of milli-Q water with 0.1% formic acid as solvent A and 0.1% formic acid/80% acetonitrile as solvent B. Each fraction was reconstituted in 15 μl of solvent A and 1 μg of digested peptides were loaded on to a pre-analytical column (100 μm × 2 cm, nanoViper C18, 5 μm, 100 A; Thermo Fisher Scientific). Isocratic gradient of 10–35% B in 103 min, 35–95% B in 2 min and holds at 95% B for 15 min at 300 nl/min flow rate were used on an analytical column (75 μm × 50 cm, 3 μm particle, and 100 Å pore size; Thermo Fisher Scientific). A single Orbitrap MS scan from 375 to 1700 $m/z$ at a resolution of 60,000 with automatic gain control (AGC) set at 5e (ref. [4]) was followed by up to 20 ms/ms scans at a resolution of 30,000 with AGC set at 4e (ref. [5]). MS/MS spectra were collected with a normalized collision energy of 35% and an isolation width of 1.2 $m/z$. Dynamic exclusion was set to 40 s, and the peptide match was set to on. Surveys scans were performed in the Orbitrap mass analyzer and data-dependent MS2 scans were performed in Orbitrap mass analyzer trap using higher-energy collisional dissociation (HCD) following isolation with the instrument's quadrupole. The intensity threshold of the peptide was set 5e (ref. [3]). Internal calibration was carried out using a lock mass option ($m/z$ 445.1200025) from ambient air. The same parameters were used for LFQ ($n = 21$), except collision energy, which was set 30%.

**Database search for peptide and protein identification**. TMT 6-plex- and TMT 10-plex-based quantitative proteomic analysis were carried out using individual samples of malaria and dengue patients. Raw instrument files were processed using Proteome Discoverer (PD) version 2.2 (Thermo Fisher Scientific). In each TMT experiment, .raw files for all fractions were merged and MS2 spectra were searched using the Sequest HT and Mascot (v2.6.0) search engine against *Homo sapiens* fasta (71,523 sequence entries, dated: 24/06/2018) from Uniprot database (Proteome ID: UP000005640, Organism ID: 9606). All searches were configured with dynamic modifications for the TMT reagents (+229.163 Da) on lysine and N-termini, oxidation of methionine residues (+15.9949 Da) and static modification as carbamidomethyl (+57.021 Da) on cysteines, monoisotopic masses, and trypsin cleavage (max 2 missed cleavages). The peptide precursor mass tolerance was 10 ppm, and MS/MS tolerance was 0.05 Da. The false discovery rate (FDR) for proteins and peptides was kept 1%[71]. TMT signals were corrected for isotope impurities based on the manufacturer's instructions.

In LFQ, the .raw files from the label-free method were searched against the database of *Plasmodium falciparum 3D7* (Plasmo TaxID=36329_and_subtaxonomies) (v2017-08-25). The parasite data were downloaded from PlasmoDB (https://plasmodb.org/plasmo/) on 28/06/2018. The search parameters were kept the same as above mentioned for the TMT 6-plex method except for dynamic modifications for the TMT reagents (+229.163 Da) on lysine and N-termini of the peptide.

**Machine learning and feature selection**. The PSMs values were processed using MSstatsTMT[72,73], where run-to-run normalization using a reference pool and quantile normalization were performed. Sixty-six proteins were selected based on adjusted $p$ value ($p < 0.05$) in malaria and $p$ value ($p < 0.05$) in DF (Supplementary Data 7). Since missing values are associated with the proteins with low levels of expression, we imputed the missing values by drawing samples from a normal distribution with a mean that is down-shifted from the sample mean and a standard deviation that is a fraction of the standard deviation of the sample distribution.

The elastic net regularized logistic regression method was used to classify: (1) dengue vs. malaria: HC vs. Dengue, HC vs. malaria, dengue vs. malaria; (2) type of malaria: HC vs. FM, HC vs. VM, FM vs. VM; (3) severity of FM: HC vs. NSFM, HC vs. SFM, NSFM vs. SFM; (4) severity of VM: HC vs. NSVM, HC vs. SVM, NSVM vs. SVM; (5) severity of dengue: HC vs. NSD, HC vs. SD, NSD vs. SD. Of note, the elastic net regularized logistic regression model offers more flexibility in two ways —(i) the $l_1$ norm helps attain parsimony in the sense that it optimally chooses the number of covariates (in the context of present dataset the covariates are proteins) by driving coefficients of unimportant covariates to zero and (ii) $l_2$ norm helps address the issue of multicollinearity[74–82]. $k$-fold nested cross-validation ($k = 10$ for malaria vs. dengue, $k = 10$ for falciparum vs. vivax, and $k = 5$ for cerebral vs. severe malaria anemia) was used as it provides robust and almost unbiased parameter estimates and model performance evaluation even for small sample sizes[83,84].

The nested cross-validation approach involves using—(i) $k$-fold inner cross-validation loop for hyperparameter tuning and model selection and (ii) $k$-fold outer cross-validation loop for evaluating the model selected by the inner cross-validation. The entire dataset was randomly split into $k$-folds, out of which the $k$th fold was used as a test set and the remaining $k-1$ folds were used for training

purpose. For each split, a model was trained and validated on the training set using (inner) $k$-fold cross-validation and tested on the held-out test set. The hyperparameters alpha and lambda were tuned by inner $k$-fold cross-validation over a grid of values ranging from 0 to 1 with a step-size of 0.1 for alpha and another grid of 100 values ranging from $10^{-2}$ to $10^2$ for lambda, respectively. To evaluate model performance, ROC–AUC, balanced accuracy, $F1$-score, and Kappa metrics were computed. These metrics were chosen because we had imbalanced classes. The average variable importance of a protein was measured as the weighted average of variable importance of that protein in all $k$ predictive models in the outer $k$-fold cross-validation with weights being the balanced accuracy of each model. This implicates that the importance of a protein in the classification of samples is directly proportional to its occurrence in the predictive models, which can classify the samples better. A schematic diagram of the method used here is shown in Figs. 1d and 5a. The proteins with non-zero variable importance from an elastic net regularized logistic regression model were considered for further analysis.

For validation of the selected proteins as good classifiers for dengue vs. malaria, FM vs. VM, and CB vs. SA models, we created (i) heat map and (ii) three-dimensional PLS-DA plot (Fig. 5b–f and Supplementary Data 5). To arrive at the best panel of biomarkers, six different biomarker models were created from different numbers of proteins selected out of all the important covariates identified in the machine learning model. The ROC curves for these biomarker models shown in Supplementary Fig. 9b–d enabled comparison of different models based on their corresponding AUC values and confidence intervals. We have provided a detailed discussion on the justification and performance of the net regularized logistic regression model under the Supplementary Information (Supplementary Methods and Supplementary Data 5).

The elastic net regularized logistic regression model equations for the final biomarker models identified from MRM were as follows:

dengue vs. malaria:

$$\log_e\left(\frac{p_{\text{dengue}}}{1 - p_{\text{dengue}}}\right) = -0.4269 + 2.23 \times \text{LRG1} - 1.7374 \times \text{CP}. \quad (1)$$

Falciparum vs. vivax malaria:

$$\log_b\left(\frac{p_{\text{VM}}}{1 - p_{\text{VM}}}\right) = 0.0893 + 0.169 \times \text{AZGP1} + 0.1425 \times \text{HRG}. \quad (2)$$

Cerebral vs. severe malaria anemia:

$$\log_b\left(\frac{p_{\text{CB}}}{1 - p_{\text{CB}}}\right) = -0.2118 - 0.2365 \times \text{SERPINA3} + 0.4309 \times \text{AHSG}, \quad (3)$$

where $p_{\text{Positive Class}}$ is the probablility of event that $Y$=Postitive Class. For these three models, the hyperparameters (alpha and lambda) and model performance evaluation metrics (balanced accuracy, $F1$-score, Kappa, ROC–AUC, ROC–AUC confidence interval) were as follows: (i) dengue vs. malaria: 0.22, 0.02, 0.72, 0.89, 0.46, 0.807 (0.666–0.947); (ii) FM vs. VM: 0.25, 4.47, 0.8, 0.77, 0.6, 0.865 (0.772–0.958); and (iii) CB vs. SA: 0.02, 41.10, 0.8, 0.83, 0.56, 0.944 (0.816–1), respectively. These models serve two purposes, and they are as follows:

They help assess the impact of a particular protein on the estimated log odds or the estimated odds of an individual, for example, being a dengue patient as opposed to being a malaria patient.

For illustration sake, let us consider the model Eq. (1). Note that this model equation yields:

$$\frac{\hat{p}_{\text{dengue}}}{1 - \hat{p}_{\text{dengue}}} = \exp(-0.4269 + 2.23 \times \text{LRG1} - 1.7374 \times \text{CP})$$
$$= \exp(-0.4269) \times \exp(2.23)^{\text{LRG1}} \times \exp(-1.7374)^{\text{CP}}$$
$$= 0.6525 \times 9.2999^{\text{LRG1}} \times 0.176^{\text{CP}}.$$

This can be interpreted by observing that the estimated odds of an individual being a dengue patient as opposed to s(he) being a malaria patient increase multiplicatively (decrease multiplicatively) by 9.2999 (0.176) for every unit increase in the value of the protein LRG1 (CP).

Note that the equation involving the estimated odds $\frac{\hat{p}_{\text{dengue}}}{1-\hat{p}_{\text{dengue}}}$ given in I further leads to

$$\hat{p}_{\text{dengue}} = \frac{\exp(-0.4269 + 2.23 \times \text{LRG1} - 1.7374 \times \text{CP})}{1 + \exp(-0.4269 + 2.23 \times \text{LRG1} - 1.7374 \times \text{CP})}$$
$$= \frac{0.6525 \times 9.2999^{\text{LRG1}} * 0.176^{\text{CP}}}{1 + 0.6525 \times 9.2999^{\text{LRG1}} \times 0.176^{\text{CP}}}.$$

and this, in turn, enables one to classify a new patient as a dengue or a malaria patient depending upon whether $\hat{p}_{\text{dengue}}$ is ≥0.5 or not for specified values of LRG1 and CP for the patient under consideration.

The other two logistic regression models corresponding to falciparum vs. vivax malaria or cerebral vs. severe malaria anemia could be interpreted analogously.

**Interaction network and bioinformatics analysis**. In order to investigate the complex interactions among the candidate marker proteins and for prediction of the pathways associated with the differentially altered proteins identified in FM, VM, and DF patients, diverse bioinformatic analyses were carried out. The pathway

enrichment was performed using Protein Analysis THrough Evolutionary Relationships (PANTHER) classification system, version 12.0 (www. pantherdb.org)[85], and Reactome pathway Knowledgebase, version 62 (www.reactome.org)[86]. A multi-functional online software NetworkAnalyst (http://www.networkanalyst.ca/) was applied for constructing and visualizing the PPI networks. The batch selection option of NetworkAnalyst was used to narrow down the network nodes, and the clusters were specified by highlighting with different colors.

**Validation using MRM**. Plasma proteins showing a prominent differential abundance in malaria patients were further validated using MRM assays on a triple quadrupole mass spectrometer Altis (Thermo Fisher Scientific) equipped with an Easy-Spray electrospray ionization ion source (Thermo Fisher Scientific). Peptides separations were performed using a C18 column (Hypersil GOLD, 150 mm, 2.1 mm, 1.9 μm, Thermo Fisher Scientific). The mobile phase consists of Milli-Q water with 0.1% formic acid as solvent A and 0.1% formic acid/80% acetonitrile as solvent B. Following chromatographic conditions were used: 20-min gradient at a flow rate of 300 nl/min starting with 100% A (water), stepping up to 2% B (ACN) in 0 min, followed by 45% B at 16 min, followed by a steep increase to 95% B at 17 min and static for 1 min at 95%. The steep decrease to equilibrate the column 5% of B at 19 and then static for another 1 min at 5% of B. The suitability of the system was evaluated using iRT peptide and digested BSA (Supplementary Fig. 10a). To assess the suitability of the selected peptides for MRM, 4.8 nmol/peptide ($n = 17$) of each heavy synthetic peptide was spiked into 10 μg of the digested samples, with 142 peptides having 901 transitions over a 20-min chromatographic gradient with a scheduling method. The total 40 methods were used to accommodate all the transitions (approx. 176 transition/method) for host and parasite proteins. During the first phase of optimization, six proteins were not detected and hence removed those from the list. The refinement of the transitions performed by running different plasma pool samples and some individual samples on different days to check the inter-day variation and stability of the peptides (Supplementary Fig. 10b). The 46 proteins finalized, which has good dotp value for the peptides (dotp > 0.85) (Supplementary Fig. 10c). The scheduling of peptides was performed, and the run was carried out as a single method to quantify the proteins from FM, VM, dengue, and HC samples. Targeted acquisition of the eluting ions was performed by the mass spectrometer operated in SRM-MS mode with Q1 and Q3 set to 0.7 m/z full-width at half-maximum resolution and a cycle time of 2 s. A single scheduled method was utilized with a 2-min elution window for all MRM-MS runs. Data analysis was performed using Skyline daily[87].

**Statistics and reproducibility**. The proteins that were quantified with ≥1 unique peptide and detected in at least 60% of the samples were selected for the subsequent differential analysis. The PSMs values were exported from Proteome Discoverer 2.2 (PD) for combined analysis for FM, VM, and DF (FM: $n = 10$ experimental sets (TMT 6-plex), 90 fractions; VM: $n = 4$ experimental sets (TMT 10-plex), 25 fractions, and DF: $n = 4$ experimental sets (TMT 6-plex), 24 fractions). The protein-level summarization and significance analysis of the proteins was performed in R using the MSstatsTMT package[72,73]. In brief, The PSMs were preprocessed in PD and were converted into the required input format for MSstatsTMT using an in-house R-code. The protein abundance was calculated based on the peptide quantification using the "protein summarization" script (MSstatsTMT). It includes normalization between the MS runs using reference pool channels and the imputation of missing values before summarizing peptide level data into protein-level data. In the protein summarization method, MSstats assumes missing values are censored and then imputes the missing values using the accelerated failure model. Missing value imputation was performed for only of those proteins, which were quantified in ≥60% of the samples. Then, a moderated t-test was performed on quantile normalized values using "Group-Comparison-TMT" script in R and p value <0.05 were considered as statistically significant[72,73]. Proteins passing p value <0.05 threshold of moderated t-test was used for data visualization and pathway analysis. However, adjusted p value Benjamini–Hochberg (BH) was also calculated for multiple comparisons using the MSstatsTMT package in R. BH-corrected p value [p value (adjusted) <0.05] was considered in the selection of proteins for machine learning analysis (except for dengue).

**Reporting summary**. Further information on research design is available in the Nature Research Reporting Summary linked to this article.

## Data availability
All processed data associated with this study are present in the manuscript or in the Supplementary Materials. Raw MS data and search output files for TMT-based quantitative proteomics analyses described in this article are deposited to the ProteomeXchange Consortium via the PRIDE[88] partner repository with the dataset identifier PXD014991. Targeted proteomic data are deposited in the Peptide Atlas[89] and can be accessed through Dataset Identifier: PASS01467. All raw and processed data are made available as Supplementary Datasets 1–7.

## Code availability
The custom code for MS data analysis (TMT 6/10plex) and machine learning have been deposited at https://github.com/vipin786/R-code-for-MS-data as well as publicly available via Zenodo (https://doi.org/10.5281/zenodo.4022347)[90].

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

## Acknowledgements

The active support from Suman Ghosh from the Department of Medicine, Medical College Hospital Kolkata, and Dharmendra Rojh from the Department of Medicine, Malaria Research Center, S.P. Medical College, Bikaner in the clinical sample collection process is gratefully acknowledged. We are also grateful to Sandip K. Patel, Saicharan Ghantasala, Nikita Gahoi, and Apoorva Venkatesh from the Department of Biosciences and Bioengineering, Indian Institute of Technology Bombay for their insights and suggestions regarding the quantitative proteomics experiments. We also acknowledge Ting Huang from computational biology and proteomics at Northeastern University, Rohan Agarwal, and Saurabh Rajguru from the Chemical engineering department, Indian Institute of Technology Bombay for their kind support in data analysis using MSstatsTMT. We acknowledge the MASSFIITB Facility at IIT Bombay supported by the Department of Biotechnology (BT/PR13114/INF/22/206/2015) to carry out all MS-related experiments. This work was supported by the Department of Biotechnology, India grants No. BT/PR12174/MED/29/888/2014, BT/INF/22/SP23026/2017 and Ministry of Human Resource Development, Government of India (MHRD-UAY Phase-II Project (IITB_001) to S.S. V.K. and S.A. were supported by the IIT Bombay fellowship.

## Author contributions

V.K., S.S., S.P., and S.R. conceived and designed the experiments. V.K., M.J., and S.A. performed the MS-based quantitative proteomics experiments and data were analyzed by V.K., S.A., S.R., R.Y., and S.V.S. Bioinformatics analysis were performed by V.K., S.A., and D.B. Clinical samples and clinicopathological details were collected by S.B., A.T., and S.K. S.S. supervised the entire study and secured funding. The manuscript was written by V.K., S.R., and S.S. with input from all authors. All authors agreed on the interpretation of data and approved the final version of the manuscript.

## Competing interests

The authors declare no competing interests.
