## [Peer Review File · Communications Biology]

Reviewers' comments:

Reviewer #1 (Remarks to the Author):

Overall, Kumar et al, provides a comprehensive host based assessment of malaria infection, with appropriate due diligence towards comparison of controls, data processing, analysis, validation, and interpretation. At this point I have the following minor comments.

- 1) What exactly is the criteria determining severe versus mild malaria cases? If you included it I missed it. Please make this clear..
- 2) please define the acronyms of the patient populations early on, line 107?
- 3) A patient demographic table would be appropriate.
- 4) I also did not see any reference to proteomic repository data download, or inclusion of proteomic data that fits the criteria for publishing such results. Can you please include the appropriate peptide and protein level data as well as deposit it in an appropriate repository. Thank You

Reviewer #2 (Remarks to the Author):

The present manuscript uses a proteomic platform investigating plasma from patients with malaria. It is one of the most comprehensive so far as it includes comparisons between malaria severity states (already described in the literature) but also compares the two most prevalent species of Plasmodium (falciparum and vivax). It also includes disease controls, mainly Dengue fever. The present study also describes the detection of parasite proteins within plasma samples. The data analysis of the complex proteomic data is step-wise and led to description of potential pathways of pathogenesis for the different disease groups. Potential biomarkers (of human or parasite origin) were validated by MRM, which is crucial for a study with such a small cohort. The study is well designed and conducted. It is however not the first proteomic study performed, even if it is the first time using an untargeted approach utilizing TMT-based labelling, more reproducible than label-free. According to the title, one may expected a more advanced technology than the one used, which is now well-established in the proteomic community and on the verge of being outdated to more throughput methodologies (eg.SWATH)

One major weakness of the study is the cohort size. Even though it includes 143 total subjects, the division in the numerous disease groups limits the statistical power, with some groups under 5 patients. This has consequences in the design of the study. No discovery-validation cohort split up which would have been necessary to validate the regression model built for malaria prediction. The validation using MRM carried out is essential but a method validation. Cohort validation remains essential. Unless the MRM was done in a different cohort, which was not made clear.

Moreover, as a complex study with stepwise comparisons, it was made more confusing by the many different ways of analysing and representing the data (see details in comments below) and the inconsistencies from one comparison to the other. The consequence was that the main result messages are buried and not clear.

Related to the previous comment, the conclusion is not clear. The present manuscript does indeed 'provide a comprehensive landscape of plasma proteome alterations in different levels of malaria severity'. However, the novel pathways discovered are not obvious as most of them corroborates the pathogenetic pathway hypothesis currently researched.

Finally the tone to which the manuscript is written is open to over interpretation. Specific examples are in the comments below.

I have many comments of varying significance. They are listed below following the paper's chronological order.

Comments:

Title/Abstract:

The vocabulary employed to describe the overall study include "first comprehensive attempts by deep proteome investigations..." and use of "multiplex highly sensitive quantitative proteomics". It is not the first comprehensive attempt (Bachman et al 2014), although this was using a targeted proteomic approach. 'Deep proteome' means including 1,000s of proteins as most of the studies now reach 4,000-10,000 even in semi-quantitative studies using 2-dimensional LC-MS/MS. Although, in fairness, plasma short linear range makes it more difficult.

Results:

Analysis of clinicopathological parameters of malaria and dengue patients

The first time the authors use abbreviations for the disease groups, could they write them fully (line 107 p.6).

Line 108 p.6: This is probably due to my ignorance but what do the authors mean by 'multi-centric' study? Do they mean multi-centre?

Line 111 p.6: 'sequential decrease'. There is no sequence as there is no repeated samples over time of same patient.

Workflow for comprehensive plasma proteomic analysis...

Supplementary Figure 4: The authors have chosen to show in supplementary material the QC graphs, however they have not explained or commented for the non-expert readers what these graphs means. I am not suggesting writing a new paragraph but just a description in the figure caption. From these graphs, one might conclude is that there is high variance in samples in the falciparum comparisons with lower reproducibility than in other comparisons. An analytical comment in the caption would be welcome.

Figure 1: Panel b shows 'label-free' and TMT-labelled samples but there is no mention in the text and this may be confusing. It only becomes clear when reading the parasite identification paragraph. Because the authors used a paragraph to describe the workflow and draw a very detailed and helpful diagram, I think the text should be more precise and parallel to what the diagram represent to avoid confusion.

The data missing is the number of cases and demographics for each comparison cohort.

Differential plasma proteomic analyses sections

Line 148 p.7: '296 proteins were present in more than 60%'. It is unclear how the data was filtered and processed. Was the resultant data analysis was based on the 296 or how the missing value was dealt with? The information in the materials and methods warrants more details.

What was the test used to choose the significantly altered proteins and more particularly was there a correction applied for multiple comparisons and which one? With 1495 supposedly included in the comparison, a multiple comparison test is necessary. Also, what were the other criteria for positive identity (eg. No. peptide>2?)

I have concerns about the amount of data with missing value and how much missing value imputation was performed for this data. Could you confirm what the filter cut-off to exclude a protein was and how many were in the final comparison and have this number in the figure 2a diagram. The only reference to imputation is in the materials and methods for the Machine learning section and it is not very detailed.

A statistical analysis section in the materials and methods section might resolve these issues.

Fig.2b: In the non-severe to severe falciparum comparison, why not comparing these two states directly, particularly for the volcano plot (Figure 2b). Directly comparing severe to non-severe cases might bring out the specific markers to severe disease. Same can be said about the CB vs SA comparison and SVM vs NSVM (Fig.3a and 4a)

The volcano plots are obsolete to the manuscript, particularly as top proteins are not shown. Similarly, how well do cluster NSFMvsSFM, SVMvsNSVM, CBvsSA and DFvsSD in Suppl.Fig 5, 7 and 9

Heatmaps: What criteria were selected to include proteins for the supervised heatmap?

PCA plots: In Figure 2 the PCA plot is described as being 'performed on all identified proteins' (line 153 p.7), it is unclear whether it is indeed based on the 1,495 identified proteins.. In Figure 3 it is described as performed on 'differentially abundant proteins' (line 196 p.9) and 'Common and specific significantly altered proteins' (Figure 3c caption), which is clear. Is there an inconsistency?

8 out of the 12 depleted proteins are in the list of proteins altered. Is the fact that they are present in relative abundance suggesting the depleting columns were saturated? It is difficult to assess as multi-protein depletion was carried out. Nevertheless, should these be taken in account or changes validated using total plasma (or albumin depleted plasma)?

Line 158p.8, TF is referred as Tissue factor. Line249 p.11, it is referred as Serrotransferin.

There are many ways to study pathway or network analysis and some will complement each other and will give different views on the data. I believe it is a bit confusing to have the analysis of the clusters (line 156-167 p.8) and the network analysis (which I understand to be drawn from NetworkAnalyst?)(Lines 169-179 p.8). I do not suggest to remove the heatmap or the cluster quadrants but in the text choosing which list of pathway and proteins to describe (network analysis) would make it less confusing, particularly as they overlap greatly.

With the absence of a cohort Table with demographics, it is difficult to know why the 'vivax' comparison only has 4 controls, whilst the 'falciparum' comparison has 10. Was it from a different cohort altogether? Why are the controls not mixed? Is there any demographic differences between the 2 set. It might be more robust to directly compare NSVM to SVM without the HC.

The comparison of non-severe versus severe dengue fever seems out of topic. It should be directly compared to the malaria cases alongside heatmap figure 2d and 4e. The CSA and CCB would be better appreciated in heatmap Figure 3f directly compared to SA and CB, respectively.

Identification of Plasmodium falciparum parasite proteins

Any differences of detection in severe versus non severe falciparum cases?

Lines 267-274 p.12: the paragraph is out of sequence

Machine learning model:

It is difficult to assess appropriateness of the model and more details (statistical) would be appreciative to the lay reader. How do the signatures compare to the protein lists found in the previous comparisons. How did it perform overall. What were the training and test sets composition and size? The resulting ROC curves show that the models are not much improved over 10 protein sets and are quite robust with low amounts of proteins.

Validation:

Could the authors be clear on whether the samples used for validation were the same as in the total proteome analyses?

Could the authors add a table with inferred changes in the 86 protein levels from validation alongside changes observed using discovery total proteome data for comparison (Supplementary). Or the number of protein showing the same trend between the two methodologies.

In most of the manuscript, proteins are described by their symbol but in Fig. 6b,c,d, they are described by uniprot ID. Would it be possible to change it to protein symbol for ease?

Discussion:

Lines 344-346 p.16: So far the most comprehensive studies are from targeted approaches (Bachman et al 2014 and Reuterward et al 2018) Although the present study uses a non targeted approach, the amount of data is not more comprehensive and pathways observed are similar to those previously mentioned (including previous studies from the authors). Moreover, the linear regression model does not confer causality only associations and only a study containing successive samples from same patient would confer progression. Finally there is a contention as to whether there is a progression from non-severe to severe, particularly in CM when cases are so acute.

Similarly, in line 427, the statement 'we also observed that dysregulation of hemostasis-related proteins causes severe anemia' is erroneous. Again, there was no causality analysis only association.

It is difficult to extract novel pathways discovered as most have been previously reported (as referenced in the discussion). The discussion would benefit from highlighting what are the novel pathways and proteins in a more pronounced and specific way.

There was no discussion about the parasite proteins detected, how they can be used for future research and how they compare to previous studies (eg. Moussa et al 2018)

Finally, the last statement in line 433 p.19, 'opportunities for developing clinical tests', it would be beneficial to add the specific proteins targetted in such tests (refer to previous comment about machine learning model section).

Reviewer #3 (Remarks to the Author):

The manuscript by Kumar et al. provides a detailed quantitative proteomics study in plasma of malaria patients with different complications, and also compares malaria caused by *Plasmodium falciparum* and *P. vivax* among them. Of particular value is the choice of controls for complications or disease that can be confused with clinical malaria symptoms, such as Dengue virus infection and meningitis.

The manuscript can be considered as providing a wealth of new information on the response to malaria in humans, and does so with a complex and technologically advanced experimental approach.

Nevertheless there are several points that may require the attention of the authors to provide a more smooth reading for the interested readers.

Major:

- 1) In the results sections where experimental groups analysed are compared not always is easy to follow the rationale behind the research. In particular, the groups, abbreviations and number of samples defined in several places (e.g. lines 107-108; lines 440-444, and along the results section) are not always coincidental. There is also some disagreements between the numbers of samples for a given group in the text and in the Figures (e.g. in the text is mentioned the existence of 58 samples in the HC group but apparently only 10 are shown in Figures 3d or 4 in Figure 4). Thus I suggest to include a table with 3 columns indicating each clinical group defined, the abbreviation used and the number of samples at each group. This will make easy to follow also several of the main Figures.
- 2) Can the authors provide information about parasitemia levels and results of RTD in parallel in all samples? How can the authors be sure that healthy controls (or other disease controls used) did not have subclinical plasmodium parasitemia if RTD systems do not test positive below 100 parasites per ul? PCR for *P. falciparum* and *P. vivax* was performed in all control samples? What it's the sensitivity of their RDTs?
- 3) Line 84: "dark matter" is used as a metaphor here but it would be convenient to define exactly what the authors wish to mean with it.
- 4) Line 120: what is the meaning of "subclinical" in this line?
- 5) Diagnosis of Dengue should be clarified since it can be distinguished two different clinical stages and IgG / IgM markers for them. Is SD haemorrhagic fever?
- 6) Why elastic net regularized logistic regression method was chosen?
- 7) As the manuscript mostly describes the plasma effect of the host response to malaria, there are human genetic polymorphisms of malaria tolerance that could be response modifiers. Are there data on polymorphisms in haemoglobin or G6PD in the samples used?
- 8) Line 149: is any of the 31 plasma proteins altered in NSFM in coincidence with the 38 found in SFM? Please detail findings to this respect.
- 9) Line 161: These two controls are not clearly defined in terms of clinical and diagnostic characterization. How many of each?
- 10) Supplementary Figure 5: what are the common proteins within the SFM group between CB and SA subgroups? If any, this should be mentioned.
- 11) Do the authors consider that changes in protein expression in the range +1 to -0.5 are sufficiently appropriate to draw mechanistic conclusions?
- 12) Line 254. How many of the proteins found significantly as NSVM markers also appear in NSFM? This would be interesting in case they have similar response pathways.
- 13) Lines 267-274: how consistent were the presence of this proteins in all samples? Can they be

used for developing new diagnostic procedures?

14) Several of the up-regulated proteins found in severe malaria (cerebral malaria) have been previously found upregulated in murine models of malaria thus reinforcing the authors' findings. Providing these previous findings as reinforcement of the authors' own findings would give more strength to their discussion and to the mechanistic meaning of the markers discovered.

Minor:

Although there are easily distinguishable abbreviations, they are not always defined in the text. It is suggested that all of them are defined (e.g. at line 121).

Multiplexed Quantitative Proteomics Provides Mechanistic Cues for Malaria Severity and Complexity

Response to the Comments of Reviewer # 1

Overall, Kumar et al, provides a comprehensive host-based assessment of malaria infection, with appropriate due diligence towards comparison of controls, data processing, analysis, validation, and interpretation. At this point I have the following minor comments.

This reviewer considers that our manuscript “provides a comprehensive host-based assessment of malaria infection, with appropriate due diligence towards comparison of controls, data processing, analysis, validation, and interpretation”. We are grateful to this reviewer for an insightful review of our article. At the outset, we would also like to thank the reviewer for the positive feedbacks and valuable suggestions regarding our study, which have aided in improvement of the manuscript. Detail responses to the comments and queries from the reviewer are appended below.

Comment 1: What exactly is the criteria determining severe versus mild malaria cases? If you included it I missed it. Please make this clear.

Response: We appreciate this advice from the reviewer, and we wish to clarify that the criteria for severe and non-severe malaria were adopted directly from the WHO guidelines.

Revision made: According to the suggestion of this reviewer, we have clarified this point in the revised manuscript and provided the associated references. Moreover, we have included an additional supplementary table to describe the case definition for severe malaria as per the WHO guidelines (WHO 2000).

Changes has been incorporated on following Page no.: 20-21

Supplementary Table 12: WHO case definition of severe malaria. The infection is considered as severe if the patients exhibit one or more of these following conditions.

S.no.	Complication	Condition criteria for adults
1.	Impaired consciousness	A Glasgow Coma Score <11
2.	Acidosis Clinical manifestation respiratory distress	>8 meq/l Base deficit or, <15 mM bicarbonate in plasma or >5 mM plasma lactate.
3.	Hypoglycemia	Glucose level in blood or plasma <2.2 mM (<40 mg/dl)
4.	Severe malarial anemia	<7 g/dl of haemoglobin concentration or <20% of hematocrit together with a parasite count >10 000/ μ l

5.	Renal impairment	>265 µM (3 mg/dl) of creatinine in Plasma or serum or >20 mM urea in blood
6.	Jaundice	>50 µM (3 mg/dl) of bilirubin in plasma or serum with >100 000/µl parasite count
7.	Pulmonary edema	Radiologically confirmed, or >30/min respiratory rate with <92% oxygen saturation in air
8.	Significant bleeding	Recurrent or prolonged bleeding from nose gums or venipuncture sites; hematemesis or melena
9.	Compensated Shock	≥3 s capillary refill or temperature gradient on leg (mid to proximal limb), but no hypotension. Decompensated shock is defined as systolic blood pressure <70 mm Hg in children or <80 mm Hg in adults with evidence of impaired perfusion (cool peripheries or prolonged capillary refill)
10.	Hyper-parasitemia*	>10% P. falciparum parasitemia

* Hyper-parasitemia is not applicable in defining *P. vivax* severity as the pathogen can cause severe infections even at a low level of parasitemia.

Comment 2: Please define the acronyms of the patient populations early on, line 107?

Response: We would like to thank the reviewer for this suggestion regarding the usage of abbreviations in our manuscript.

Revision made: In the revised manuscript, acronym expansions are provided during the first usage of the abbreviations.

Following changes are incorporated in the text:

We have incorporated the changes in the manuscript. “A total of 106 subjects were analyzed in the discovery phase quantitative proteomics, while 111 subjects were included in the targeted validation study. 143 subjects were included for analysis of the clinical parameters. The following cohorts were included in this study – Healthy control (HC), Non-severe falciparum malaria (NSFM), Severe falciparum malaria (SFM), Cerebral malaria (CB), Severe anemia (SA), Control for severe anemia (CSA), Control for cerebral malaria (CCB), Non-severe dengue fever (NSD), Severe dengue (SD), Non-severe vivax malaria (NSVM) and Severe vivax malaria (SVM). The subjects were recruited from different malaria epidemic regions in India and the number of samples analyzed for each group has been demonstrated in Supplementary Table 1.”

Changes has been incorporated on following Page no.: 06

Comment 3: A patient demographic table would be appropriate.

Response: We highly appreciate this constructive suggestion from the reviewer

Revision made: We have included a Supplementary Table 9 to provide hematological and biochemical parameters in healthy controls and in different diseased groups.

Supplementary Table 9: Clinicopathological analysis of hematological and biochemical parameters in healthy controls and the patients suffering from NSF, SFM_Others, CB, SA, VM and DF

	Healthy control (HC = 58)	Non-severe Falciparum Malaria (NSFM, n = 15)	Severe Falciparum Malaria (SFM, n = 38)	Other complications (SFM_others, n = 27)#	Cerebral Malaria (CB, n = 7)	Severe Anemia (SA, n = 5)φ	Falciparum malaria (FM, n = 53)‡	Vivax malaria (VM, n = 16)‡	Dengue Fever (DF, n = 15)
Hematological parameters									
Hemoglobin (g/dl)	13.3 (11.8-14.7)	12 (5-14)	10.5 (3.7-15.9)	11 (6.1-15.9)	11.7 (7.5-13)	4.25 (3.7-4.9)	11.2 (3.7-15.9)	10.29 (6.8-13.9)	13.55 (9.9-17.8)
Platelets	3 (1 - 4.3)	1.7 (0.46-4)	0.75 (0.19-3.1)	0.895 (0.19-3.1)	1.3 (0.3-2.9)	0.66 (0.2-2.05)	1.05 (0.19-4)	2.3 (0.8-4.6)	0.56 (0.1-1.3)
Biochemical parameters									
SGOT	36 (23 -74)	29 (23-110)	62 (11-340)	62 (12-340)	37 (11-210)	68 (24-234)	36 (23-74)	54 (11-340)	45 (14-140)
SGPT	38 (19 -87)	35 (15-114)	81 (15-459)	81 (15-243)	92 (32-300)	57 (17-459)	38 (19-87)	63 (15-459)	41 (18-92)
ALP	104 (66-145)	108 (66-289)	157 (12-427)	159 (43-427)	157 (13-254)	123 (12-188)	104 (66-145)	145 (12-427)	93 (42-256)
Total Bilirubin	0.9 (0.3-2)	1.2 (0.6-2.9)	1.9 (0.45-11.96)	1.6 (0.45-7.2)	1.9 (0.7-5.7)	1.35 (1.1-2.9)	0.9 (0.3-2)	1.4 (0.45-7.2)	0.9 (0.9-2.1)

Data represented as mean (lowest value – highest value).

#SFM_others, n is variable for different parameters: SGPT and ALP (n = 25), and Total Bilirubin (n = 23).

φ SA, n is variable for different parameters: Total Bilirubin (n = 4).

‡ FM, n is variable for different parameters: SGPT (n = 52), and Total Bilirubin (n = 49).

* VM, n is variable for different parameters: Platelets, SGPT and ALP (n = 17), and Total Bilirubin (n = 4)

Changes has been incorporated on following Page no.: 20

Comment 4: I also did not see any reference to proteomic repository data download, or inclusion of proteomic data that fits the criteria for publishing such results. Can you please include the appropriate peptide and protein level data as well as deposit it in an appropriate repository?

Response: We wish to clarify that we had submitted all the raw quantitative proteomics data in standard data repositories (PRIDE and Peptide Atlas) and the details were provided in the “Data and materials availability”. Required links and passwords were also provided for accession of the data sets by the reviewers.

Data and materials availability: “All processed data associated with this study are present in the manuscript and details provided in the Supplementary Materials. Raw MS data and search output files for TMT-based quantitative proteomics analyses described in this article are deposited to the ProteomeXchange Consortium via the PRIDE (Perez-Riverol et al. 2016) partner repository with the dataset identifier PXD014991 (Link: <https://www.ebi.ac.uk/pride/archive/login>, Username: reviewer91815@ebi.ac.uk, Password: ScHxqhnv). Targeted proteomic data (Dataset Identifier: PASS01467) are deposited in the Peptide Atlas and can be accessed through this link: <ftp://PASS01467:FR4478ci@ftp.peptideatlas.org/>. (Server name: ftp.peptideatlas.org Username: PASS01467, Password: FR4478ci)(Farrah et al. 2012).”

Code availability: The custom code for MS data analysis (TMT6/10plex) and machine learning have been deposited at <https://github.com/vipin786/R-code-for-MS-data>.

We also provided the identified proteins and peptide details as supplementary tables (Excel format). However, the supplementary tables (processed data files) were distorted during the conversion of the Excel files to the PDF while submitting our manuscript, which unfortunately remained unnoticed. Our sincere apology for any inconvenience that may have happened due to this unintentional formatting error.

Revision made: We have rectified the formats of the supplementary tables (processed data files) in the revised manuscript.

Changes has been incorporated on following Page no.: 28-29

Response to the Comments of Reviewer # 2

The present manuscript uses a proteomic platform investigating plasma from patients with malaria. It is one of the most comprehensive so far as it includes comparisons between malaria severity states (already described in the literature) but also compares the two most prevalent species of Plasmodium (falciparum and vivax). It also includes disease controls, mainly Dengue fever. The present study also describes the detection of parasite proteins within plasma samples.

The data analysis of the complex proteomic data is step-wise and led to description of potential pathways of pathogenesis for the different disease groups. Potential biomarkers (of human or parasite origin) were validated by MRM, which is crucial for a study with such a small cohort.

The study is well designed and conducted. It is however not the first proteomic study performed, even if it is the first time using an untargeted approach utilizing TMT-based labelling, more reproducible than label-free. According to the title, one may have expected a more advanced technology than the one used, which is now well-established in the proteomic community and on the verge of being outdated to more throughput methodologies (eg. SWATH).

One major weakness of the study is the cohort size. Even though it includes 143 total subjects, the division in the numerous disease groups limits the statistical power, with some groups under 5 patients. This has consequences in the design of the study. No discovery-validation cohort split up which would have been necessary to validate the regression model built for malaria prediction. The validation using MRM carried out is essential but a method validation. Cohort validation remains essential. Unless the MRM was done in a different cohort, which was not made clear.

Moreover, as a complex study with stepwise comparisons, it was made more confusing by the many different ways of analyzing and representing the data (see details in comments below) and the inconsistencies from one comparison to the other. The consequence was that the main result messages are buried and not clear.

Related to the previous comment, the conclusion is not clear. The present manuscript does indeed 'provide a comprehensive landscape of plasma proteome alterations in different levels of malaria severity'. However, the novel pathways discovered are not obvious as most of them corroborates the pathogenetic pathway hypothesis currently researched. Finally, the tone to which the manuscript is written is open to over interpretation. Specific examples are in the comments below. I have many comments of varying significance. They are listed below following the paper's chronological order.

Thank you for considering our manuscript "well designed and conducted". We are extremely thankful to the reviewer for an insightful review of our article and for these highly encouraging comments and productive suggestions to improve the manuscript.

We agree with the reviewer that there are earlier studies on malaria proteomics from our and other research groups. However, we wish to clarify all of the earlier serum or plasma

proteomics studies (including our previous reports) described the parasite-induced alterations in human serum/plasma in specific species of Plasmodial infections. In the present study, we systematically investigated the proteome level alterations of human (adult patient populations) plasma in non-severe and severe, falciparum, and vivax malaria comparatively and comprehensively. In addition to the host proteome, we also investigated the parasite proteins in the plasma samples of malaria patients. According to the suggestion from the reviewer, we have toned down and modified the statements regarding the novelty of the present study.

Multiplexing using stable isotope labelling (specifically TMT reagents) results in increased throughput, higher precision, better reproducibility, reduced technical variations and very fewer missing values than label-free methods (Erickson et al. 2015; Mertins et al. 2018; O'Connell et al. 2018). Very recently, with the introduction of the TMTPro-16 plex reagents, multiplexity for proteome-wide measurements by using TMT reagents is even increased further (Li et al. 2020). In the present scenario, according to the published literature, TMT-based multiplexing is considered as an extremely reliable technical approach which is most extensively used in MS-based multiplexed quantitative proteomics (compared to the other approaches).

We concur that as multiple groups of the patients (different species of plasmodium, different severity levels of malaria, and different controls) are analyzed in our study, some of the cohorts are relatively smaller. We would like to clarify that in the validation phase (MRM-based assays) of the study, we selected a few subjects from the discovery phase randomly, and also recruited new subjects in for most of the study groups. Therefore, the validation results reflect "method validation" as well as "cohort validation".

We provide detailed responses to the comments and queries below.

Comment 1: The vocabulary employed to describe the overall study include “first comprehensive attempts by deep proteome investigations...” and use of “multiplex highly sensitive quantitative proteomics”. It is not the first comprehensive attempt (Bachman et al 2014), although this was using a targeted proteomic approach. ‘Deep proteome’ means including 1,000s of proteins as most of the studies now reach 4,000-10,000 even in semi-quantitative studies using 2-dimensional LC-MS/MS. Although, in fairness, plasma short linear range makes it more difficult.

Response: We highly appreciate these valued criticisms and suggestions from the reviewer. We concur that an earlier study by Bachmann et al. provided a comprehensive map of plasma proteome alterations in children with cerebral malaria using an antibody suspension bead arrays-based targeted approach (Bachmann et al. 2014).

We agree with the reviewer that using the ultra-sensitive next-generation mass spectrometry platforms it is now possible to quantify 10,000 or more proteins in human cell lines or tissue samples. However, the achievable coverage for plasma or serum samples remained substantially lower due to the extremely dynamic range of protein concentration and the presence of several highly abundant proteins (Anderson and Anderson 2002; Geyer et al.

2017; Ray et al. 2011). To the best of our knowledge, the most comprehensive MS-based studies (method) reported hitherto are able to quantify around 2000-4000 proteins in human plasma (Keshishian et al. 2017).

Substantial prefractionation (such as 20 or more fractions per sample) can increase proteome coverage (Keshishian et al. 2017). At the same time, it increases the sample processing time and more importantly the analysis cost drastically (due to a huge increase in mass spec running time). While it is possible to include such extensive prefractionation in the small-scale protocol development studies, from a realistic outlook it often turns difficult in large-scale studies (like ours) where several samples are processed. There is always a need to make a balance between how to obtain the deep proteome coverage vs. sample processing time/cost/ availability of Mass Spec run time etc.

Revision made: We have modified the statements regarding the novelty and proteome coverage of the present study and portrayed it as one of the most comprehensive blood-based untargeted proteomic analyses on falciparum and vivax malaria.

Changes has been incorporated on following Page no.: 02

Comment 2: The first time the authors use abbreviations for the disease groups, could they write them fully (line 107 p.6).

Response: We would like to thank the reviewer for this suggestion regarding the usage of abbreviations in our manuscript.

Revision made: In the revised manuscript, acronym expansions are provided during the first usage of the abbreviations. We have also included an additional supplementary table 1 to clarify the numbers of samples that were analyzed for different experimental groups (along with their abbreviations) in the analysis of clinical parameters, and in the discovery and validation phase quantitative proteomics analyses.

Following changes are incorporated in the text:

We have incorporated the changes in the manuscript. “A total of 106 subjects were analyzed in the discovery phase quantitative proteomics, while 111 subjects were included in the targeted validation study. 143 subjects were included for analysis of the clinical parameters. The following cohorts were included in this study – Healthy control (HC), Non-severe falciparum malaria (NSFM), Severe falciparum malaria (SFM), Cerebral malaria (CB), Severe anemia (SA), Control for severe anemia (CSA), Control for cerebral malaria (CCB), Non-severe dengue fever (NSD), Severe dengue (SD), Non-severe vivax malaria (NSVM) and Severe vivax malaria (SVM). The subjects were recruited from different malaria epidemic regions in India and the number of samples analyzed for each group has been demonstrated in Supplementary Table 1.”

Changes has been incorporated on following Page no.: 06

Comment 3: Line 108 p.6: This is probably due to my ignorance but what do the authors mean by ‘multi-centric’ study? Do they mean multi-centre?

Response: We would like to thank the reviewer for this query. We wish to clarify that by ‘multi-centric’ study we mean to convey that the patients were recruited in this study from multiple tertiary care centers in two different malaria endemic regions of India - Kolkata, and Bikaner. We understand that this phrasing may not be clear to the readers. Therefore, we have amended this phrasing in the revised manuscript.

Revision made: We have removed this parlance ‘multi-centric’ in the revised manuscript.

Changes has been incorporated on following Page no.: 06

Comment 4: Line 111 p.6: ‘sequential decrease’. There is no sequence as there is no repeated samples over time of same patient.

Response: We appreciate this remark of the reviewer and admit that there is no repeated measurement performed from the same patients in a longitudinal manner. Here we mean to say that the magnitude of alterations (decrease) in platelet counts and Hb levels correlates with disease severity, i.e. the SFM patients had much lower levels of these hematological parameters compared to the NSFAM patients.

Revision made: We have amended this statement as per the suggestion from this reviewer. Interestingly, the magnitude of alterations (decrease) in platelet counts and Hb levels were found to be more prominent in severe malaria, i.e. the SFM patients had lower levels of these hematological parameters compared to the NSFAM patients.

Changes has been incorporated on following Page no.: 06

Comment 5: Supplementary Figure 4: The authors have chosen to show in supplementary material the QC graphs, however they have not explained or commented for the non-expert readers what these graphs means. I am not suggesting writing a new paragraph but just a description in the figure caption. From theses graphs, one might conclude is that there is high variance in samples in the falciparum comparisons with lower reproducibility than in other comparisons. An analytical comment in the caption would be welcome.

Response: This suggestion from the reviewer is greatly appreciated. We totally agree with the reviewer that explanation of the QC results shown in Sup Fig. 4 would be essential for many readers from diverse fields who are not experts of the proteomics technologies. There was redundancy in the representation of the QC results as the raw data were represented as Whisker (Box) plots, while both raw and normalized data were again shown as Density plots. In order to get rid of this redundancy, we have kept only the Density plots before and after normalization.

Revision made: Supp. A detailed explanation for the QC results are provided in the legend of the figure (Supplementary Fig. 4).

Supplementary Fig. 4. Quality control check and pre-processing of proteomic data. **a-c** Density plots representing the proteome expression (TMT-based quantification) profiles of Falciparum malaria **a**, Vivax malaria **b** and Dengue **c** data sets. We plotted the protein

abundance values from raw data, which were sequentially normalized by reference pool and quantile normalization. These profile density plots are representing the distribution of abundance of all proteins in one plot. Y axis shows the no. of proteins count and x-axis shows the abundance of the proteins in log₂ values. We observed that the raw data were not perfectly normally distributed, while after the normalization, the data are normally distributed (symmetrical distribution indicating that the normalization methods effectively reduced the variations observed in the raw data sets).

Comment 6: Figure 1: Panel b shows ‘label-free’ and TMT-labelled samples but there is no mention in the text and this may be confusing. It only becomes clear when reading the parasite identification paragraph. Because the authors used a paragraph to describe the workflow and draw a very detailed and helpful diagram, I think the text should be more precise and parallel to what the diagram represent to avoid confusion.

Response: We appreciate this comment. In the revised manuscript, we have clarified that TMT-based multiplexed quantitative proteomics was used to map the plasma proteome (host) alterations, while we used a label-free quantitation (LFQ) approach for detection and quantification of the parasite (*Plasmodium falciparum*) proteins in host plasma. We have also made it clear in the schematics of the workflow in Fig. 1b and in its legend.

Revision made: Malaria and dengue (another infectious disease control) samples were confirmed using different diagnostic techniques, and the positive cases were incorporated in this TMT-based multiplexed quantitative proteomic analysis (Fig. 1). Such multiplexing using stable isotope labeling provides increased throughput, higher precision, better reproducibility,

reduced technical variations, and lower number of missing values (Mertins et al. 2018; O'Connell et al. 2018; Ray et al. 2019). TMT-based multiplexed quantitative proteomics was used to map the plasma proteome (host) alterations, while we used a label-free quantitation approach for detection and quantification of the parasite (*Plasmodium falciparum*) proteins in host plasma.

Changes has been incorporated on following Page no.: 07

Fig. 1 Schematic representation of the experimental strategy used in discovery-phase proteomics and for targeted validation of potential biomarkers in plasma samples. **a**

Malaria samples were diagnosed by Microscopy and RDT, and some randomly selected samples were confirmed further using PCR. Similarly, dengue samples used as a non-malaria febrile infectious disease control was diagnosed using IgM and NS1 antigen. **b Depleted plasma samples were trypsin digested, and TMT labelled for studying the plasma proteome (host) alterations in malaria patients. A label-free quantitation (LFQ) approach was used for detection and quantification of the parasite (*Plasmodium falciparum*) proteins in host plasma.** **c.** Protein search was performed using proteome discoverer 2.2, and then PSMs files were analyzed using the MSstatsTMT package. **d** Machine learning was performed using the elastic net regression method. **e** Top hits of differentially abundant proteins were selected for validation using Multiple Reaction Monitoring (MRM) assays. *Patients number is based on available clinical data.

Comment 7: The data missing is the number of cases and demographics for each comparison cohort.

Response: We highly appreciate this constructive suggestion from the reviewer.

Revision made: We have included a supplementary table 9 to provide hematological and biochemical parameters in healthy controls and in different diseased groups. We have also included an additional supplementary table 1 to clarify the numbers of samples that were analyzed for different experimental groups (along with their abbreviations) in the analysis of clinical parameters, and in the discovery and validation phase quantitative proteomics analyses.

Changes has been incorporated on following Page no.: 20

Supplementary Table 9: Clinicopathological analysis of hematological and biochemical parameters in healthy controls and the patients suffering from NSF, SFM_Others, CB, SA, VM and DF

	Healthy control (HC = 58)	Non-severe Falciparum Malaria (NSFM, n = 15)	Severe Falciparum Malaria (SFM, n = 38)	Other complications (SFM_others, n = 27)#	Cerebral Malaria (CB, n = 7)	Severe Anemia (SA, n = 5)φ	Falciparum malaria (FM, n = 53)*	Vivax malaria (VM, n = 16)*	Dengue Fever (DF, n = 15)
Hematological parameters									
Hemoglobin (g/dl)	13.3 (11.8-14.7)	12 (5-14)	10.5 (3.7-15.9)	11 (6.1-15.9)	11.7 (7.5-13)	4.25 (3.7-4.9)	11.2 (3.7-15.9)	10.29 (6.8-13.9)	13.55 (9.9-17.8)
Platelets	3 (1 - 4.3)	1.7 (0.46-4)	0.75 (0.19-3.1)	0.895 (0.19-3.1)	1.3 (0.3-2.9)	0.66 (0.2-2.05)	1.05 (0.19-4)	2.3 (0.8-4.6)	0.56 (0.1-1.3)
Biochemical parameters									
SGOT	36 (23 -74)	29 (23-110)	62 (11-340)	62 (12-340)	37 (11-210)	68 (24-234)	36 (23-74)	54 (11-340)	45 (14-140)
SGPT	38 (19 -87)	35 (15-114)	81 (15-459)	81 (15-243)	92 (32-300)	57 (17-459)	38 (19-87)	63 (15-459)	41 (18-92)
ALP	104 (66-145)	108 (66-289)	157 (12-427)	159 (43-427)	157 (13-254)	123 (12-188)	104 (66-145)	145 (12-427)	93 (42-256)
Total Bilirubin	0.9 (0.3-2)	1.2 (0.6-2.9)	1.9 (0.45-11.96)	1.6 (0.45-7.2)	1.9 (0.7-5.7)	1.35 (1.1-2.9)	0.9 (0.3-2)	1.4 (0.45-7.2)	0.9 (0.9-2.1)

Data represented as mean (lowest value – highest value).

#SFM_others, n is variable for different parameters: SGPT and ALP (n = 25), and Total Bilirubin (n = 23).

φ SA, n is variable for different parameters: Total Bilirubin (n = 4).

*FM, n is variable for different parameters: SGPT (n = 52), and Total Bilirubin (n = 49).

*VM, n is variable for different parameters: Platelets, SGPT and ALP (n = 17), and Total Bilirubin (n = 4).

Sample detail table

Supplementary Table 1: Details of the sample numbers for the discovery phase quantitative proteomics, targeted validation, and clinicopathological analyses.

Experimental groups	Discovery phase quantitative proteomics (TMT-based multiplexed quantitation)	Targeted validation (MRM assay)	Analysis of clinical parameters
Falciparum malaria (FM)			
Non-severe Falciparum malaria (NSFM)	8	10 (DP = 05, NS = 05)	15
SFM_others	12	7 (DP = 04, NS = 03)	26
Severe anemia malaria (SA)	6	4 (DP = 04, NS = 00)	5
Cerebral malaria (CB)	6	3 (DP = 01, NS = 02)	7
Negative control: Control for severe anemia (CSA)	4	3 (DP = 03, NS = 00)	*
Negative control: Control for cerebral malaria (CCB)	4	3 (DP = 03, NS = 00)	*
Healthy Control (HC)	10	24 (DP = 18, NS = 06)	58
Disease Control: Non-severe dengue (NSD)	4	10 (DP = 04, NS= 06)	15
Disease Control: Severe dengue (SD)	12	17 (DP = 17, NS= 00)	
Vivax malaria (VM)			
Non-severe vivax malaria (NSVM)	12	10 (DP = 07, NS= 03)	17
Severe vivax malaria (SVM)	20	20 (DP = 20, NS = 00)	
Healthy Control (HC)	04	24 (DP = 18, NS = 06)	58
Disease Control: Non-severe dengue (NSD)	4	10 (DP = 04, NS= 06)	15
Disease Control: Severe dengue (SD)	12	17 (DP = 17, NS= 00)	

SFM_Others: It includes case of severe falciparum malaria patients excluding CB and SA complications.

*not considered for clinical analysis.

DP – subjects from discovery phase, NS – new subjects.

Comment 8: Line 148 p.7: ‘296 proteins were present in more than 60%’. It is unclear how the data was filtered and processed. Was the resultant data analysis was based on the 296 or how the missing value was dealt with? The information in the materials and methods warrants more details.

Response: We appreciate this critical observation of the reviewer. We wish to clarify that 1495 proteins were identified across 60 samples involved in falciparum malaria study, among which 296 proteins were quantified in more than 36 samples (i.e. $\geq 60\%$ of the samples). The proteins that were quantified with ≥ 1 unique peptide and detected in at least 60% of the samples were only selected for the subsequent differential analysis. Missing value imputation was performed for only those proteins, which were quantified in $\geq 60\%$ of the samples. Of note, the differentially abundant proteins which were quantified in all the samples were prioritized in the validation phase.

Revision made: We have also included a tabular workflow for demonstrating the cut-off filters (and the number of proteins corresponding to each threshold) that were used to select the proteins for the final comparative analysis. We have also included the required details in the Materials and methods section of our revised manuscript.

Steps	Falciparum malaria
Total PSMs	239034
Total identified proteins	1495
Proteins with ≥ 1 unique peptide	957
Proteins with ≥ 2 unique peptides	538
Proteins quantified in $\geq 60\%$ of the samples (≥ 1 unique peptide)	296
Proteins quantified in $\geq 60\%$ of the samples (≥ 2 unique peptides)	271
Proteins quantified in all samples (≥ 1 unique peptide)	155
Proteins quantified in all samples (≥ 2 unique peptide)	153

Following changes are incorporated in the text:

The protein level summarization and significance analysis of the proteins was performed in R using the MSstatsTMT package. In brief, The PSMs were preprocessed in PD and were converted into the required input format for MSstatsTMT using an in-house R code. The protein abundance was calculated based on the peptide quantification using the “protein

summarization” script (MSstatsTMT) (Huang et al. 2019). It includes normalization between the MS runs using reference pool channels and the imputation of missing values before summarizing peptide level data into protein-level data. In the protein summarization method, MSstats assumes missing values are censored and then imputes the missing values using the accelerated failure model. Missing value imputation was performed for only those proteins which were quantified in $\geq 60\%$ of the samples. Then, a moderated T-test was performed on quantile normalized values using “Group-Comparison-TMT” script in R and p -value < 0.05 were considered as statistically significant (Huang et al. 2019).

Changes has been incorporated on following Page no.: 24-25

Comment 9: What was the test used to choose the significantly altered proteins and more particularly was there a correction applied for multiple comparisons and which one? With 1495 supposedly included in the comparison, a multiple comparison test is necessary. Also, what were the other criteria for positive identity (eg. No. peptide >2 ?)

Response: We would like to clarify that, the proteins that were quantified with ≥ 1 unique peptide and detected in at least 60% of the samples were selected for the subsequent differential analysis. We have used ≥ 1 unique peptide as a threshold as the proteins identified and quantified even with one peptide in a next-generation ultra-sensitive orbitrap instrument could also be considered for subsequent analyses. However, majority of these proteins (91%, 271 out of 296) that were detected in at least 60% of the samples were quantified with ≥ 2 unique peptides.

p -values obtained from a paired t-test were used to determine the significance of differences in protein abundances between control and diseased groups (comparison between any two groups). p -value < 0.05 were considered as statistically significant. We also calculated adjusted p -value [Benjamin–Hochberg (BH)] for multiple comparisons using the MSstatsTMT package in R. We used p -value < 0.05 threshold for most of the analyses including heat map, volcano plot, and pathway analysis in order to be more inclusive in terms of selection of the differentially abundant proteins. However, BH corrected p -value [p -value (adjusted) < 0.05] value was considered in the selection of proteins for machine learning analysis.

Revision made: We have included a statistical analysis section and provided all these details in that section. We have also included a tabular workflow for demonstrating the cut-off filters (and the number of proteins corresponding to each threshold) that were used to select the proteins for the final comparative analysis (Fig. 2a). We have also included a Supplementary table for demonstrating the uncorrected and corrected (BH) p -values for all the differentially abundant plasma proteins (Supplementary Table 2). In the Table 1B, we have marked those proteins that remained significant after false discovery rate (FDR) correction (Benjamini-Hochberg).

Supplementary Table 2 (Microsoft excel format):

https://drive.google.com/drive/folders/1fB11zVEXYQCU5_jj6oX0H2McMIu-26QT?usp=sharing

Changes has been incorporated on following Page no.: 25

Comment 10: I have concerns about the amount of data with missing value and how much missing value imputation was performed for this data. Could you confirm what the filter cut-off to exclude a protein was and how many were in the final comparison and have this number in the figure 2a diagram. The only reference to imputation is in the materials and methods for the Machine learning section and it is not very detailed. A statistical analysis section in the materials and methods section might resolve these issues.

Response: We would like to thank the reviewer for these insightful comments and queries. We would like to clarify that the missing values were imputed using R-script MBimpute under the MSstatsTMT package (Huang et al. 2019). In this analysis, accelerated failure time (AFT) model was applied for missing value imputation (Qi et al. 2018). We have selected the proteins that are quantified with ≥ 1 unique peptide and detected in at least 60% of the samples for subsequent differential analysis. Missing value imputation was performed for only of those proteins which were quantified in $\geq 60\%$ of the samples. Of note, the differentially abundant proteins which were quantified in all the samples were prioritized for the validation phase.

Revision made: As suggested by this reviewer, we have included a statistical analysis section and provided all these details in this section. We have also included a tabular workflow for demonstrating the cut-off filters (and the number of proteins corresponding to each threshold) that were used to select the proteins for the final comparative analysis (Fig. 2a).

Following changes are incorporated in the text:

The proteins that were quantified with ≥ 1 unique peptide and detected in at least 60% of the samples were selected for the subsequent differential analysis. The PSMs values were exported from Proteome Discoverer (PD 2.2) for combined analysis for FM, VM and DF (FM: n = 10 reactions, 90 fractions, VM; n = 4, 25 fractions, and DF; n = 4 reactions, 24 fractions). The protein level summarization and significance analysis of the proteins was performed in R using the MSstatsTMT package. In brief, The PSMs were preprocessed in PD and were converted into the required input format for MSstatsTMT using an in-house R code. The protein abundance was calculated based on the peptide quantification using the “protein summarization” script (MSstatsTMT) (Huang et al. 2019). It includes normalization between the MS runs using reference pool channels and the imputation of missing values before summarizing peptide level data into protein-level data. In the protein summarization method, MSstats assumes missing values are censored and then imputes the missing values using the accelerated failure model. Missing value imputation was performed for only of those proteins, which were quantified in $\geq 60\%$ of the samples. Then, a moderated T-test was performed on quantile normalized values using “Group-Comparison-TMT” script in R and p -value < 0.05 were considered as statistically significant.

Changes has been incorporated on following Page no.: 24-25

Comment 11: Fig.2b: In the non-severe to severe falciparum comparison, why not comparing these two states directly, particularly for the volcano plot (Figure 2b). Directly comparing

severe to non-severe cases might bring out the specific markers to severe disease. Same can be said about the CB vs SA comparison and SVM vs NSVM (Fig.3a and 4a)

Response: We appreciate this comment from the reviewer, and the suggested plots have been included in revised main Figure 2b, Figure. 3a, and Figure 4a.

Figure 2b, Figure 3a, Figure 4a

Comment 12: The volcano plots are obsolete to the manuscript, particularly as top proteins are not shown. Similarly, how well do cluster NSFVvsSFM, SVMvsNSVM, CBvsSA and DFvsSD in Supplementary Fig 5, 7 and 9

Response: According to the suggestion from the reviewer we have labeled the most highly altered proteins in the volcano plots (Figure 2b, 3a and 4a) and revised the suppl. Figure 5, and 7.

Supplementary Fig. 5 Unsupervised clustering of significantly altered proteins ($p < 0.05$) in falciparum malaria. a Heat map of individual samples for non-severe falciparum malaria (NSFM) and severe falciparum malaria (SFM). b Heat map of individual samples for cerebral malaria (CB) and severe malaria anemia (SA).

Supplementary Fig. 7 Unsupervised clustering of differentially abundant proteins ($p < 0.05$) in vivax malaria. Heat map of individual samples for non-severe vivax malaria (NSVM) and severe vivax malaria (SVM)

Comment 13: What criteria were selected to include proteins for the supervised heatmap?

Response: Plasma proteins that were quantified with ≥ 1 unique peptide and detected in at least 60% of the samples were selected for the subsequent differential analysis. We have considered all significantly altered proteins ($p < 0.05$) for the supervised heatmap analysis. p -values obtained from a paired t-test were used to determine the significance of differences in protein abundances between control and diseased groups (comparison between any two groups). p -value < 0.05 were considered as statistically significant.

Comment 14: In Figure 2 the PCA plot is described as being ‘performed on all identified proteins’ (line 153 p.7), it is unclear whether it is indeed based on the 1,495 identified proteins. In Figure 3 it is described as performed on ‘differentially abundant proteins’ (line 196 p.9) and ‘Common and specific significantly altered proteins’ (Figure 3c caption), which is clear. Is there an inconsistency?

Response: We appreciate this remark. We would like to clarify that and that the Principal component analysis (PCA) was not performed on all identified proteins. We included only the significantly altered proteins ($p < 0.05$) in this analysis. This is a typo, and we sincerely apologizing for this unintentional error.

Revision made: We have rectified this mistake in the revised manuscript.

Following changes are incorporated in the text:

Principal component analysis (PCA) was performed on the differentially abundant proteins ($p < 0.05$), and it successfully distinguished between the three study populations; HC, NSF, and SFM (Fig. 2c).

Changes has been incorporated on following Page no.: 08

Comment 15: 8 out of the 12 depleted proteins are in the list of proteins altered. Is the fact that they are present in relative abundance suggesting the depleting columns were saturated? It is difficult to assess as multi-protein depletion was carried out. Nevertheless, should these be taken in account or changes validated using total plasma (or albumin depleted plasma)?

Response: This is a very useful remark and we highly appreciate the reviewer for such an in-depth review of our manuscript. We substantially optimized the plasma sample volume for obtaining the maximal removal of the high abundant proteins (top 12) from the crude plasma samples. However, there were some residual proteins remained after the depletion as none of these commercially available depletion columns/kits are 100% efficient.

We agree with the reviewer that it would not be accurate to measure fold-changes for these proteins after depletion as it may not reflect the actual alterations induced by the infection.

Revision made: We have excluded some of these depleted proteins from the altered protein list and those which were physiologically relevant to malaria pathobiology (FGA, FGB, APOA1, ORM1) are still considered. However, we have included a footnote stating that for these proteins fold-change analysis was measured after partial depletion.

Comment 16: Line 158p.8, TF is referred as Tissue factor. Line249 p.11, it is referred as Serrotransferin.

Response: We appreciate this observation of the reviewer. We have replaced the term “tissue factor” with “Serotransferrin” protein in the revised manuscript.

Changes has been incorporated on following Page no.: 11

Comment 17: There are many ways to study pathway or network analysis and some will complement each other and will give different views on the data. I believe it is a bit confusing

to have the analysis of the clusters (line 156-167 p.8) and the network analysis (which I understand to be drawn from NetworkAnalyst?) (Lines 169-179 p.8). I do not suggest to remove the heatmap or the cluster quadrants but in the text choosing which list of pathway and proteins to describe (network analysis) would make it less confusing, particularly as they overlap greatly.

Response: We are thankful to the reviewer for this suggestion regarding the pathway analysis. We have removed the overlapping details in the text, which were explaining about the heatmap, and we have kept only the network analysis section drawn by NetworkAnalyst.

Revision made: Supervised hierarchical clustering of the proteome profiles stratifies the different groups of malaria patients (Fig. 2d). Significantly altered proteins from this study and proteins from literature (Bachmann et al. 2014; Ray et al. 2015) were considered for protein network analysis (Fig. 2e, Supplementary Table 2) using NetworkAnalyst. We observed the overlapping of the clusters for a group of the proteins between heatmap and network analysis. The platelet degranulation process was highly active in falciparum malaria. Histidine-rich glycoprotein (HRG), Plasminogen (PLG), Platelet factor 4 (PF4), Profilin-1 (PFN1), Kininogen-1 (KNG1) and Platelet basic protein (PPBP) were down-regulated in NSFM as compared to SFM, while Alpha-1-antitrypsin (SERPINA1) and Alpha-2-antiplasmin (SERPINF2) were found to be upregulated in SFM as compared to NSFM (Fig. 2e and Supplementary Table 2).

Supplementary Table 2 (Microsoft excel format):

https://drive.google.com/drive/folders/1fBI1zVEXYQCU5_jj6oX0H2McMIu-26QT?usp=sharing

Changes has been incorporated on following Page no.: 08

Comment 18: With the absence of a cohort Table with demographics, it is difficult to know why the 'vivax' comparison only has 4 controls, whilst the 'falciparum' comparison has 10. Was it from a different cohort altogether? Why are the controls not mixed? Is there any demographic differences between the 2 set? It might be more robust to directly compare NSVM to SVM without the HC.

Response: We accept that the number of controls (HC) analyzed in TMT-based comparative quantitative proteomics analysis with vivax malaria is relatively much lower than what we used for falciparum malaria. The lower number of HC used in vivax malaria discovery phase analysis was due to the technical limitations of sample processing (TMT reagent, Mass spec running time, etc.). However, we wish to clarify that a higher number of HC samples were analyzed in the validation phase MRM assays for comparative analysis with vivax malaria (a similar number that has been used for falciparum malaria).

Revision made: According to the reviewer's suggestion, we have performed a direct comparative analysis between SVM and NSVM and included the results in the revised manuscript (Supplementary Table 5). We have also provided a new supplementary table to clarify the exact number of samples that have been used in the discovery and validation phases of quantitative proteomics analyses in vivax and falciparum malaria (Supplementary Table 1).

Supplementary Table 5 (Microsoft excel format):

https://drive.google.com/drive/folders/1fB11zVEXYQCU5_jj6oX0H2McMIu-26QT?usp=sharing

Changes has been incorporated on following Page no.: 11 and 3

Comment 19: The comparison of non-severe versus severe dengue fever seems out of topic. It should be directly compared to the malaria cases alongside heatmap figure 2d and 4e. The CSA and CCB would be better appreciated in heatmap Figure 3f directly compared to SA and CB, respectively.

Response: We solely agree with the reviewer that comparison of the diseased controls such as dengue or CSA and CCB directly with the respective malaria samples would be much more informative.

Revision made: According to the suggestions from this reviewer, we have revised the following figures - Fig 2d, Fig 3f, Fig 4d.

Fig 2d: Heat map representation showing abundances of the altered proteins in NSFMs, SFMs & DF and different clusters were identified as 1, 2 and 3 on the basis of proteins abundance. in DF, Malaria (NSFM+SFMs) and SFMs respectively.

Fig 3f: Heat map representing discrimination of plasma protein abundances in CB, SA, SFM with other complications, and their negative controls (CCB+CSA) and different clusters were identified as 1, 2 in severe malaria (CB+SA+SFM), 3 in CB and 4 in SA on the basis of proteins abundance.

Fig 4d: Heat map representing discrimination of plasma protein abundances in NSVM, SVM and DF and different clusters were identified as 1, 2 and 3 on the basis of proteins abundance in SVM, NSVM and DF respectively.

Comment 20: Identification of *Plasmodium falciparum* parasite proteins. Any differences of detection in severe versus non-severe falciparum cases?

Response: We would like to clarify that we were able to detect and quantify quite a few parasite proteins (*Plasmodium falciparum*) only in the plasma samples of the severe falciparum malaria patients. Of note, most of these parasite proteins we detected in a substantial proportion of the SFM patients consistently (please see Table 1B). Such as Enolase, Heat shock protein 70 and 90, and Actin 1 were detected over 95% of the patient population (Table 1B). However, these parasite proteins were not detected (maybe below the detection limit of the assays) in the non-severe falciparum malaria patients i.e. the patients with much lower levels of parasitemia, which is consistent with our earlier studies in vivax malaria.

Parasite proteins that are consistently detected in almost all the severe malaria patients could be an indicator of severe infection as those are specifically detected in severe malaria patients. However, our findings need to be validated in larger clinical cohorts before the evaluation of their definite translational potentiality.

Comment 21: Lines 267-274 p.12: the paragraph is out of sequence

Response: We appreciate the reviewer for this critical observation. We admit that the concerned paragraph is out of context under this subheading. We have moved this under the host proteome section and expanded the details of the parasite proteome here.

Revision made:

We identified parasite proteins in plasma samples of falciparum malaria patients using a label-free quantitation approach. 23 proteins were identified across the 21 severe malaria samples (Supplementary Table 6). We further selected ten proteins based on the minimum number identified unique peptides (≥ 2 unique peptides) present in the samples for validation using the Multiple Reaction Monitoring (MRM) approach (Table 1B). Six out of ten parasite proteins were found consistently ($> 80\%$) present in the malaria patients. These parasite proteins include heat shock protein 90, enolase, actin I, heat shock 70 kDa protein, fructose-bisphosphate aldolase, and serine repeat antigen 4. These parasite proteins have catalytic activity and may play a vital role in the survival and virulence of the pathogen in the host system.

Heat shock protein 90 and 70 are reported as upregulated at temperature 38°C and above (Su and Wellems 1994), which helps the survival of the parasite in the erythrocytic stage of its life cycle in the hyperthermic condition of the host. Enolase along with heat shock 70 kDa and iron superoxide dismutase forms the DegP complex which protects the parasite from heat and oxidative stress in the host system (Sharma et al. 2014). Fructose-bisphosphate aldolase and actin have been reported to interact with TRAP and TRAP like protein (TLP) for sporozoites gliding and invasion (Nemetski et al. 2015). Serine repeat antigen 4 (SERA4) along with the other SERA member proteins, helps in maintaining the blood stage of the pathogen's life cycle. However, their clear physiological functions still remain unknown (Miller et al. 2002), and need to be investigated further.

Supplementary Table 6 (Microsoft excel format):

https://drive.google.com/drive/folders/1fB11zVEXYQCU5_jj6oX0H2McMIu-26QT?usp=sharing

Changes has been incorporated on following Page no.: 12

Comment.22: It is difficult to assess appropriateness of the model and more details (statistical) would be appreciative to the lay reader. How did it perform overall? What were the training and test sets composition and size? The resulting ROC curves show that the models are not much improved over 10 protein sets and are quite robust with low amounts of proteins.

Response: We would like to thank the reviewer for this query. Please find below a detailed explanation of the machine learning framework used, its performance and ROC curves for biomarker panels.

The elastic net regularized logistic regression method was used to classify: 1) dengue vs. malaria; 2) Type of malaria: FM vs. VM; 3) CB vs. SA. An explanation of why elastic net regularized logistic regression method was used is provided in response to comment 3.6.

k-fold nested cross-validation (k =10 for malaria vs. dengue, k = 10 for falciparum vs. vivax and k=5 for cerebral vs. severe malaria anemia) was used as it provides robust and almost unbiased parameter estimates and model performance evaluation even for small sample sizes (Rhenman et al. 2015; Vabalas et al. 2019). The nested cross-validation approach involves using – (i) k-fold inner cross-validation loop for hyperparameter tuning and model selection, (ii) k-fold outer cross-validation loop for evaluating the model selected by the inner cross-validation. The entire dataset was randomly split into k-folds, out of which the kth fold was used as a test set and the remaining k-1 folds were used for training purpose. For each split, a model was trained and validated on the training set using (inner) k-fold cross validation and tested on the held-out test set. The hyperparameters alpha and lambda were tuned by inner k-fold cross-validation over a grid of values ranging from 0 to 1 with a step-size of 0.1 for alpha and another grid of 100 values ranging from 10^{-2} to 10^2 for lambda respectively.

To evaluate model performance, ROC-AUC, balanced accuracy, F1-score and Kappa metrics were computed. These metrics were chosen because we had imbalanced classes. The average variable importance of a protein was measured as the weighted average of variable importance of that protein in all k predictive models in the outer k-fold cross-validation with weights being the balanced accuracy of each model. This implicates that the importance of a protein in the classification of samples is directly proportional to its occurrence in the predictive models which can classify the samples better. A schematic diagram of the method used here is shown in Figs. 1c and 5a. The proteins with non-zero variable importance from elastic net regularized logistic regression model were considered for further analysis. For validation of the selected proteins as good classifiers for dengue vs. malaria, FM vs. VM and CB vs. SA models, we created (i) heatmap (Fig. 5e, Supplementary Table 7) and (ii) three-dimensional PCA plot (Fig. 5b, Supplementary Table 7). To arrive at the best panel of biomarkers, six different biomarker models were created from different numbers of proteins selected out of all the important covariates identified in the machine learning model. The ROC curves for these

biomarker models shown in (Figs. 6b, 6c, and 6d) enabled comparison of different models based on their corresponding AUC values and confidence intervals.

The elastic net regularized logistic regression model hyperparameters (alpha and lambda) and performance metrics (balanced accuracy, F1-score and Kappa) for the three models were as follows: (i) dengue vs. malaria: 0.1, 0.26, 0.975, 0.967, 0.96 (ii) FM vs. VM: 0, 80.2, 1, 1, 1 and (iii) CB vs. SA: 0.04, 41.73, 1, 1, 1 respectively. It is evident from the model performance metrics that the resulting elastic net regularized logistic regression model is able to predict and classify dengue vs. malaria, falciparum vs. vivax malaria, and cerebral vs. severe malaria anemia cases almost perfectly. These results are quite fairly stable across all (outer) k-fold iterations (Supplementary table 7).

The ROC curves of six biomarker models show that the model performs very well with AUC > 0.8 even for small number of proteins and no significant improvement in AUC is observed with a larger number of proteins. Keeping parsimony in mind and using this as a criterion to decide the panel size, we identified the final panel of two biomarkers for each model using MRM-based mass spectrometric assays (refer to Validation using Multiple Reaction Monitoring (MRM) section). The elastic net regularized logistic regression model equations for the final biomarker models identified from MRM were as follows:

Dengue vs. Malaria:

$$\log_e \left(\frac{P_{\text{Dengue}}}{1 - P_{\text{Dengue}}} \right) = -0.4269 + 2.23 * \text{LRG1} - 1.7374 * \text{CP} \quad \dots\dots(\text{I})$$

Falciparum vs. Vivax Malaria:

$$\log_b \left(\frac{P_{\text{VM}}}{1 - P_{\text{VM}}} \right) = 0.0893 + 0.169 * \text{AZGP1} + 0.1425 * \text{HRG} \quad \dots\dots(\text{II})$$

Cerebral vs. Severe Malaria Anemia:

$$\log_b \left(\frac{P_{\text{CB}}}{1 - P_{\text{CB}}} \right) = -0.2118 - 0.2365 * \text{SERPINA3} + 0.4309 * \text{AHSG} \quad \dots\dots(\text{III})$$

where $P_{\text{Positive Class}}$ is the probability of event that Y=Positive Class. For these three models, the hyperparameters (alpha and lambda) and model performance evaluation metrics (balanced accuracy, F1-score, Kappa, ROC-AUC, ROC-AUC Confidence Interval) were as follows: (i) dengue vs. malaria: 0.22, 0.02, 0.72, 0.89, 0.46, 0.807, (0.666-0.947) (ii) FM vs. VM: 0.25, 4.47, 0.8, 0.77, 0.6, 0.865, (0.772-0.958) and (iii) CB vs. SA: 0.02, 41.10, 0.8, 0.83, 0.56, 0.944 (0.816-1) respectively. These models basically serve two purposes and they are as follows:

- i) They help assess the impact of a particular protein on the estimated log odds or the estimated odds of an individual, for example, being a dengue patient as opposed to being a malaria patient.

For illustration sake, let us consider the model equation I. Note that this model equation yields:

$$\begin{aligned} \frac{\hat{P}_{\text{Dengue}}}{1 - \hat{P}_{\text{Dengue}}} &= \exp(-0.4269 + 2.23 * \text{LRG1} - 1.7374 * \text{CP}) \\ &= \exp(-0.4269) * \exp(2.23)^{\text{LRG1}} * \exp(-1.7374)^{\text{CP}} \\ &= 0.6525 * 9.2999^{\text{LRG1}} * 0.176^{\text{CP}} \end{aligned}$$

This can be interpreted by observing that the estimated odds of an individual being a dengue patient as opposed to s(he) being a malaria patient increase multiplicatively (decrease multiplicatively) by 9.2999 (0.6525) for every unit increase in the value of the protein LRG1 (CP).

- ii) Note that the equation involving the estimated odds $\frac{\hat{P}_{\text{Dengue}}}{1 - \hat{P}_{\text{Dengue}}}$ given in I further leads to

$$\begin{aligned} \hat{P}_{\text{Dengue}} &= \frac{\exp(-0.4269 + 2.23 * \text{LRG1} - 1.7374 * \text{CP})}{1 + \exp(-0.4269 + 2.23 * \text{LRG1} - 1.7374 * \text{CP})} \\ &= \frac{0.6525 * 9.2999^{\text{LRG1}} * 0.176^{\text{CP}}}{1 + 0.6525 * 9.2999^{\text{LRG1}} * 0.176^{\text{CP}}} \end{aligned}$$

and this, in turn, enables one to classify a new patient as a dengue or a malaria patient depending upon whether \hat{P}_{Dengue} is greater than or equal to 0.5 or not for specified values of LRG1 and CP for the patient under consideration.

The other two logistic regression models corresponding to falciparum vs. vivax malaria or cerebral vs. severe malaria anemia could be interpreted analogously.

The combined ROC curve of two proteins (LRG1 and CP) for Dengue vs. Malaria

The combined ROC curve of two proteins (AZGP1 and HRG) for FM vs VM

The combined ROC curve of two proteins (AHSB and SERPINA3) for CB vs. SA.

Supplementary Table 7 (Microsoft excel format):

https://drive.google.com/drive/folders/1fB11zVEXYQCU5_jj6oX0H2McMIu-26QT?usp=sharing

Changes has been incorporated on following Page no.: 12-13 and 25-27

Comment 23: Could the authors be clear on whether the samples used for validation were the same as in the total proteome analyses?

Response: We highly appreciate this query from the reviewer. We would like to clarify that in the validation phase (MRM-based assays) of the study, we selected a few subjects from the discovery phase randomly, and also recruited new subjects in many of the groups (please see the Supplementary Table 1). Therefore, the validation results reflect "method validation" as well as "cohort validation".

Comment 24: Could the authors add a table with inferred changes in the 86 protein levels from validation alongside changes observed using discovery total proteome data for comparison (Supplementary). Or the number of protein showing the same trend between the two methodologies.

Response: We would like to clarify that in the MRM-based validation workflow we started with 86 potential target proteins, which corresponds to 403 peptides, 7049 transitions. The total 40 methods were used to accommodate all the transitions (approx. 176 transition/method). During the first phase of optimization, we were not able to detect six proteins and hence did not include that in subsequent list. We started to refine the transitions by running different plasma pool samples and some individual samples on different days to check the inter-day variation and stability of the peptides (Supplementary Figure 10b). We were able to finalize peptides with good dotp (> 0.80), which corresponds to 46 proteins (Supplementary Figure 10c). The scheduling of peptides was performed, and the run was carried out as a single method to quantify the proteins from falciparum malaria, vivax malaria, dengue and healthy control samples.

Revision made: We have included a Supplementary Table 8 showing the protein level alterations observed in the discovery (TMT-based) and the validation (MRM assays) studies. The table contains 46 proteins showing all the possible comparisons across the various clinical conditions.

Supplementary Figure 10: b The representative screenshot of the Skyline interface, which allows to select the best peptide by monitoring the peak shape. The peptides, which has distorted peak shape are deleted from the list of the peptides. **c** The transition of same peptide is optimized by scheduling the run. The dotp value shows the similarity between the obtained MS spectra with existing spectral library. The scale of dotp is between 0 and 1. So the values closer to 1, indicate more similarity with spectral library, and hence more significant.

Optimization of Transition

86 proteins, 403 peptides,
7049 transition, 40 methods

Refining transition

80 proteins, 261 peptides,
2641 transition, 15 methods

Scheduling of peptides

46 proteins, 128 peptides,
847 transition, 1 method

HC (n=24), FM (n=30)
VM (n=30), DF (n=27)

Figure: Steps to filter out the proteins from MRM experiment: The experiment was started with 86 potential target proteins corresponding to 403 peptides, and 7049 transitions. We were not able to detect six proteins from plasma samples and hence removed from the list. We considered 2641 transitions for refining and evaluated different samples and MS parameters to check the consistency of transition. We were able to finalize the good peptides (dotp > 0.80) and their transitions, which corresponds 46 proteins. The scheduling of peptides was performed and run as a single method to quantify the proteins from falciparum malaria, vivax malaria, dengue and healthy control samples.

We have included a supplementary table to show the trend of the differentially altered proteins quantified using TMT-based quantitative and MRM-based targeted proteomics in the revised manuscript.

Supplementary Table 8 (Microsoft excel format):

https://drive.google.com/drive/folders/1fB11zVEXYQCU5_jj6oX0H2McMIu-26QT?usp=sharing

Changes has been incorporated on following Page no.: 28

Comment 25: In most of the manuscript, proteins are described by their symbol but in Fig. 6b, c, d, they are described by uniprot ID. Would it be possible to change it to protein symbol for ease?

Response: According to the reviewer's suggestion the uniprot IDs are replaced by the Gene symbol in Fig. 6b, c, and d.

Fig 6b-d: The optimization for MRM method and able to finalize the ROC curve of the best panel of protein biomarkers, **b** Dengue vs. Malaria, **c** Falciparum vs. Vivax, and **d** Cerebral vs. Severe anemia malaria.

Comment 26: Lines 344-346 p.16: So far the most comprehensive studies are from targeted approaches (Bachman et al 2014 and Reuterward et al 2018) Although the present study uses a non-targeted approach, the amount of data is not more comprehensive and pathways observed are similar to those previously mentioned (including previous studies from the authors). Moreover, the linear regression model does not confer causality only associations and only a study containing successive samples from same patient would confer progression. Finally there is a contention as to whether there is a progression from non-severe to severe, particularly in CM when cases are so acute.

Response: We agree with the reviewer that there are earlier studies on severe malaria proteomics from our and other research groups. However, we wish to clarify all of the earlier serum or plasma proteomics studies (including our previous reports) described the parasite-induced alterations in specific species of Plasmodial infections. In the present study, we systematically investigated the proteome level alterations human plasma in non-severe and severe, falciparum, and vivax malaria in a comparative and comprehensive manner. In addition to the host proteome, we also investigated the parasite proteins in the plasma samples of malaria patients.

We also concur with the reviewer that the best mechanistic conclusions could be obtained from a longitudinal analysis of the patients who are turning to a severe and complicated form of the infection from a non-severe clinical manifestation. However, in reality,

obtaining such clinical samples is highly challenging in any tertiary care hospital. Moreover, as correctly emphasized by this reviewer, it is also not clear whether severe infections are always initiated as a non-severe one and gradually progresses in a severe form of the disease or severe manifestations may also appear directly all of a sudden (such as in the acute cases of cerebral malaria). These would be some interesting and imperative aspects of severe malaria to be investigated further.

Revision made: We have toned down the statement regarding the novelty of the present study.

Following changes are incorporated in the text:

To the best of our knowledge, this is one of the most comprehensive blood-based untargeted proteomic analyses on falciparum and vivax malaria to obtain novel insights into malaria severity and complexity.

Changes has been incorporated on following Page no.: 02

Comment 27: Similarly, in line 427, the statement ‘we also observed that dysregulation of hemostasis-related proteins causes severe anemia’ is erroneous. Again, there was no causality analysis only association.

Response: We would like to thank the reviewer for this clarification and suggestion. We agree with the reviewer that with our existing results from GO analysis, we can only confer that the differentially abundant proteins (hemostasis-related proteins) are associated with the clinical manifestation of severe anemia.

Revision made: According to the suggestion from this reviewer, we have amended the concerning statement in the revised manuscript.

Changes has been incorporated on following Page no.: 19

Comment 28: It is difficult to extract novel pathways discovered as most have been previously reported (as referenced in the discussion). The discussion would benefit from highlighting what are the novel pathways and proteins in a more pronounced and specific way.

Response: We appreciate the reviewer for this comment. We agree with the reviewer that there are earlier studies on falciparum and vivax malaria serum/plasma proteomics from our and other research groups which have defined altered abundances of several host proteins associated with different physiological pathways (Bachmann et al. 2014; Francischetti, Seydel, and Monteiro 2008; Moussa et al. 2018; Perkins et al. 2011; Ray et al. 2012, 2016, 201; Reuterswärd et al. 2018). Some of the previous studies have also demonstrated the role of the dysregulated proteins in either murine model or in-vitro culture of *Pf* parasite and then extrapolated the findings on the context of human physiology (Bauer et al. 2002; Coban, Lee, and Ishii 2018; Ockenhouse et al. 1992; Olszewski et al. 2009; Somner, Black, and Pasvol 2000; Wilson et al. 2011).

In the present study, we systematically investigated the proteome level alterations of human (adult patient populations) plasma in non-severe and severe, falciparum, and vivax malaria comparatively and comprehensively. We have shown the association of the host proteins that are dysregulated in malaria patients with different physiological pathways and explained the mechanism of proteins to define the different complications of severe falciparum malaria. We agree with the reviewer that many of our identified pathways are also reported previously (expectedly), while some of the biological networks such as platelet degranulation and integrin cell surface interaction in severe malaria and hemostasis in severe anemia identified in the present study were not reported earlier, or at least were not known to be associated with the specific complications or severity of falciparum malaria. Moreover, we were able to identify many dysregulated proteins within some of the physiological pathways in malaria, which were not known earlier (even though those pathways were known to be associated with falciparum or vivax malaria as per the earlier reports from our and other groups).

Revision made: According to the suggestion from this reviewer, we highlighted those pathways were not earlier known to be associated with the specific complications or severity of falciparum malaria earlier.

Comment 29: There was no discussion about the parasite proteins detected, how they can be used for future research and how they compare to previous studies (eg. Moussa et al 2018)

Response: This is a very important suggestion from the reviewer. According to reviewer's suggestions, we have discussed about the parasite proteins detected in severe malaria patients' plasma.

We would like to clarify that we quantified quite a few parasite proteins (*Plasmodium falciparum*) in the plasma samples of the severe falciparum malaria patients. Of note, most of these parasite proteins we detected in a substantial proportion of the SFM patients consistently (please see Table 1B). Such as Enolase, Heat shock protein 70 and 90, and Actin 1 were detected over 95% of the patient population (Table 1B). However, these parasite proteins were not detected (maybe below the detection limit of the assays) in the non-severe falciparum malaria patients i.e. the patients with much lower levels of parasitemia, which is consistent with our earlier studies in vivax malaria (Venkatesh et al. 2020).

Revision made: The identified parasite proteins such as fructose-bisphosphate aldolase, heat shock protein 90, heat shock protein 70, enolase, and serine repeat antigen 4 from plasma samples have shown the antibody response from plasma samples in another study from our lab (Venkatesh et al. 2019). Moussa et al. has also reported fructose-bisphosphate aldolase, serine repeat antigen protein and histone H3 from pediatric plasma samples (Moussa et al. 2018) (Moussa et al. 2018). Most of these proteins play a very important role in catalytic activity and protein binding (Table 1B). However, *P. vivax* does not express *var* genes and hence binding to erythrocytes is very less as compared to *P. falciparum* (Phillips et al. 2017). We observed that the major complications associated with severe falciparum malaria include cerebral malaria, severe anemia, acidosis, and multiple organ failure. However, most of the complications observed in cases with *falciparum* malaria were similar to those seen in severe

vivax malaria except renal failure, splenic rupture, and hepatic dysfunction along with gastrointestinal symptoms (Coban, Lee, and Ishii 2018). Earlier we have reported five unique parasite proteins in *vivax* malaria patients named as three plasmodium exported proteins (PVX_003545, PVX_003555 and PVX_121935), a hypothetical protein (PVX_083555) and Pvstp1 (subtelomeric transmembrane protein 1, PVX_094303) from plasma and parasite isolates (Venkatesh et al. 2020).

Parasite proteins that are consistently detected in almost all the severe malaria patients would be highly promising for developing new diagnostic approaches. Moreover, those could be an indicator of severe infection as those are specifically detected in severe malaria patients. However, our findings need to be validated in larger clinical cohorts before the evaluation of their definite translational potentiality. The development of rapid and cost-effective assays for selective detection of these proteins in blood samples will also be required for their translational applications. These could be an effective future continuance of the present investigation.

Changes has been incorporated on following Page no.: 15

Comment 30: Finally, the last statement in line 433 p.19, ‘opportunities for developing clinical tests’, it would be beneficial to add the specific proteins targeted in such tests (refer to previous comment about machine learning model section).

Response: We appreciate this suggestion. We would like to clarify that some of the host proteins such as AZGP1, CRP, and FGB (adjusted p value <0.05) in *falciparum* malaria and APOA1 and CP (adjusted p value <0.05) in *vivax* malaria were found to be associated with the malaria severity and complexity, and could be studied further to evaluate their potentially and efficacy in predicting severe malaria.

Moreover, we quantified quite a few parasite proteins (*Plasmodium falciparum*) in the plasma samples of the severe *falciparum* malaria patients. Most of these parasite proteins we detected in a substantial proportion of the SFM patients consistently. Such as heat shock protein 90, enolase, actin I, heat shock 70 kDa protein, fructose-bisphosphate aldolase, and serine repeat antigen 4 were detected over 80% of the patient population (Table 1B). Parasite proteins that are consistently detected in almost all the severe malaria patients could be an indicator of severe infection as those are specifically detected in severe malaria patients.

However, our findings need to be validated in larger clinical cohorts before the evaluation of their definite translational potentiality. The development of rapid and cost-effective assays for selective detection of these proteins in blood samples will also be required for their translational applications (for developing clinical tests). These could be an effective future continuance of the present investigation.

Revision made: We have specified the candidate proteins (both host and parasite factors) that are promising for monitoring malaria severity and complexity in the revised manuscript.

Changes has been incorporated on following Page no.: 15

Response to the Comments of Reviewer # 3

The manuscript by Kumar et al. provides a detailed quantitative proteomics study in plasma of malaria patients with different complications, and also compares malaria caused by *Plasmodium falciparum* and *P. vivax* among them. Of particular value is the choice of controls for complications or disease that can be confused with clinical malaria symptoms, such as Dengue virus infection and meningitis.

The manuscript can be considered as providing a wealth of new information on the response to malaria in humans and does so with a complex and technologically advanced experimental approach.

Nevertheless, there are several points that may require the attention of the authors to provide a more smooth reading for the interested readers.

We are most grateful to this reviewer for an in-depth evaluation of our manuscript. Importantly the reviewer feels that “The manuscript can be considered as providing a wealth of new information on the response to malaria in humans and does so with a complex and technologically advanced experimental approach.” We highly appreciate these encouraging comments and constructive suggestions for improvement of our manuscript. Detailed responses to the comments and queries are given below.

Comment 1: In the results sections where experimental groups analyzed are compared not always is easy to follow the rationale behind the research. In particular, the groups, abbreviations and number of samples defined in several places (e.g. lines 107-108; lines 440-444, and along the results section) are not always coincidental. There is also some disagreements between the numbers of samples for a given group in the text and in the Figures (e.g. in the text is mentioned the existence of 58 samples in the HC group but apparently only 10 are shown in Figures 3d or 4 in Figure 4). Thus I suggest to include a table with 3 columns indicating each clinical group defined, the abbreviation used and the number of samples at each group. This will make easy to follow also several of the main Figures.

Response: We highly appreciate this suggestion and admit that the number of samples analyzed in different experimental groups was not mentioned clearly in our manuscript. In particular, this ambiguity happened as different numbers of samples were analyzed for different experimental groups in the discovery and validation phases of our study. We have resolved this issue in the revised manuscript.

Revision made: According to the suggestion of this reviewer, we have included an additional supplementary table to clarify the numbers of samples that were analyzed for different experimental groups (along with their abbreviations) in the analysis of clinical parameters, and in the discovery and validation phase quantitative proteomics analyses.

Changes has been incorporated on following Page no.: 06

Supplementary Table 1: Details of the sample numbers for the discovery phase quantitative proteomics, targeted validation, and clinicopathological analyses.

Experimental groups	Discovery phase quantitative proteomics (TMT-based multiplexed quantitation)	Targeted validation (MRM assay)	Analysis of clinical parameters
Falciparum malaria (FM)			
Non-severe Falciparum malaria (NSFM)	8	10 (DP = 05, NS = 05)	15
SFM_others	12	7 (DP = 04, NS = 03)	26
Severe anemia malaria (SA)	6	4 (DP = 04, NS = 00)	5
Cerebral malaria (CB)	6	3 (DP = 01, NS = 02)	7
Negative control: Control for severe anemia (CSA)	4	3 (DP = 03, NS = 00)	*
Negative control: Control for cerebral malaria (CCB)	4	3 (DP = 03, NS = 00)	*
Healthy Control (HC)	10	24 (DP = 18, NS = 06)	58
Disease Control: Non-severe dengue (NSD)	4	10 (DP = 04, NS= 06)	15
Disease Control: Severe dengue (SD)	12	17 (DP = 17, NS= 00)	
Vivax malaria (VM)			
Non-severe vivax malaria (NSVM)	12	10 (DP = 07, NS= 03)	17
Severe vivax malaria (SVM)	20	20 (DP = 20, NS = 00)	
Healthy Control (HC)	04	24 (DP = 18, NS = 06)	58
Disease Control: Non-severe dengue (NSD)	4	10 (DP = 04, NS= 06)	15
Disease Control: Severe dengue (SD)	12	17 (DP = 17, NS= 00)	

SFM_Others: It includes case of severe falciparum malaria patients excluding CB and SA complications.

*not considered for clinical analysis.

DP – subjects from discovery phase, NS – new subjects.

Comment 2: Can the authors provide information about parasitemia levels and results of RTD in parallel in all samples? How can the authors be sure that healthy controls (or other disease controls used) did not have subclinical plasmodium parasitemia if RDT systems do not test positive below 100 parasites per ul? PCR for *P. falciparum* and *P. vivax* was performed in all control samples? What it's the sensitivity of their RDTs?

Response: In this study, diagnosis of malaria was carried out primarily by microscopic examination of thin peripheral blood smears followed by rapid diagnostic tests (RDTs,

FalciVax, Zephyr Biomedicals). Although microscopy was performed involving highly experience personnel for detection of the parasites, parasitemia count in individual samples was not reordered (in particular, due to the paucity of time in handling the extreme burden of malaria patents in the tertiary care hospitals in a highly populated country like India). Additionally, in a randomized fashion some samples (malaria and control) were confirmed using PCR-based molecular diagnosis.

The controls selected in this study were entirely devoid of any clinical symptoms of malaria. However, we concur with the reviewer that there might be some possibilities of asymptomatic infection. In fact, that is a concern or challenge for any case-control studies as subjects with asymptomatic infections may remain in our population. We accept that it is a possible limitation of the study. Nevertheless, as the RDT method used in this study was reasonably sensitive, it almost excludes the possibilities of asymptomatic infections in the control populations.

The sensitivity of the FALCIVAX RDT kit used in this study is 94.7% in comparison to the gold standard methods (such as microscopy) even at a low parasitaemia (80 parasites/ μ l) in case of *P. falciparum* malaria (Falcivax®, Zephyr Biomedical Systems, India, Page 715, - <http://www.tulipgroup.com/ProductEvaluations/Falcivax.pdf>). In the case of the detection of *P. vivax*, it also provides a sensitivity of more than 90% at a low parasitaemia level. (Sreekanth, Shenoy, and Sai Lella 2011). Therefore, we consider that the possibility of obtaining a false-negative observation in the control populations is extremely low.

Comment 3: Line 84: “dark matter” is used as a metaphor here but it would be convenient to define exactly what the authors wish to mean with it.

Response: We thank the reviewer for this suggestion. The concerned statement is clarified in revised manuscript.

Comment 4: Line 120: what is the meaning of “subclinical” in this line?

Response: We would like to thank the reviewer for this query. We observed that the liver function parameters (such as SGPT, SGOT, ALP, etc.) were slightly higher in the non-severe malaria patients as compared to the healthy controls. As we know that abnormal liver enzyme levels may signal liver damage or alteration in bile flow. Here, using the word 'subclinical' we mean to say that the altered levels of the liver function parameters were very mild (clinically insignificant) and not necessarily a clear indication of liver damage, which is generally defined by a very high level of increase (5 times or more) in these liver function parameters. Of note, liver enzymes in healthy subjects sometimes may also show a slightly higher level than the normal range due to several non-clinical physiological reasons (in absence of any clinical features).

Revision made: We have removed the phrase “sub-clinical” in the revised manuscript to avoid any confusion.

Comment 5: Diagnosis of Dengue should be clarified since it can be distinguished two different clinical stages and IgG / IgM markers for them. Is SD haemorrhagic fever?

Response: We would like to thank the reviewer for these query and suggestions.

We would like to clarify that Severe Dengue (SD) is dengue hemorrhagic fever. According to the new (2009) classification of dengue by WHO (World Health Organization. Geneva, Switzerland: WHO; 2009. Dengue: Guidelines for Diagnosis, Treatment, Prevention, and Control), there are two categories - 1. Dengue (D), and 2. Severe dengue (SD). This replaced the earlier (1997) case classification i.e. dengue fever (DF), dengue hemorrhagic fever (DHF), and dengue shock syndrome (DSS). In our study, IgM was used as the confirmatory marker for Dengue infection. If NS1 was found to be positive, we retested for IgM during the 5-7 days of fever to confirm the diagnosis. According to the recent WHO dengue classification, we classified dengue patients based on clinical features (BP, bleeding manifestation, etc.) and hematological parameters (hematocrit, platelet counts, organ failure, etc.).

Revision made: We have included the details for diagnosis of dengue patients in the revised manuscript.

Changes has been incorporated on following Page no.: 21

Comment 6: Why elastic net regularized logistic regression method was chosen?

Response: We would like to thank the reviewer for this imperative query.

The elastic net regularized logistic regression model was chosen because it offers more flexibility in two ways - (i) the l_1 norm helps attain parsimony in the sense that it optimally chooses the number of covariates (in the context of present data set the covariates are proteins) by driving coefficients of unimportant covariates to zero, and (ii) l_2 norm helps address the issue of multicollinearity (De Mol, De Vito, and Rosasco 2009; Parvande et al. 2020; Sirimongkolkasem and Drikvandi 2019; Witten, Shojaie, and Zhang 2014; Zhu et al. 2020; Zou and Hastie 2005).

Comment 7: As the manuscript mostly describes the plasma effect of the host response to malaria, there are human genetic polymorphisms of malaria tolerance that could be response modifiers. Are there data on polymorphisms in haemoglobin or G6PD in the samples used?

Response: We are thankful to the reviewer for this insightful remark. However, we would like to clarify that the genetic polymorphisms such as haemoglobin and G6PD were not evaluated in our study. The frequency of G6PD is only 8.5% in the Indian population (Lauden et al. 2019). Moreover, G6PD deficiency is an X-linked recessive trait in nature. Hence, out of this 8.5%, only $1/4^{\text{th}}$ population might have a non-functional G6PD gene (Howes et al. 2016). Overall, the possibility of obtaining Class I G6PD deficiency condition is very rare (Gómez-Manzo et al. 2016). Additionally, India being a developing country, majority of the patients enrolled in this study cannot afford the extra charges associated with each added test and rather opted out for these additional tests.

Comment 8: Line 149: is any of the 31 plasma proteins altered in NSFm in coincidence with the 38 found in SFM? Please detail findings to this respect.

Response: There are 16 common proteins that are altered both in NSFm and SFM.

Revision made: According to the suggestion from this reviewer we have shown this comparative analysis as a Venn diagram in the revised manuscript. We have also provided the details for the commonly altered proteins in NSFm and SFM in the revised manuscript (Supplementary Table 3).

Supplementary Table 3: List of significantly altered common host proteins in SFM and NSFm as compared to healthy control from plasma samples.

Fig: The Venn diagram represents the common and unique proteins (differential abundance compared to HC) between non-severe and severe falciparum malaria.

Changes has been incorporated on following Page no.: 8

Comment 9: Line 161: These two controls are not clearly defined in terms of clinical and diagnostic characterization. How many of each?

Response: We would like to thank the reviewer for this query. We have used anemic patients [control for severe anemia (CSA)] without any malaria symptoms as a control for severe malaria anemia. In a similar way, we have used meningitis patients [control for cerebral malaria (CCB)] as a control for cerebral malaria.

Revision made: We have provided the clinical and diagnostic details for these CSA and CBB patients in the revised manuscript.

The negative control for severe anemia i.e. CSA (anemic patients without malaria) patients were defined by hemoglobin < 7gm/dl and negative control for cerebral malaria i.e. CCB (meningitis patients without malaria) were defined as clinically altered sensorium, neck rigidity, meningeal enhancement on MRI brain and abnormal CSF study.

Changes has been incorporated on following Page no.: 20

Comment 10: Supplementary Figure 5: what are the common proteins within the SFM group between CB and SA subgroups? If any, this should be mentioned.

Response: We would like to thank the reviewer for this query. We have modified the Supplementary Figure 5. It represents the common proteins that are significantly altered ($p < 0.05$) in all the sub-groups under SFM i.e. CB and SA.

Revision made: We have included a supplementary table 4 to display this comparative analysis more clearly.

Changes has been incorporated on following Page no.: 9

Supplementary Fig. 5b Unsupervised clustering of significantly altered proteins ($p < 0.05$) in falciparum malaria. Heat map of individual samples for cerebral malaria (CB) and severe malaria anemia (SA).

Comment 11: Do the authors consider that changes in protein expression in the range +1 to -0.5 are sufficiently appropriate to draw mechanistic conclusions?

Response: We highly appreciate this remark from the reviewer. We totally agree with the reviewer that the proteins showing a higher fold-change in differential abundance increase the confidence for drawing mechanistic conclusions. We wish to clarify that many of the differentially abundant plasma proteins identified in our study have a higher level of fold-change (>1.5) (Table 1B).

Here we used a multiplexed tandem mass tag (TMT)-based quantitative approach for obtaining excellent reproducibility and better statistics (Hurley et al. 2018; Rey et al. 2018; Wang et al. 2019). Multiplexing using stable isotope or labelling (TMT) results in increased throughput, higher precision, better reproducibility, reduced technical variations and very

fewer missing values than other methods (Erickson et al. 2015; Mertins et al. 2018; O'Connell et al. 2018). However, it is established that TMT (or iTRAQ)-based measurement provides better statistics and higher precision at the cost of lower dynamic range, largely because isolation interference (or reporter ion interference) and ratio compression reduce the fold-change magnitude in multiplexed TMT measurement (Hogrebe et al. 2018; O'Connell et al. 2018). Therefore, a lower amplitude/magnitude (compared to label-free quantitation) is always to be expected for TMT-based quantification.

Regarding the selection of the fold-change and *p-value* thresholds for differentially abundant proteins in our multiplexed label-based (TMT) quantitative proteomics analysis, we wish to clarify that fold-change ≥ 1.25 and $p < 0.05$ are well-accepted standards; particularly when only the peptides uniquely identifying individual proteins are utilized for calculation of the fold-change values. According to the relevant published literature, it is apparent that fold-change values 1.5 (or even lower) measured by any highly-sensitive mass spectrometer are quite reliable. To this end, we would like to mention that there are several published articles where very low fold-change thresholds (1.5 or even lower) have been used in screening proteome level alterations in serum/plasma samples (Gollapalli et al. 2012; Kassa et al. 2011; Kumar et al. 2012; Liu et al. 2015; Peng et al. 2013; Song et al. 2014; Wan et al. 2006; Zhang et al. 2015). Consequently, we are afraid that the implementation of a very high fold-change cut-off would eventually lead to an exclusion of several imperative candidates in the discovery phase of the analysis itself.

Comment 12: Line 254. How many of the proteins found significantly as NSVM markers also appear in NSF? This would be interesting in case they have similar response pathways.

Response: We would like to clarify that we identified 8 proteins that showed alerted plasma abundances in both NSF and NSVM i.e. the non-severe malaria caused by *P. falciparum* and *P. vivax*, respectively.

These proteins include fibrinogen alpha chain, leucine-rich alpha-2-glycoprotein, C-reactive protein, alpha-1-acid glycoprotein 1, cDNA FLJ55673, highly similar to Complement Factor B (EC 3.4.21.47), fibrinogen beta chain, apolipoprotein A-II, and histidine-rich glycoprotein

Of note, we observed that several vital physiological pathways/biological processes such as platelet degranulation, response to elevated platelet cytosolic Ca²⁺ and platelet activation, signaling and aggregation pathways are associated with these commonly altered plasma proteins identified in NSF and NSVM patients.

Revision made: Considering the suggestion given by this reviewer, we have provided details for all the proteins in the revised manuscript (Supplementary Table 14).

Revision made: Several vital physiological pathways/biological processes such as platelet degranulation, response to elevated platelet cytosolic Ca²⁺ and platelet activation, signaling and aggregation pathways are associated with 8 commonly altered plasma proteins identified in NSF and NSVM patients. These proteins include fibrinogen alpha chain, leucine-rich alpha-2-glycoprotein, C-reactive protein, alpha-1-acid glycoprotein 1, cDNA FLJ55673, highly similar to Complement Factor B (EC 3.4.21.47), fibrinogen beta chain, apolipoprotein A-II, and histidine-rich glycoprotein.

a

ID	Name	#Gene	FDR
R-HSA-114608	Platelet degranulation	4	3.13e-04
R-HSA-76005	Response to elevated platelet cytosolic Ca2+	4	3.13e-04
R-HSA-76002	Platelet activation, signaling and aggregation	4	4.11e-03
R-HSA-354194	GRB2:SOS provides linkage to MAPK signaling for Integrins	2	9.75e-03
R-HSA-372708	p130Cas linkage to MAPK signaling for integrins	2	9.75e-03
R-HSA-5686938	Regulation of TLR by endogenous ligand	2	1.32e-02
R-HSA-140875	Common Pathway of Fibrin Clot Formation	2	1.53e-02
R-HSA-354192	Integrin alphaIIb beta3 signaling	2	2.03e-02
R-HSA-168249	Innate Immune System	5	2.25e-02
R-HSA-6802948	Signaling by high-kinase activity BRAF mutants	2	2.25e-02

b

ID:R-HSA-114608 Name:Platelet degranulation			
C=126; O=4; E=0.07; R=53.62; PValue=3.35e-07; FDR=3.13e-04			
userid	Gene Symbol	Gene Name	Entrez Gene
P02671	FGA	fibrinogen alpha chain	2243
P02675	FGB	fibrinogen beta chain	2244
P04196	HRG	histidine rich glycoprotein	3273
P02763	ORM1	orosomuroid 1	5004
ID:R-HSA-76005 Name:Response to elevated platelet cytosolic Ca2+			
C=131; O=4; E=0.08; R=51.57; PValue=3.92e-07; FDR=3.13e-04			
userid	Gene Symbol	Gene Name	Entrez Gene
P02671	FGA	fibrinogen alpha chain	2243
P02675	FGB	fibrinogen beta chain	2244
P04196	HRG	histidine rich glycoprotein	3273
P02763	ORM1	orosomuroid 1	5004
ID:R-HSA-76002 Name:Platelet activation, signaling and aggregation			
C=276; O=4; E=0.16; R=24.48; PValue=7.74e-06; FDR=4.11e-03			
userid	Gene Symbol	Gene Name	Entrez Gene
P02671	FGA	fibrinogen alpha chain	2243
P02675	FGB	fibrinogen beta chain	2244
P04196	HRG	histidine rich glycoprotein	3273
P02763	ORM1	orosomuroid 1	5004

Fig: Physiological pathways enriched for the plasma proteins that are significantly altered in both NSVM and NSF. **a.** 10 common pathways associated with the plasma proteins that are significantly altered in both NSVM and NSF (represented in descending order of the FDR value) **b.** top three pathways with minimum FDR and list of the mapped proteins to Platelet degranulation, Response to elevated platelet cytosolic Ca²⁺ and Platelet activation, signaling and aggregation pathways, respectively.

Changes has been incorporated on following Page no.: 11

Comment 13: Lines 267-274: how consistent were the presence of this proteins in all samples? Can they be used for developing new diagnostic procedures?

Response: This is a very important query from the reviewer. We would like to clarify that we quantified quite a few parasite proteins (*Plasmodium falciparum*) in the plasma samples of the severe falciparum malaria patients. Of note, most of these parasite proteins we detected in a substantial proportion of the SFM patients consistently (please see Table 1B). Six out of ten parasite proteins were found consistently (> 80%) present in malaria patients. These parasite proteins include heat shock protein 90, enolase, actin I, heat shock 70 kDa protein, fructose-bisphosphate aldolase, and serine repeat antigen 4. (Table 1B). However, these parasite proteins were not detected in the non-severe falciparum malaria patients i.e. the patients with much lower levels of parasitemia even after profiling several times. A possible reason behind this could be that the parasite proteins secreted in the plasma samples of the NSF patients were extremely low and below the sensitivity level of the LFQ approach even using the ultrasensitive next generation orbitrap mass spectrometers.

Parasite proteins that are consistently detected in almost all the severe malaria patients would be highly promising for developing new diagnostic approaches. Moreover, those could be an indicator of severe infection as those are specifically detected in the plasma samples of the severe malaria patients. However, our findings need to be validated in larger clinical cohorts before the evaluation of their definite translational potentiality. The development of rapid and cost-effective assays for selective detection of these proteins in blood samples will also be required for their translational applications. These could be an effective future continuance of the present investigation.

Revision made: We have included a discussion on the parasite proteins that are detected in the plasma samples of the severe falciparum malaria patients and their possible translational aspects in the revised manuscript.

Changes has been incorporated on following Page no.: 12

Comment 14: Several of the up-regulated proteins found in severe malaria (cerebral malaria) have been previously found upregulated in murine models of malaria thus reinforcing the authors' findings. Providing these previous findings as reinforcement of the authors' own findings would give more strength to their discussion and to the mechanistic meaning of the markers discovered.

Response: We highly appreciate this constructive and insightful advice from the reviewer. Indeed, some of the up-regulated proteins in severe malaria (cerebral malaria) identified in our study were previously reported in the murine malaria model, which substantially enhances the strength of our findings.

Revision made: Following the suggestion given by this reviewer we have included discussion on this aspect in the revised manuscript.

Following changes are incorporated in the text:

Upregulated proteins in cerebral malaria identified in our study were previously reported in the murine malaria model (Bauer et al. 2002; Schofield and Grau 2005), which substantially enhances the strength of our findings. Bauer et al. reported the upregulation of ICAM-1, P-selectin, and VCAM-1 on brain vascular endothelium in *P. berghei* ANKA infection. Upregulated ICAM-1 levels may help the parasite sequestration on the epithelial cells causing brain injury and creating hypoxia conditions (Favre et al.). Blocking of the ICAM1 receptor caused a more than 150% increase of schizonts in the peripheral blood in mice. VCAM-1 plays an important role in the resetting of the parasite to the blood vessels. However, it is limited to large blood vessels unlike ICAM-1 (Chakravorty and Craig 2005).

Sl. no.	Proteins	Our data (Fold change)	Murine model study
1	ICAM-1	1.51 (CB vs. HC) 1.21 (CB vs. NSFAM)	Upregulated (Hearn et al. 2000)
2	VCAM-1	1.24 (CB vs. HC) 1.17 (CB vs. NSFAM)	Upregulated (Ma et al. 1996)

Changes has been incorporated on following Page no.: 17

Comment 15: Although there are easily distinguishable abbreviations, they are not always defined in the text. It is suggested that all of them are defined (e.g. at line 121).

Response: We appreciate the reviewer for reading our manuscript thoroughly. All the abbreviations are defined at their first usage in the revised manuscript.

References:

Anderson, N. Leigh, and Norman G. Anderson. 2002. "The Human Plasma Proteome: History, Character, and Diagnostic Prospects." *Molecular & cellular proteomics: MCP* 1(11): 845–67.

Bachmann, Julie et al. 2014. "Affinity Proteomics Reveals Elevated Muscle Proteins in Plasma of Children with Cerebral Malaria." *PLoS pathogens* 10(4): e1004038.

- Bauer, Phillippe R. et al. 2002. "Regulation of Endothelial Cell Adhesion Molecule Expression in an Experimental Model of Cerebral Malaria." *Microcirculation (New York, N.Y.: 1994)* 9(6): 463–70.
- Chakravorty, Srabasti J., and Alister Craig. 2005. "The Role of ICAM-1 in Plasmodium Falciparum Cytoadherence." *European Journal of Cell Biology* 84(1): 15–27.
- Coban, Cevayir, Michelle Sue Jann Lee, and Ken J. Ishii. 2018. "Tissue-Specific Immunopathology during Malaria Infection." *Nature Reviews. Immunology* 18(4): 266–78.
- De Mol, Christine, Ernesto De Vito, and Lorenzo Rosasco. 2009. "Elastic-Net Regularization in Learning Theory." *Journal of Complexity* 25(2): 201–30.
- Erickson, Brian K. et al. 2015. "Evaluating Multiplexed Quantitative Phosphopeptide Analysis on a Hybrid Quadrupole Mass Filter/Linear Ion Trap/Orbitrap Mass Spectrometer." *Analytical Chemistry* 87(2): 1241–49.
- Farrah, Terry et al. 2012. "PASSEL: The PeptideAtlas SRMexperiment Library." *Proteomics* 12(8): 1170–75.
- Francischetti, Ivo M. B., Karl B. Seydel, and Robson Q. Monteiro. 2008. "Blood Coagulation, Inflammation, and Malaria." *Microcirculation (New York, N.Y.: 1994)* 15(2): 81–107.
- Geyer, Philipp E., Lesca M. Holdt, Daniel Teupser, and Matthias Mann. 2017. "Revisiting Biomarker Discovery by Plasma Proteomics." *Molecular Systems Biology* 13(9): 942.
- Gollapalli, Kishore et al. 2012. "Investigation of Serum Proteome Alterations in Human Glioblastoma Multiforme." *Proteomics* 12(14): 2378–90.
- Gómez-Manzo, Saúl et al. 2016. "Glucose-6-Phosphate Dehydrogenase: Update and Analysis of New Mutations around the World." *International Journal of Molecular Sciences* 17(12). <https://www.ncbi.nlm.nih.gov/pmc/articles/PMC5187869/> (June 11, 2020).
- Hearn, Jocelyn et al. 2000. "Immunopathology of Cerebral Malaria: Morphological Evidence of Parasite Sequestration in Murine Brain Microvasculature." *Infection and Immunity* 68(9): 5364–76.
- Hogrebe, Alexander et al. 2018. "Benchmarking Common Quantification Strategies for Large-Scale Phosphoproteomics." *Nature Communications* 9(1): 1045.
- Howes, Rosalind E. et al. 2016. "Global Epidemiology of Plasmodium Vivax." *The American Journal of Tropical Medicine and Hygiene* 95(6 Suppl): 15–34.
- Huang, Ting, Meena Choi, Sicheng Hao, and Olga Vitek. 2019. *MSstatsTMT: Protein Significance Analysis in Shotgun Mass Spectrometry-Based Proteomic Experiments with Tandem Mass Tag (TMT) Labeling*. Bioconductor version: Release (3.9). <https://bioconductor.org/packages/MSstatsTMT/> (October 3, 2019).
- Hurley, Jennifer M. et al. 2018. "Circadian Proteomic Analysis Uncovers Mechanisms of Post-Transcriptional Regulation in Metabolic Pathways." *Cell Systems* 7(6): 613–626.e5.
- Kassa, Fikregabrail Abera et al. 2011. "New Inflammation-Related Biomarkers during Malaria Infection" ed. Gordon Langsley. *PLoS ONE* 6(10): e26495.
- Keshishian, Hasmik et al. 2017. "Quantitative, Multiplexed Workflow for Deep Analysis of Human Blood Plasma and Biomarker Discovery by Mass Spectrometry." *Nature Protocols* 12(8): 1683–1701.
- Kumar, Yadunanda et al. 2012. "Serum Proteome and Cytokine Analysis in a Longitudinal Cohort of Adults with Primary Dengue Infection Reveals Predictive Markers of DHF." *PLoS neglected tropical diseases* 6(11): e1887.

- Lauden, Stephanie M. et al. 2019. "Prevalence of Glucose-6-Phosphate Dehydrogenase Deficiency in Cameroonian Blood Donors." *BMC research notes* 12(1): 195.
- Li, Jiaming et al. 2020. "TMTpro Reagents: A Set of Isobaric Labeling Mass Tags Enables Simultaneous Proteome-Wide Measurements across 16 Samples." *Nature Methods* 17(4): 399–404.
- Liu, Jiyan et al. 2015. "Comparative Proteomic Analysis of Serum Diagnosis Patterns of Sputum Smear-Positive Pulmonary Tuberculosis Based on Magnetic Bead Separation and Mass Spectrometry Analysis." *International Journal of Clinical and Experimental Medicine* 8(2): 2077–85.
- Ma, N., N. H. Hunt, M. C. Madigan, and T. Chan-Ling. 1996. "Correlation between Enhanced Vascular Permeability, up-Regulation of Cellular Adhesion Molecules and Monocyte Adhesion to the Endothelium in the Retina during the Development of Fatal Murine Cerebral Malaria." *The American Journal of Pathology* 149(5): 1745–62.
- Mertins, Philipp et al. 2018. "Reproducible Workflow for Multiplexed Deep-Scale Proteome and Phosphoproteome Analysis of Tumor Tissues by Liquid Chromatography-Mass Spectrometry." *Nature Protocols* 13(7): 1632–61.
- Miller, Susanne K. et al. 2002. "A Subset of Plasmodium Falciparum SERA Genes Are Expressed and Appear to Play an Important Role in the Erythrocytic Cycle." *Journal of Biological Chemistry* 277(49): 47524–32.
- Moussa, Ehab M. et al. 2018. "Proteomic Profiling of the Plasma of Gambian Children with Cerebral Malaria." *Malaria Journal* 17(1): 337.
- Nemetski, Sondra Maureen et al. 2015. "Inhibition by Stabilization: Targeting the Plasmodium Falciparum Aldolase–TRAP Complex." *Malaria Journal* 14(1): 324.
- Ockenhouse, C. F. et al. 1992. "Human Vascular Endothelial Cell Adhesion Receptors for Plasmodium Falciparum-Infected Erythrocytes: Roles for Endothelial Leukocyte Adhesion Molecule 1 and Vascular Cell Adhesion Molecule 1." *The Journal of Experimental Medicine* 176(4): 1183–89.
- O'Connell, Jeremy D., Joao A. Paulo, Jonathon J. O'Brien, and Steven P. Gygi. 2018. "Proteome-Wide Evaluation of Two Common Protein Quantification Methods." *Journal of Proteome Research* 17(5): 1934–42.
- Olszewski, Kellen L. et al. 2009. "Host-Parasite Interactions Revealed by Plasmodium Falciparum Metabolomics." *Cell Host & Microbe* 5(2): 191–99.
- Parvande, Saeid, Hung-Wen Yeh, Martin P. Paulus, and Brett A. McKinney. 2020. "Consensus Features Nested Cross-Validation." *Bioinformatics (Oxford, England)* 36(10): 3093–98.
- Peng, Liang et al. 2013. "Serum Proteomics Analysis and Comparisons Using ITRAQ in the Progression of Hepatitis B." *Experimental and Therapeutic Medicine* 6(5): 1169–76.
- Perez-Riverol, Yasset et al. 2016. "PRIDE Inspector Toolsuite: Moving Toward a Universal Visualization Tool for Proteomics Data Standard Formats and Quality Assessment of ProteomeXchange Datasets." *Molecular & cellular proteomics: MCP* 15(1): 305–17.
- Perkins, Douglas J. et al. 2011. "Severe Malarial Anemia: Innate Immunity and Pathogenesis." *International Journal of Biological Sciences* 7(9): 1427–42.
- Phillips, Margaret A. et al. 2017. "Malaria." *Nature Reviews. Disease Primers* 3: 17050.
- Qi, Lihong et al. 2018. "Strategies for Imputing Missing Covariates in Accelerated Failure Time Models." *Statistics in Medicine* 37(24): 3417–36.
- Ray, Sandipan et al. 2011. "Proteomic Technologies for the Identification of Disease Biomarkers in Serum: Advances and Challenges Ahead." *Proteomics* 11(11): 2139–61.

- Ray, Sandipan et al. 2012. "Proteomic Investigation of Falciparum and Vivax Malaria for Identification of Surrogate Protein Markers." *PloS One* 7(8): e41751.
- Ray, Sandipan et al. 2015. "Proteomic Analysis of Plasmodium Falciparum Induced Alterations in Humans from Different Endemic Regions of India to Decipher Malaria Pathogenesis and Identify Surrogate Markers of Severity." *Journal of Proteomics* 127(Pt A): 103–13.
- Ray, Sandipan et al. 2016. "Clinicopathological Analysis and Multipronged Quantitative Proteomics Reveal Oxidative Stress and Cytoskeletal Proteins as Possible Markers for Severe Vivax Malaria." *Scientific Reports* 6: 24557.
- Ray, Sandipan et al. 2019. "Phenotypic Proteomic Profiling Identifies a Landscape of Targets for Circadian Clock-Modulating Compounds." *Life Science Alliance* 2(6).
- Reuterswärd, Philippa et al. 2018. "Levels of Human Proteins in Plasma Associated with Acute Paediatric Malaria." *Malaria Journal* 17(1): 426.
- Rey, Guillaume et al. 2018. "Metabolic Oscillations on the Circadian Time Scale in Drosophila Cells Lacking Clock Genes." *Molecular Systems Biology* 14(8): e8376.
- Rhenman, A. et al. 2015. "Which Set of Embryo Variables Is Most Predictive for Live Birth? A Prospective Study in 6252 Single Embryo Transfers to Construct an Embryo Score for the Ranking and Selection of Embryos." *Human Reproduction (Oxford, England)* 30(1): 28–36.
- Schofield, Louis, and Georges E. Grau. 2005. "Immunological Processes in Malaria Pathogenesis." *Nature Reviews. Immunology* 5(9): 722–35.
- Sharma, Shweta et al. 2014. "A Secretory Multifunctional Serine Protease, DegP of Plasmodium Falciparum, Plays an Important Role in Thermo-Oxidative Stress, Parasite Growth and Development." *The FEBS journal* 281(6): 1679–99.
- Sirimongkolkasem, Tanin, and Reza Drikvandi. 2019. "On Regularisation Methods for Analysis of High Dimensional Data." *Annals of Data Science* 6(4): 737–63.
- Somner, Elizabeth A, Julie Black, and Geoffrey Pasvol. 2000. "Multiple Human Serum Components Act as Bridging Molecules in Rosette Formation by Plasmodium Falciparum-Infected Erythrocytes." 95(2): 10.
- Song, Sang Hoon et al. 2014. "Proteomic Profiling of Serum from Patients with Tuberculosis." *Annals of Laboratory Medicine* 34(5): 345–53.
- Sreekanth, B., Shalini Shenoy, and K. Sai Lella. 2011. "Evaluation of Blood Smears, Quantitative Buffy Coat and Rapid Diagnostic Tests in the Diagnosis of Malaria." *Journal of Bacteriology & Parasitology* 02(08). <https://www.omicsonline.org/evaluation-of-blood-smears-quantitative-buffy-coat-and-rapid-diagnostic-tests-in-the-diagnosis-of-malaria-2155-9597.1000125.php?aid=3265> (June 20, 2020).
- Su, X. Z., and T. E. Wellems. 1994. "Sequence, Transcript Characterization and Polymorphisms of a Plasmodium Falciparum Gene Belonging to the Heat-Shock Protein (HSP) 90 Family." *Gene* 151(1–2): 225–30.
- Vabalas, Andrius, Emma Gowen, Ellen Poliakoff, and Alexander J. Casson. 2019. "Machine Learning Algorithm Validation with a Limited Sample Size." *PloS One* 14(11): e0224365.
- Venkatesh, Apoorva et al. 2020. "Comprehensive Proteomics Investigation of P. Vivax-Infected Human Plasma and Parasite Isolates." *BMC infectious diseases* 20(1): 188.
- Wan, Jia et al. 2006. "Inflammation Inhibitors Were Remarkably Up-Regulated in Plasma of Severe Acute Respiratory Syndrome Patients at Progressive Phase." *Proteomics* 6(9): 2886–94.

- Wang, Dandan et al. 2019. "ELF4 Facilitates Innate Host Defenses against *Plasmodium* by Activating Transcription of *Pf4* and *Ppbp*." *Journal of Biological Chemistry* 294(19): 7787–96.
- WHO. 2000. "Severe Falciparum Malaria. World Health Organization, Communicable Diseases Cluster." *Transactions of the Royal Society of Tropical Medicine and Hygiene* 94 Suppl 1: S1-90.
- Wilson, Nana O. et al. 2011. "CXCL4 and CXCL10 Predict Risk of Fatal Cerebral Malaria." *Disease Markers* 30(1): 39–49.
- Witten, Daniela M., Ali Shojaie, and Fan Zhang. 2014. "The Cluster Elastic Net for High-Dimensional Regression With Unknown Variable Grouping." *Technometrics: A Journal of Statistics for the Physical, Chemical, and Engineering Sciences* 56(1): 112–22.
- Zhang, Yushi, Yi Cai, Hongyan Yu, and Hanzhong Li. 2015. "ITRAQ-Based Quantitative Proteomic Analysis Identified HSC71 as a Novel Serum Biomarker for Renal Cell Carcinoma." *BioMed Research International* 2015: 802153.
- Zhu, Xueru et al. 2020. "Predictive Model of the First Failure Pattern in Patients Receiving Definitive Chemoradiotherapy for Inoperable Locally Advanced Non-Small Cell Lung Cancer (LA-NSCLC)." *Radiation Oncology* 15(1): 43.
- Zou, Hui, and Trevor Hastie. 2005. "Regularization and Variable Selection via the Elastic Net." *Journal of the Royal Statistical Society: Series B (Statistical Methodology)* 67(2): 301–20.

REVIEWERS' COMMENTS:

Reviewer #2 (Remarks to the Author):

As reviewed previously, I believe that it is one of the most comprehensive studies so far as it includes comparisons between malaria severity states (already described in the literature) and between the two most prevalent species of Plasmodium (falciparum and vivax). It also includes parasite proteins and disease controls. There is a validation step, which often is missed in such studies. The study suffers from a low number of patients in some of the disease groups (which is limited by the nature of the disease and clinical setting), but the use of the validation process shows a robust analysis for the cohort in hand.

All comments previously raised have been answered favourably.

I still have 2 very minor comments about the amendments but this is very minor and should not reflect on the suitability of the present manuscript to the larger community.

Comment 4: Line 111 p.6: 'sequential decrease'. There is no sequence as there is no repeated samples over time of same patient.

Response: We appreciate this remark of the reviewer and admit that there is no repeated measurement performed from the same patients in a longitudinal manner. Here we mean to say that the magnitude of alterations (decrease) in platelet counts and Hb levels correlates with disease severity, i.e. the SFM patients had much lower levels of these hematological parameters compared to the NSFMs patients. Revision made: We have amended this statement as per the suggestion from this reviewer. Interestingly, the magnitude of alterations (decrease) in platelet counts and Hb levels were found to be more prominent in severe malaria, i.e. the SFM patients had lower levels of these hematological parameters compared to the NSFMs patients. Changes has been incorporated on following Page no.: 06

Additional comment:

I still believe the revision unclear.

'Interestingly, the magnitude of alterations (decrease) in platelet counts and Hb levels were found to be more prominent in severe malaria, i.e., the SFM patients had lower levels of these hematological parameters compared to the NSFMs patients.'

It is not all surprising that haematological features are worsen in severe malaria. In my mind, the simpler first section of the sentence is clear enough. 'The magnitude of alterations (decrease) in platelet counts and Hb levels were found to be more prominent in severe malaria'

Comment 24:

Supplementary table 8 did not have the corresponding protein ID included

Reviewer #3 (Remarks to the Author):

The new revised manuscript by Kumar et al. has improved substantially and with the corrections made and the answers given has provided a satisfactory response to almost all the points that were raised. However, my points 1, 2 and 7 are not entirely resolved and should be reconsidered.

I will now comment on these points and the answers obtained:

Comment 1: With the new supplementary table provided it has been improved the perception of the samples used. Nevertheless, it's not yet clear how many are coincidental between each phase (if any) and how many of them were used for the analysis of clinical parameters. A complete table of all individual samples and its particular analytical use would certainly help to clarify it. For example, how many of the 26 in SA group were analyzed at TMT and MRM assays? It's also unclear why the figures for Discovery Phase (DP) samples in HC and SD groups are higher in the TMT column than in the MRM column at DP.

Comment 2: This reviewer does not agree of not having this essential data of parasitemia levels at each analysed sample by PCR. If the authors do not have such information, then a caution notice should be provided in the results and discussion sections indicating that some possibilities of asymptomatic infection in the controls and other groups may remain in the analysed population and that it is a possible limitation of the study. Even high sensitivity RDTs can fail for many samples when from asymptomatic malaria and in many endemic areas asymptomatic malaria can range up to 50% with parasitaemia below 80 parasites per ul, and then many can pass as from healthy people.

Comment 7: At this point the authors have only answered to the G6PD deficiency but not to the haemoglobinopathies. It is true that although it is difficult to establish the percentages of polymorphisms associated with population variants of hemoglobins and G6PD in India because of the particular characteristics of the population in this country, it can be considered according to the available literature that the carrier frequency of haemoglobinopathies varies between 3 and 17% in different populations of India. The cumulative gene frequency of the three most predominant abnormal haemoglobins, i.e. sickle cell, haemoglobin D and haemoglobin E has been found to be 5-6% in India. On the other hand Beta-thalassemia trait is the predominant genetic Hb disorder accounting for >11% in some regions.

Having therefore accumulated percentages of these polymorphisms as a whole (G6PDs and haemoglobins) that can substantially modify the response to malaria and given that (as in the previous point) this lack of information can be a limitation of the study, I believe that the authors should be aware of this and mention a cautionary note in the discussion that the lack of genetic analysis of human samples allows the prediction of a certain percentage of polymorphisms associated with the response to malaria (G6PD and haemoglobins) that could bias part of the results obtained.

Multiplexed Quantitative Proteomics Provides Mechanistic Cues for Malaria Severity and Complexity

Response to the Comments of the Reviewer # 2

As reviewed previously, I believe that it is one of the most comprehensive studies so far as it includes comparisons between malaria severity states (already described in the literature) and between the two most prevalent species of Plasmodium (falciparum and vivax). It also includes parasite proteins and disease controls. There is a validation step, which often is missed in such studies. The study suffers from a low number of patients in some of the disease groups (which is limited by the nature of the disease and clinical setting), but the use of the validation process shows a robust analysis for the cohort in hand.

All comments previously raised have been answered favourably.

I still have 2 very minor comments about the amendments but this is very minor and should not reflect on the suitability of the present manuscript to the larger community.

Comment 4: Line 111 p.6: 'sequential decrease'. There is no sequence as there is no repeated samples over time of same patient.

Response: We appreciate this remark of the reviewer and admit that there is no repeated measurement performed from the same patients in a longitudinal manner. Here we mean to say that the magnitude of alterations (decrease) in platelet counts and Hb levels correlates with disease severity, i.e. the SFM patients had much lower levels of these hematological parameters compared to the NSFAM patients.

Revision made: We have amended this statement as per the suggestion from this reviewer. Interestingly, the magnitude of alterations (decrease) in platelet counts and Hb levels were found to be more prominent in severe malaria, i.e. the SFM patients had lower levels of these hematological parameters compared to the NSFAM patients.

Changes has been incorporated on following Page no.: 06

Additional comment:

I still believe the revision unclear.

'Interestingly, the magnitude of alterations (decrease) in platelet counts and Hb levels were found to be more prominent in severe malaria, i.e., the SFM patients had lower levels of these hematological parameters compared to the NSFAM patients.'

It is not all surprising that haematological features are worsen in severe malaria. In my mind, the simpler first section of the sentence is clear enough. 'The magnitude of alterations (decrease) in platelet counts and Hb levels were found to be more prominent in severe malaria'

Response: We highly appreciate this suggestion from the reviewer. We have revised the concerned sentence as per the suggestion from this reviewer.

Revision made: We have removed the 2nd part of the statement and kept only the first section of the sentence.

~~Interestingly,~~ The magnitude of alterations (decrease) in platelet counts and Hb levels were found to be more prominent in severe malaria. ~~i.e., the SFM patients had lower levels of these hematological parameters compared to the NSFEM patients.~~

Changes has been incorporated on following Page no.: 5

Comment 24:

Supplementary table 8 did not have the corresponding protein ID included

Response: We are most obliged to this reviewer for this critical observation and our sincere apologies for this inadvertent error. We have included the UniProt IDs for the proteins in the revised Supplementary Data 8.

Response to the Comments of the Reviewer # 3

The new revised manuscript by Kumar et al. has improved substantially and with the corrections made and the answers given has provided a satisfactory response to almost all the points that were raised. However, my points 1, 2 and 7 are not entirely resolved and should be reconsidered.

I will now comment on these points and the answers obtained:

Comment 1: With the new supplementary table provided it has been improved the perception of the samples used. Nevertheless, it's not yet clear how many are coincidental between each phase (if any) and how many of them were used for the analysis of clinical parameters. A complete table of all individual samples and its particular analytical use would certainly help to clarify it. For example, how many of the 26 in SA group were analyzed at TMT and MRM assays? It's also unclear why the figures for Discovery Phase (DP) samples in HC and SD groups are higher in the TMT column than in the MRM column at DP.

Response: We thank the reviewer for this suggestion. We have included further details regarding the sample number and study cohorts in the revised manuscript.

Revision made: We have provided the details of all samples used in each experiment, i.e., discovery phase, validation phase, and clinicopathological analyses in the Supplementary Data 1.

Supplementary Data 1: Details of the sample numbers used in the discovery phase (DP) quantitative proteomics, targeted validation phase (VP), and clinicopathological analyses.

Experimental groups	Discovery phase quantitative proteomics (TMT-based multiplexed quantitation)	Targeted validation (MRM assay)	Analysis of clinical parameters
Falciparum malaria (FM)			
Non-severe Falciparum malaria (NSFM)	8	10 (DP = 5, NS = 5)	15 (DP and VP = 5, DP = 3, VP = 4, NS = 3)
SFM_others	12	7 (DP = 4, NS = 3)	26 (DP and VP = 4, DP = 8, VP = 3, NS = 11)
Severe anemia malaria (SA)	6	4 (DP = 4, NS = 0)	5 (DP and VP = 4, DP = 1, NS = 0)
Cerebral malaria (CB)	6	3 (DP = 1, NS = 2)	7 (DP = 5, VP = 2)
Negative control: Control for severe anemia (CSA)	4	3 (DP = 3, NS = 0)	*
Negative control: Control for cerebral malaria (CCB)	4	3 (DP = 3, NS = 0)	*
Healthy Control (HC)	10	24 (DP = 10, NS = 14)	58 (DP and VP = 10, VP = 14, NS = 34)
Disease Control: Non-severe dengue (NSD)	4	10 (DP = 4, NS = 6)	5 (DP and VP = 4, VP = 1, NS = 0)
Disease Control: Severe dengue (SD)	12	17 (DP = 12, NS = 5)	10 (DP and VP = 10, NS = 0)
Vivax malaria (VM)			
Non-severe vivax malaria (NSVM)	12	10 (DP = 7, NS = 3)	*
Severe vivax malaria (SVM)	20	20 (DP = 20, NS = 0)	17 (DP and VP = 17, NS = 0)

Healthy Control (HC)	4	24 (DP = 4, NS = 20)	58 (DP and VP = 4, VP = 20, NS = 34)
Disease Control: Non-severe dengue (NSD)	4	10 (DP = 4, NS = 6)	5 (DP and VP = 4, VP = 1, NS = 0)
Disease Control: Severe dengue (SD)	12	17 (DP = 12, NS = 5)	10 (DP and VP = 10, NS = 0)

SFM_Others: It includes case of severe falciparum malaria patients excluding CB and SA complications.

*not considered for clinical analysis.

DP – subjects from discovery phase, VP – subjects from targeted validation phase, NS – new subjects.

Comment 2: This reviewer does not agree of not having this essential data of parasitemia levels at each analysed sample by PCR. If the authors do not have such information, then a caution notice should be provided in the results and discussion sections indicating that some possibilities of asymptomatic infection in the controls and other groups may remain in the analysed population and that it is a possible limitation of the study. Even high sensitivity RDTs can fail for many samples when from asymptomatic malaria and in many endemic areas asymptomatic malaria can range up to 50% with parasitaemia below 80 parasites per ul, and then many can pass as from healthy people.

Response: We thank the reviewer for this suggestion. We agree with the reviewer that even high sensitivity RDTs sometimes fail to detect asymptomatic malaria patients with very low levels of parasitaemia. We also accept that the lack of information regarding parasitaemia for each subject included in this study (particularly for the healthy and diseased control groups) is a potential caveat.

Revision made: We have included a cautionary sentence regarding this in the revised manuscript as suggested by this reviewer.

“One potential caveat to this study is the possibility for asymptomatic malaria infection in the controls, although these were entirely devoid of any clinical symptoms of malaria.”

Changes has been incorporated on following Page no.: 16-17

Comment 7: At this point the authors have only answered to the G6PD deficiency but not to the haemoglobinopathies. It is true that although it is difficult to establish the percentages of polymorphisms associated with population variants of hemoglobins and G6PD in India because of the particular characteristics of the population in this country, it can be considered according to the available literature that the carrier frequency of haemoglobinopathies varies between 3 and 17% in different populations of India. The cumulative gene frequency of the three most predominant abnormal haemoglobins, i.e. sickle cell, haemoglobin D and haemoglobin E has been found to be 5-6% in India. On the other hand, Beta-thalassemia trait is the predominant genetic Hb disorder accounting for >11% in some regions.

Having therefore accumulated percentages of these polymorphisms as a whole (G6PDs and haemoglobins) that can substantially modify the response to malaria and given that (as in the previous point) this lack of information can be a limitation of the study, I believe that the authors should be aware of this and mention a cautionary note in the discussion that the lack of genetic analysis of human samples allows the prediction of a certain percentage of polymorphisms associated with the response to malaria (G6PD and haemoglobins) that could bias part of the results obtained.

Response: We highly appreciate this constructive and insightful advice from the reviewer.

Revision made: We have incorporated the following changes in the revised manuscript as per the reviewer's suggestions.

“Moreover, genetic analysis of human plasma samples for glucose 6 phosphate dehydrogenase (G6PD) deficiency and haemoglobinopathies was outside the scope of this study. This leaves a lack of information on certain key host responses during the regulation of host proteins in response to the malaria parasites. These could be an effective future continuation of the present investigation, especially as the findings continue to move towards translational research.”

Changes has been incorporated on following Page no.: 16-17